

# Osteology of the axial skeleton of *Aucasaurus garridoi*: phylogenetic and paleobiological inferences

Mattia Antonio Baiano[1,2,3,4], Rodolfo Coria[4,5], Luis M. Chiappe[6], Virginia Zurriaguz[7] and Ludmila Coria[5]

[1] Chinese University of Hong Kong, Hong Kong, Hong Kong
[2] Consejo Nacional de Investigaciones Científicas y Técnicas (CONICET), Buenos Aires, Argentina
[3] Museo Municipal Ernesto Bachmann, Villa el Chocón, Argentina
[4] Universidad Nacional de Río Negro, General Roca, Argentina
[5] Museo Municipal Carmen Funes, Plaza Huincul, Argentina
[6] Dinosaur Institute, Natural History Museum of Los Angeles County, Los Angeles, United States of America
[7] Instituto de Investigación en Paleobiología y Geología (IIPG), General Roca, Argentina

Corresponding author
Mattia Antonio Baiano,
mbaiano@unrn.edu.ar

## ABSTRACT

*Aucasaurus garridoi* is an abelisaurid theropod from the Anacleto Formation (lower Campanian, Upper Cretaceous) of Patagonia, Argentina. The holotype of *Aucasaurus garridoi* includes cranial material, axial elements, and almost complete fore- and hind limbs. Here we present a detailed description of the axial skeleton of this taxon, along with some paleobiological and phylogenetic inferences. The presacral elements are somewhat fragmentary, although these show features shared with other abelisaurids. The caudal series, to date the most complete among brachyrostran abelisaurids, shows several autapomorphic features including the presence of pneumatic recesses on the dorsal surface of the anterior caudal neural arches, a tubercle lateral to the prezygapophysis of mid caudal vertebrae, a marked protuberance on the lateral rim of the transverse process of the caudal vertebrae, and the presence of a small ligamentous scar near the anterior edge of the dorsal surface in the anteriormost caudal transverse process. The detailed study of the axial skeleton of *Aucasaurus garridoi* has also allowed us to identify characters that could be useful for future studies attempting to resolve the internal phylogenetic relationships of Abelisauridae. Computed tomography scans of some caudal vertebrae show pneumatic traits in neural arches and centra, and thus the first reported case for an abelisaurid taxon. Moreover, some osteological correlates of soft tissues present in *Aucasaurus* and other abelisaurids, especially derived brachyrostrans, underscore a previously proposed increase in axial rigidity within Abelisauridae.

## INTRODUCTION

Abelisauridae is among the best known groups of non-avian theropods that reached the end of the Cretaceous (*Bonaparte, 1985*; *Wilson et al., 2003*; *Krause et al., 2007*; *Novas et al., 2010*; *Gasparini et al., 2015*). Abelisaurids are mostly known from Gondwanan landmasses, which have provided the best record in terms of abundance and specimen completeness (*e.g.*, *Krause et al., 2007*; *Novas et al., 2013*; *Zaher et al., 2020*). In contrast, the Laurasian record is scant; it is mostly derived from the Cretaceous of France (*Buffetaut, Mechin & Mechin-Salessy, 1988*; *Le Loeuff & Buffetaut, 1991*; *Accarie et al., 1995*; *Allain & Suberbiola, 2003*; *Tortosa et al., 2014*), although some putative abelisaurids have been reported from the Cretaceous of Hungary and Spain (*Ősi, Apesteguía & Kowalewski, 2010*; *Ősi & Buffetaut, 2011*; *Isasmendi et al., 2022*).

Since they were first discovered, abelisaurids were recognized as having a peculiar cranial anatomy and striking differences in their appendicular and axial skeleton when compared to other theropods. In particular, the axial skeleton shows traits, mostly in the vertebrae, which are unique of this group. Among Gondwanan abelisaurids, several taxa have preserved axial elements (*e.g.*, *Ekrixinatosaurus*, *Ilokelesia*, *Pycnonemosaurus*; *Coria & Salgado (2000)*; *Kellner & Campos, 2002*; *Calvo, Rubilar-Rogers & Moreno, 2004*, but only seven taxa have preserved complete portions (articulated or semi-articulated) of the vertebral series: *Aucasaurus*, *Eoabelisaurus*, *Carnotaurus*, *Majungasaurus*, *Skorpiovenator*, *Spectrovenator*, and *Viavenator* (*Bonaparte, Novas & Coria, 1990*; *Coria, Chiappe & Dingus, 2002*; *O'Connor, 2007*; *Canale et al., 2009*; *Pol & Rauhut, 2012*; *Filippi et al., 2016*; *Zaher et al., 2020*). Among them, detailed osteological descriptions of the vertebral column have been provided for *Carnotaurus* (*Méndez, 2014a*), *Majungasaurus* (*O'Connor, 2007*), and *Viavenator* (*Filippi et al., 2018*).

Here, we have carried out a detailed description of the axial skeleton of the holotype of *Aucasaurus garridoi* (MCF-PVPH-236), which is the second detailed study of the anatomy of this abelisaurid after the study of its braincase (*Paulina-Carabajal, 2011*). The axial skeleton of MCF-PVPH-236 is composed of cervical, dorsal, and caudal vertebrae, cervical and dorsal ribs, gastralia, and haemal arches. In spite of *Coria, Chiappe & Dingus (2002)* proposing a valid diagnosis for *Aucasaurus*, after the discovery of new abelisaurid species in the ensuing 20 years, we propose a new revised diagnosis using information from the axial skeleton. An exhaustive comparison between *Aucasaurus* and other abelisaurids, especially Argentinian specimens, has allowed us to detect several anatomical traits of the axial skeleton shared by these taxa, thus strengthening the diagnosis of Abelisauridae and adding new data for future phylogenetic analyses. We have also used computer tomographic (CT) scans of some caudal vertebrae to visualize their internal structure. We thus offer the first CT data of the axial skeleton of Abelisauridae, and investigate its pneumaticity. Finally, our detailed study of the axial anatomy has revealed traits in *Aucasaurus* and other brachyrostran abelisaurids that are functionally related to increased rigidity of the axial skeleton.

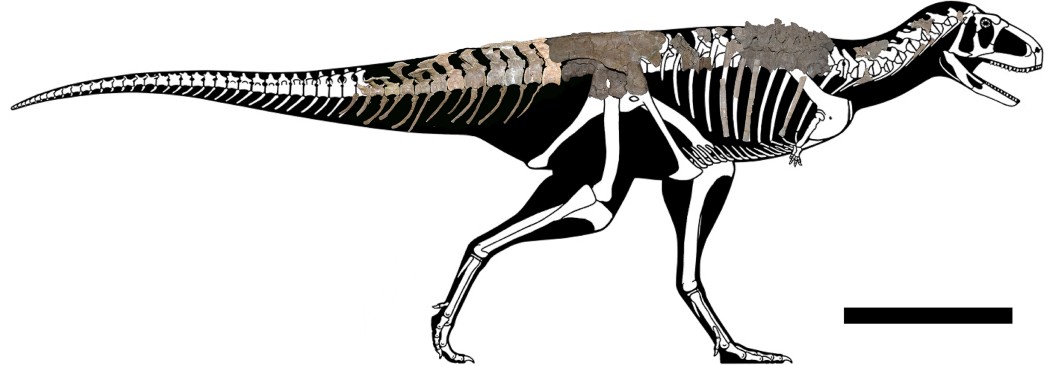

**Figure 1 Axial skeleton of *Aucasaurus garridoi*.** Lateral right view of the axial elements of the holotype MCF-PVPH-236. Scale bar: 1 m. Silhouette modified from Scott Hartman (https://www.skeletaldrawing.com).

## MATERIAL AND METHODS

The axial skeleton of the holotype of *Aucasaurus garridoi* (MCF-PVPH-236) includes the atlas and fragments of the cervical vertebrae, the second to seventh dorsal vertebrae, fragmentes of posterior dorsal vertebrae, the complete sacrum, the first to thirteenth caudal vertebrae, posterior caudal vertebrae, cervical and dorsal ribs, gastralia, and the first to thirteenth haemal arches (Fig. 1). We conducted a detailed comparison of MCF-PVPH-236 with several theropods, particularly Argentinian abelisauroids. In the case of specimens in which the position of the vertebrae was confidently identified, comparisons used the same vertebral element. However, in those cases in which the position of specific axial elements was not known with certainty, comparisons were carried out at a more regional level: anterior, middle, and posterior (see Discussion). Table 1 shows all taxa used in the present study (examined directly or whose data were taken from the literature). We followed the anatomical nomenclature of *Wilson (1999, 2012)* and *Wilson et al. (2011)* to describe laminae and fossae. These structures are spelled out when first mentioned in the text (plus acronym), subsequently they are cited only using their acronyms.

All measurements were taken using a digital calliper (Tables S1–S3) and images for figures (both single pothographs and photogrammetry renderings) were captured using a Nikon 3100 digital camera.

To test the phylogenetic position of *Aucasaurus* based on new axial information, we carried out an analysis based on the most recently studies of Ceratosauria (*Tortosa et al., 2014*; *Filippi et al., 2016*; *Rauhut & Carrano, 2016*; *Baiano, Coria & Cau, 2020*; *Baiano et al., 2021, 2022*; *Aranciaga Rolando et al., 2021*; *Gianechini et al., 2021*; *Cerroni et al., 2022*). We added 11 characters (seven new and four from other sources) to the data matrices of *Baiano et al. (2022)* and *Cerroni et al. (2022)*; we also added three new taxa (*i.e.*, *Kurupi*, *Thanos*, and MPM 99). The resulting data matrix consisted of 246 characters and 46 taxa (Data S1). Moreover, we provided 17 new scorings (either missing data or previously scored characters) for *Aucasaurus* (characters 96, 98, 107, 112, 115, 116, 117, 119, 120, 121, 123, 123, 128, 133, 134, 136, 137). The data matrix (Data S2) was edited in MESQUITE 3.61 (*Maddison & Maddison, 2019*). The analysis was performed using TNT

**Table 1 Taxa used for anatomical comparisons.**

| Taxa examined directly | Specimen no. | First reference |
|---|---|---|
| *Arcovenator escotae* | MHNA-PV-2011.12.5/198/213 | *Tortosa et al. (2014)* |
| *Aucasaurus garridoi* | MCF-PVPH-236 | *Coria, Chiappe & Dingus (2002)* |
| *Carnotaurus sastrei* | MACN-PV-CH 894 | *Bonaparte (1985)* |
| *Ekrixinatosaurus novasi* | MUC Pv 294 | *Calvo, Rubilar-Rogers & Moreno (2004)* |
| *Elemgasem nubilus* | MCF-PVPH-380 | *Baiano et al. (2022)* |
| *Eoabelisaurus mefi* | MPEF Pv 3990 | *Pol & Rauhut (2012)* |
| *Huinculsaurus montesi* | MCF-PVPH-36 | *Baiano, Coria & Cau (2020)* |
| *Ilokelesia aguadagrandensis* | MCF-PVPH-35 | *Coria & Salgado (2000)* |
| *Niebla antiqua* | MPCN-PV-796 | *Aranciaga Rolando et al. (2021)* |
| *Skorpiovenator bustingorryi* | MMCh-PV 48 | *Canale et al. (2009)* |
| *Tralkasaurus cuyi* | MPCA-PV 815 | *Cerroni et al. (2020)* |
| *Viavenator exxoni* | MAU-PV-LI 530 | *Filippi et al. (2016)* |
| *Xenotarsosaurus bonapartei* | UNPSJB-PV 612/1-2 | *Martínez et al. (1986)* see also *Ibiricu et al. (2021)* |
| Abelisauridae indet. | MACN-PV-RN 1012 | *Ezcurra & Méndez (2009)* |
| Abelisauridae indet. | MAU-Pv-LI 547 | *Méndez et al. (2018)* |
| Abelisauridae indet. | MAU-Pv-LI 665 | *Méndez et al. (2022)* |
| Abelisauridae indet. | MCF-PVPH-237 | *Coria, Currie & Paulina-Carabajal, 2006* |
| Abelisauridae indet. | MMCh-PV 69 | *Canale et al. (2016)* |
| Abelisauridae indet. | MPCN-PV-69 | *Gianechini et al. (2015)* see also *Baiano et al. (2021)* |
| Abelisauridae indet. | MPM 99 | *Martínez, Novas & Ambrosio (2004)* |
| Abelisauroidea indet. | MPEF PV 1699/1-2 | *Rauhut et al. (2003)* |
| **Taxa drawn from literature** | **Source** | **First reference** |
| *Aerosteon riocoloradensis* | *Aranciaga Rolando et al. (2022)* | *Sereno et al. (2008)* |
| *Allosaurus fragilis* | *Madsen (1976)* | *Marsh (1877)* |
| *Camarillasaurus cirugedae* | *Sánchez-Hernández & Benton (2012)* | *Sánchez-Hernández & Benton (2012)* |
| *Ceratosaurus sp* | *Gilmore (1920)*, *Madsen & Welles (2000)* | *Gilmore (1920)* |
| *Dahalokely tokana* | *Farke & Sertich (2013)* | *Farke & Sertich (2013)* |
| *Dilophosaurus wetherilli* | *Welles (1984)*, *Marsh & Rowe (2020)* | *Welles (1954)* |
| *Elaphrosaurus bambergi* | *Rauhut & Carrano (2016)* | *Janensch (1920, 1925)* |
| *Herrerasaurus ischigualastensis* | *Sereno & Novas (1994)* | *Reig (1963)* |
| *Kurupi itaata* | *Iori et al. (2021)* | *Iori et al. (2021)* |
| *Majungasaurus crenatissimus* | *O'Connor (2007)* | *Depéret (1896)*, *Lavocat (1955)* |
| *Masiakasaurus knopfleri* | *Carrano, Sampson & Forster (2002)*, *Carrano, Loewen & Sertich (2011)* | *Sampson, Carrano & Forster, 2001* |
| *Pycnonemosaurus nevesi* | *Delcourt (2017)* | *Kellner & Campos (2002)* |
| *Rahiolisaurus gujaratensis* | *Novas et al. (2010)* | *Novas et al. (2010)* |
| *Rajasaurus narmadensis* | *Wilson et al. (2003)* | *Wilson et al. (2003)* |
| *Sinraptor dongi* | *Currie & Zhao (1993)* | *Currie & Zhao (1993)* |
| *Spectrovenator ragei* | *Zaher et al. (2020)* | *Zaher et al. (2020)* |

| Table 1 (continued) | | |
|---|---|---|
| Taxa examined directly | Specimen no. | First reference |
| *Thanos simonattoi* | *Delcourt & Iori (2018)* | *Delcourt & Iori (2018)* |
| *Tyrannosaurus rex* | *Brochu (2003)* | *Osborn (1905)* |
| Abelisauroidea indet. CPP 893 | *Novas et al. (2008)* | *Novas et al. (2008)* |

1.5 (*Goloboff, Farris & Nixon, 2008*; *Goloboff & Catalano, 2016*), conducting a traditional search through 1,000 replicates of Wagner trees (saving 10 trees per replicate) followed by tree bisection–reconnection (TBR) branch swapping. The memory to store all most parsimonious trees (MPTs) was implemented to 50,000. The MPTs obtained were submitted to a second round of TBR. All characters were weighted equally. To detect possible unstable taxa, we performed the IterPCR procedure (*Pol & Escapa, 2009*), and used Bremer support and Jackknife value through the pcrjack.run script to assess nodal support (*Pol & Goloboff, 2020*).

We CT scanned six caudal vertebrae (*i.e.*, first, fifth, sixth, ninth, twelfth, and thirteenth) to investigate their internal structure. The CT scans was performed using a Toshiba Aquilion Lightnight 16/32 scanner, in the Sanatorio Plaza Huincul in Plaza Huincul (Neuquén Province, Argentina). The CT scans were carried out along the transversal, coronal, and sagittal planes with the following settings: 120 kVp, 50 mA, and slices each 5-mm. The number of slices for each vertebra is: 36 coronal slices, 11 transversal slices, and 23 sagittal slices for the first caudal; 44 coronal slices, 12 transversal slices, and 23 sagittal slices for the fifth and sixth caudals; 30 coronal slices, nine transversal slices, and 23 sagittal slices for the ninth caudal; and 36 coronal slices, seven sagittal slices, and 19 sagittal slices for the twelfth and thirteenth caudals. The slices were observed using the K-PACS software produced by Ebit (ESAOTE).

## SYSTEMATIC PALAEONTOLOGY

Dinosauria Owen, 1842

Saurischia Seeley, 1887

Theropoda Marsh, 1881

Ceratosauria Marsh, 1884

Abelisauroidea Bonaparte & Novas, 1985

Abelisauridae Bonaparte & Novas, 1985

Brachyrostra *Canale et al., 2009*

*Aucasaurus Coria, Chiappe & Dingus, 2002*

### Etymology

The generic name was established by *Coria, Chiappe & Dingus (2002)*; in reference to Auca Mahuevo, the fossil locality in which the holotype was found, with the Greek suffix -σαῦρος (sauros), lizard or reptile.

### Diagnosis

As for the species.

*Aucasaurus garridoi* Coria, Chiappe & Dingus, 2002

### Type species and etymology

The name of the type species was erected in recognition to geologist Alberto Garrido, who discovered the holotype.

### Holotype

MCF-PVPH-236, Museo Carmen Funes (Plaza Huincul, Neuquén Province, Argentina), a partial skeleton including cranial, axial, and appendicular elements (see *Coria, Chiappe & Dingus, 2002*).

### Locality and horizon

Auca Mahuevo paleontological site (*Chiappe et al., 1998*), near Mina La Escondida, in the northeastern corner of Neuquén Province, Argentina. The holotype was recovered from strata belonging to the Anacleto Formation (lower Campanian, Upper Cretaceous), Río Colorado Subgroup, Neuquén Group of the Neuquén Basin. Sedimentological and stratigraphic descriptions of these strata and of the Anacleto Formation are provided elsewhere (see *Dingus et al., 2000*; *Coria, Chiappe & Dingus, 2002*; *Garrido, 2010a*, *2010b*).

### Comments on the original diagnosis

The original diagnosis established by *Coria, Chiappe & Dingus (2002)* was largely based on morphological comparisons with *Carnotaurus* and mentioning only one autapomorphy (*i.e.*, anterior haemal arches with proximally opened neural canal). Here, we expand the diagnosis to include the following unique features of the axial skeleton: (1) atlas with a subcircular articular surface; (2) interspinous accessory processes extended to sacral and caudal neural spine; (3) presence of a tubercle lateral to the prezygapophysis of mid caudal vertebrae (a similar structure is mentioned in *Aoniraptor*; *Motta et al., 2016*); (4) presence of pneumatic foramina laterally to the base of the neural spine in the anterior caudal vertebrae; (5) presence of a prominent tubercle and extensive rugosity on the lateral rim of the transverse processes of caudal vertebrae fourth to twelfth; (6) presence of a small ligamental scar near the anterior edge of the dorsal surface in the anteriormost caudal transverse processes; (7) distinct triangular process located at the fusion point of posterior middle gastralia. In addition, according to *Coria, Chiappe & Dingus (2002)*, the skull of *Aucasaurus* differs from that of *Carnotaurus sastrei* in having a longer and lower rostrum, frontal swells instead of horns, and a sigmoidal outline of the dentigerous margin of the maxilla. Several postcranial differences also distinguish *Aucasaurus garridoi* from *Carnotaurus sastrei*: a less developed coracoidal process, a forelimb relatively longer, a humerus with a slender and craniocaudally compressed shaft and well-defined condyles, and a proximal radius lacking a hooked ulnar process.

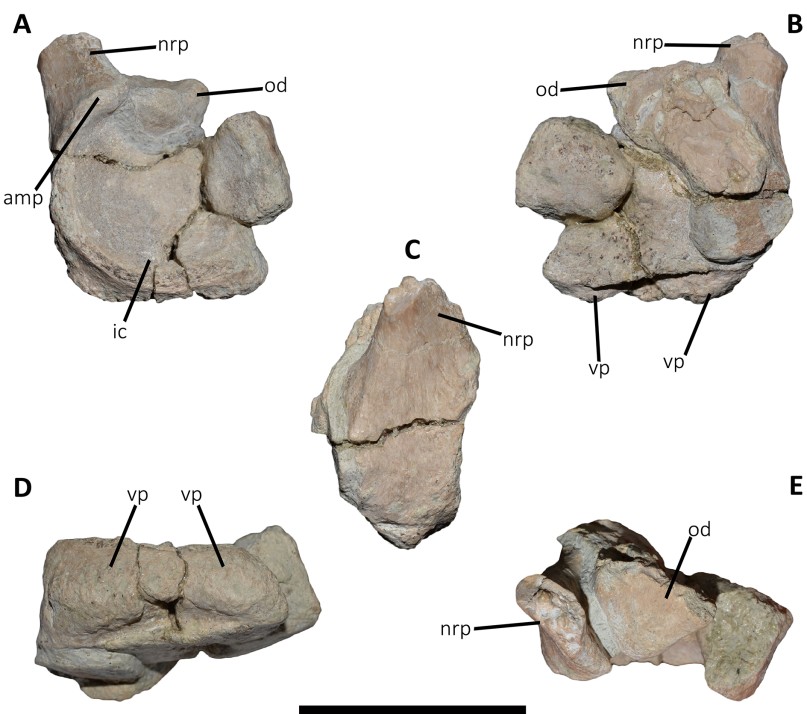

**Figure 2 Atlas of *Aucasaurus garridoi* MCF-PVPH-236.** In anterior (A), posterior (B), right lateral (C), ventral (D), and dorsal (E) views. amp, anteromedial process; ic, intercentrum; nrp, neurapophysis; od, odontoid; vp, ventral process. Scale bar: 5 cm.

# DESCRIPTION AND COMPARISONS

*Cervical Vertebrae* (Figs. 2 and 3): An almost complete atlas and several cervical fragments are preserved. The most notable piece is a right neural arch that could belong to the fifth cervical vertebra. The other remains are identified as part of isolated epipophyses.

*Atlas* (Fig. 2; Table S1): The atlas preserves the intercentrum with a fused portion of the right neurapophysis (Figs. 2A–2C). In anterior view (Fig. 2A), the articular surface for the occipital condyle is strongly concave and subcircular, which differs from the slightly transversely wider than tall atlas of *Skorpiovenator* (Mattia A. Baiano, 2018, personal observation on MMCh-PV 48) and *Viavenator* (see also Discussion, in particular the paragraph on the autapomorphic axial traits of *Aucasaurus*), and from the strongly dorsoventrally compressed atlas of *Carnotaurus*, *Ceratosaurus*, and some tetanurans (*e.g.*, *Allosaurus*, *Sinraptor*). The concave dorsal edge preserves the odontoid process in artculation. The right neurapophysis is directed dorsolaterally, and a hook-shaped process directed anteromedially on its ventromedial part seems less developed than in *Ceratosaurus*, *Majungasaurus*, *Skorpiovenator*, *Viavenator*, and *Carnotaurus*. The absence of prezygapophyses suggests that *Aucasaurus* lacked a proatlas as in *Majungasaurus*, *Skorpiovenator*, *Viavenator*, and *Carnotaurus*.

In posterior view (Fig. 2B), the articular surface is flat as in *Viavenator*, but different from the convex surface in *Ceratosaurus*, *Carnotaurus*, and some tetanurans (*e.g.*, *Allosaurus*, *Sinraptor*). The posterior articular surface is stepped due to two parapophyseal

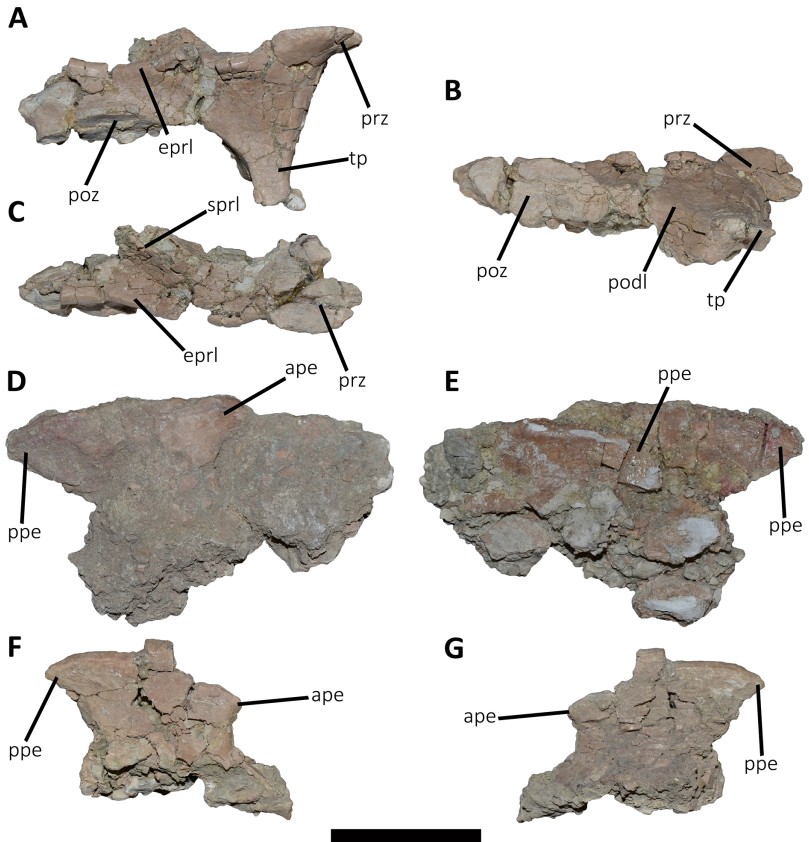

**Figure 3 Cervical vertebra fragments of *Aucasaurus garridoi* MCF-PVPH-236.** In lateral (A, G and E), ventral (B), dorsal (C), and medial (D and F) views. ape, anterior process of epipophysis; eprl, epipophyseal prezygapophyseal lamina; podl, postzygodiapophyseal lamina; poz, postzygapophysis; ppe, posterior process of epipophysis; prz, prezygapophysis; sprl, spinoprezygapophyseal lamina; tp, transverse process. Scale bar: 5 cm.

processes located on the ventral edge. In this view, the pneumatic internal arrangement can be visualized through a break in the odontoid process. There are several small chambers, resembling a camellate condition.

In lateral view (Fig. 2C), the surface has a rectangular outline and is slightly dorsoventrally concave, although it slightly narrows ventrally. The neurapophysis is firmly fused to the intercentrum and there are no visible sutures. The posterior border of the neurapophysis forms a ridge that ends ventrally in the intercentrum.

In ventral view (Fig. 2D), the surface presents two ventrally directed processes as seen in *Skorpiovenator*, *Viavenator*, and *Carnotaurus*, which could be interpreted as parapophysis-like structures for rib articulation. However, in *Aucasaurus* these processes are separated by a more superficial groove than in *Viavenator* and *Carnotaurus*.

In dorsal view (Fig. 2E), the poor preservation of the neurapophyses prevents either the evaluation of its extension, or an assessment of the morphology of the postzygapophysis and medial process. The preserved portion of the neurapophysis has an oval cross-section, although it narrows slightly anteriorly. The neurapophysis is slightly twisted with its greater axis anteromedially-posterolaterally directed. A fragment of the odontoid process

is preserved on the dorsal part of the atlas. It has a triangular shape in dorsal view, different from the more circular outline of this structure in *Ceratosaurus*, *Masiakasaurus*, *Thanos*, and *Carnotaurus*, whereas *Majungasaurus* shows an intermediate condition between *Aucasaurus* and other abelisauroids (see also Discussion, in particular the paragraph on the autapomorphic axial traits in *Aucasaurus*). Therefore, the condition present in *Aucasaurus* is here considered an autapomorphy of *Aucasaurus*. The dorsal surface of odontoid is concave, while the lateral and ventral surfaces are strongly convex to fit in the dorsal edge of the intercentrum.

*Middle cervical vertebra (Cv-05?)* (Figs. 3A–3C): Only the right lateral portion of the neural arch is preserved. In anterior view, the prezygapophysis has a flat, dorsomedially sloping facet as in *Dahalokely*, *Carnotaurus*, *Ilokelesia*, *Majungasaurus*, *Skorpiovenator*, *Viavenator*, and MPM 99.

In lateral view (Fig. 3A), a well-defined epipophyseal-prezygapophyseal lamina (eprl) connects the prezygapophysis with the epipophysis, separating the lateral part of the transverse process from the dorsal part of the neural arch, as in other abelisauroids (*e.g.*, *Carrano & Sampson, 2008*). This lamina, although broken in some parts, is straight as in *Majungasaurus*, *Viavenator*, and *Carnotaurus*, but unlike *Dahalokely* where it is strongly convex. Furthermore, in *Aucasaurus*, the posteriormost part of the eprl seems to be dorsally directed, though we cannot assess if it was less dorsally inclined as in *Majungasaurus* or oblique as in *Carnotaurus*. The transverse process is triangular in outline and directed ventrally. It has a flat, lateral surface with a straight prezygodiapophyseal lamina (prdl) and a concave postzygodiapophyseal lamina (podl). The latter is developed as a faint crest (Fig. 3B), which is a condition observed in abelisaurids such as *Skorpiovenator* and *Ilokelesia*. The postzygapophysis is partially preserved and positioned 1.5 cm from the podl. The postzygapophysis has a flat articular facet, is directed ventrolaterally, and is anteroposteriorly longer than mediolaterally wide (Fig. 3B). However, the medial border is partially broken, suggesting that it also extended medially with a teardrop-like outline. The base of an epipophysis is preserved dorsally to the postzygapophysis.

In dorsal view (Fig. 3C), a slight depression separates the prezygapophysis from a robust spinoprezygapophyseal lamina (sprl) that preserves only the base. This lamina has an anterolateral-posteromedial orientation. The prezygapophysis shows a drop-like outline, having the widest part located laterally as other abelisaurids (*e.g.*, *Dahalokely*, *Carnotaurus*, *Ilokelesia*, *Majungasaurus*, *Viavenator*).

*Other cervical remains* (Figs. 3D–3G): Several fragments of epipophyses are preserved. Two of them contacting to each other (Figs. 3D and 3E). The dorsal edges of the epipophyses are slightly convex, transversely thicker than the body and with a rough surface. At least one epipophysis shows anterior and posterior processes as in *Noasaurus*, *Rahiolisaurus*, *Viavenator*, and *Carnotaurus*, in contrast to other abelisaurids that present only a posterior process (*e.g.*, *Ilokelesia*, *Skorpiovenator*, *Spectrovenator*).

An epipophysis probably belonging to either the eighth or the ninth cervical vertebra is preserved (Figs. 3F and 3G). It has an anteroposteriorly reduced posterior process. Beneath it, the postzygapophysis is partially crushed. Most likely, the epipophyses had medially

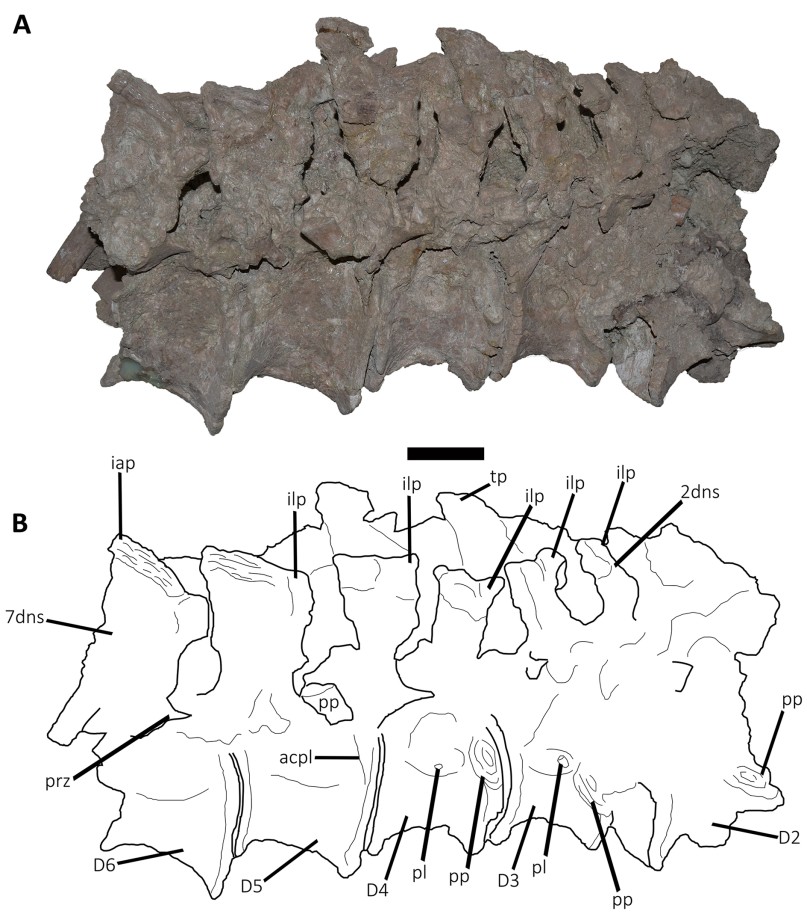

**Figure 4 (A–B) Photographs and line drawings of the anterior dorsal vertebrae of *Aucasaurus garridoi* MCF-PVPH-236.** In lateral (A) view. 2dns, second dorsal neural spine; 7dns, seventh dorsal neural spine; acpl, anterior centroparapophyseal lamina; D2–D6, second to seventh dorsal vertebrae; iap, interspinous accessory process; ilp, interspinous ligament process; pl, pleurocoel; pp, parapophysis; prz, prezygapophysis; tp, transverse process. Scale bar: 5 cm.

converging anterior processes. The hypertrophied epiphyses of *Aucasaurus* and other abelisaurids (*e.g.*, *Viavenator*, *Carnotaurus*) served as the point of origin of the *m. complexus* (on the anterior process), and the attachment point of the *m. longus colli dorsalis* (on the posterior process) (*Snively & Russell, 2007*; *Méndez, 2012*; *González, Baiano & Vidal, 2021*).

*Dorsal Vertebrae* (Figs. 4–7): The preserved dorsal vertebrae are very fragmentary. A series of articulated anterior dorsal vertebrae are regarded to range from the second to the seventh dorsal based on the morphology of the neural spines and the position of the parapophyses. In addition, a posterior dorsal vertebra, a posterior vertebral centrum, and several distal fragments of posterior dorsal neural spines are also preserved.

*Second dorsal vertebra* (D2; Figs. 4A, 4B and 5A–5D; Table S1): The second dorsal vertebra is badly preserved. The centrum is severely cracked and transversely crushed. Part

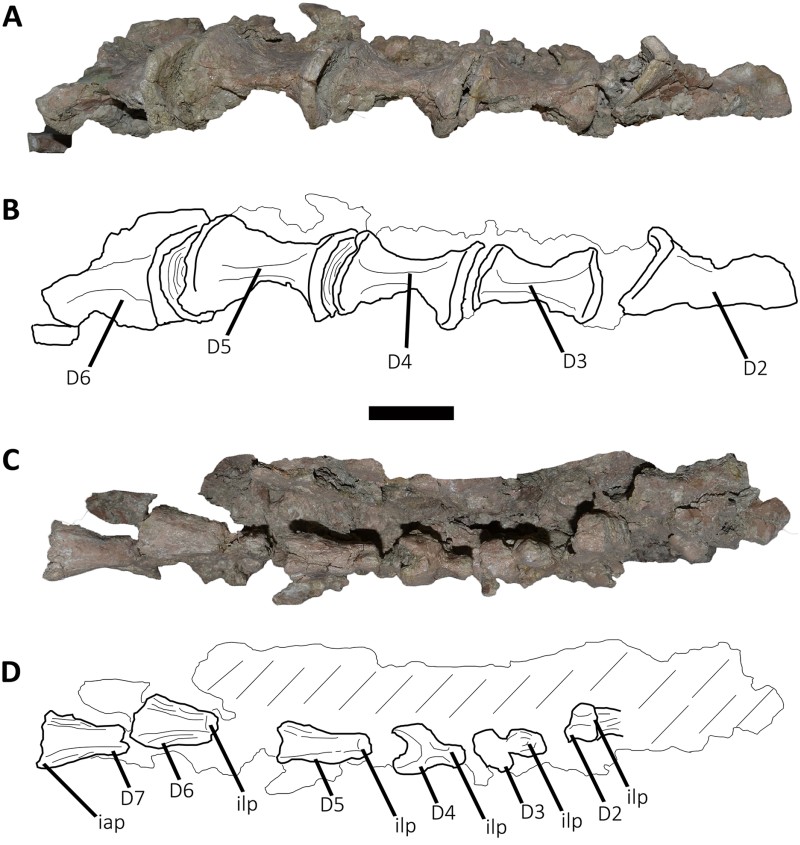

**Figure 5 Photographs and line drawings of the anterior dorsal vertebrae of *Aucasaurus garridoi* MCF-PVPH-236.** In ventral (A and B), and dorsal (C and D) views. Abbreviations: D2–D7, second to seventh dorsal vertebrae; iap, interspinous accessory process; ilp, interspinous ligament process. Scale bar: 5 cm.

of the anterior articular surface and the lateral surface are missing. The neural arch is almost entirely missing, except for the neural spine, which was posteriorly displaced.

The anterior articular surface is concave and dorsoventrally higher than transversely wide, probably due to taphonomic deformation. The right parapophysis is partially preserved. It is low and probably had a dorsoventral elliptical outline as in *Carnotaurus*, *Dahalokely*, *Skorpiovenator*, and *Xenotarsosaurus*. The posterior articular surface seems to be a little more complete than the anterior one (Figs. 4A and 4B). It is strongly concave and shows an elliptical contour probably due lateral compression. The ventral surface shows neither a groove nor a keel (Figs. 5A and 5B) as in *Dahalokely*, *Skorpiovenator*, and *Xenotarsosaurus*, but unlike *Elaphrosaurus* and *Majungasaurus* where there is a faint keel. Conversely, *Carnotaurus* and *Viavenator* have two longitudinal crests converging posteriorly.

The neural spine is transversely wider than anteroposteriorly long, being less than one third of the centrum length as in *Carnotaurus*, *Skorpiovenator*, and *Viavenator*, but shorter than in *Dahalokely*. The lateral surface of the spine is slightly concave anteroposteriorly (Figs. 4A and 4B), thus the anterior and posterior edges are more laterally protuding. The neural spine is distally thick and presents a reduced anterior process for the insertion

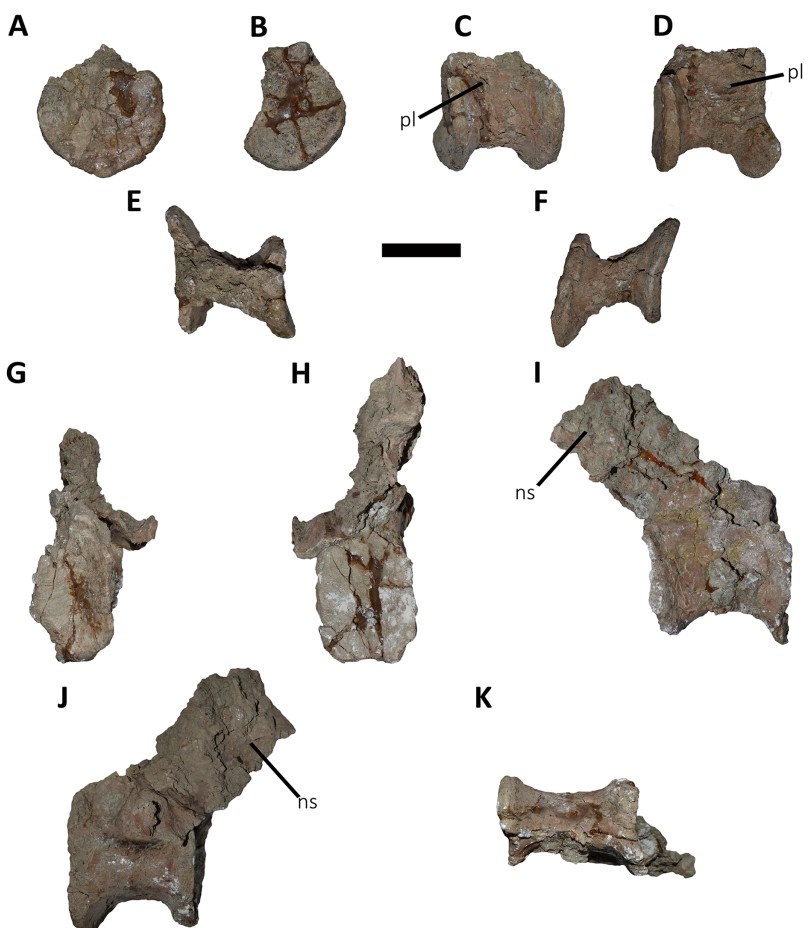

**Figure 6 Posterior dorsal vertebrae of *Aucasaurus garridoi* MCF-PVPH-236.** In anterior (A and G), posterior (B and H), lateral (C, D, I and J), dorsal (E), and ventral (F and K) views. ns, neural spine; pl, pleurocoel. Scale bar: 5 cm.

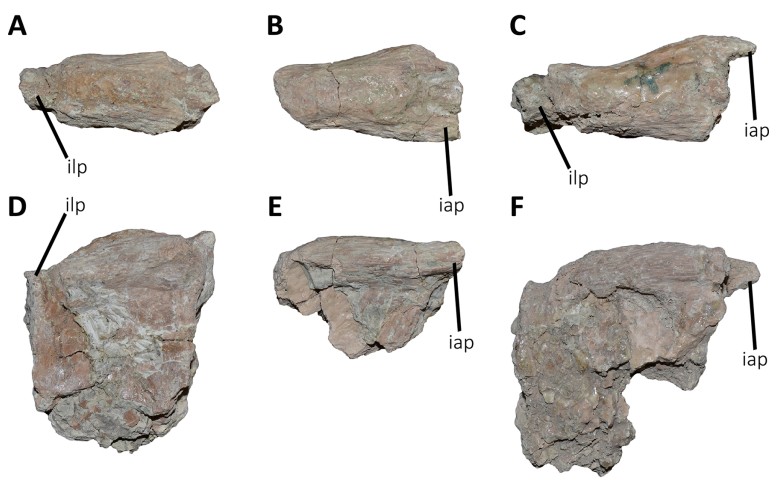

**Figure 7 Distal fragments of dorsal neural spines of *Aucasaurus garridoi* MCF-PVPH-236.** In dorsal (A–C), and left lateral (D–F) views. iap, interspinous accessory process; ilp, interspinous ligament process. Scale bar: 5 cm.
of interspinous ligaments. This process is separated from the rest of the spine by two lateral grooves. In dorsal view (Figs. 5C and 5D), a small process projects posteriorly.

*Third dorsal vertebra* (D3; Figs. 4A, 4B and 5A–5D; Table S1): The third dorsal vertebra is better preserved than the preceeding one, although it presents a significant transversal deformation and several fractures.

The anterior articular surface of the centrum is slightly concave but its articulation with the preceeding vertebra obscures other anatomical features. In lateral view (Figs. 4A and 4B), the anterior and posterior rims are parallel to each other. The parapophysis is positioned more dorsally than the previous vertebra and is elliptical in outline as in *Eoabelisaurus*, *Majungasaurus*, *Skorpiovenator*, and *Carnotaurus*, but its ventral part is slightly narrower anteroposteriorly than the dorsal one. The long axis of the parapophysis is slightly inclined posteriorly as in *Carnotaurus* and *Masiakasaurus*, but different from the dorsoventrally oriented parapophysis of *Eoabelisaurus* and *Majungasaurus*. Posterodorsally to the parapophysis and below the neurocentral suture, there is an anteroposterior oval fossa on the lateral surface. In the anterior corner of that fossa, there is a circular pleurocoel, which in turn is separated dorsally from two other small foramina by a septum. An anterior pleurocoel is also present in *Carnotaurus*, *Majungasaurus*, *Xenotarsosaurus*, and *Skorpiovenator* (the latter have also a posterior one). In posterior view, the articular surface is covered by the centrum of the next vertebra. However, a reduced part is exposed, showing a concave surface. In ventral view (Figs. 5A and 5B), the surface has neither a keel nor a groove as *Eoabelisaurus* and *Skorpiovenator*; in contrast, a faint keel is present in *Elaphrosaurus*.

The anterior surface of the neural spine has a dorsal process that protrudes anteriorly for the anchorage of interspinous ligaments. In lateral view (Figs. 4A and 4B), the right transverse process is not preserved. However, the anterior centrodiapophyseal lamina (acdl), the posterior centrodiapophyseal lamina (pcdl) and the centrodiapophyseal fossa (cdf) (or the centroparapophyseal fossa; cpaf) are visible. The neural spine is anteroposteriorly longer than the previous one, with a square cross-section, but it is shorter than the half of the centrum length as in *Carnotaurus* and *Majungasaurus*, whereas in *Eoabelisaurus* is slightly longer. Laterally, the anterodorsal process for the interspinous ligaments is visible. The two lateral grooves that separate this process from the rest of the dorsal neural spine are deeper than in the D2 (Figs. 5C and 5D). The interspinous ligamental process is also present in *Carnotaurus* and *Eoabelisaurus*, but more ventrally positioned than in *Aucasaurus* and *Skorpiovenator*. Lateral to the interspinous ligamental process, there is another process projected anteriorly as in *Eoabelisaurus*. In posterior view, only the right postzygapophysis can be observed, which, despite being articulated with the prezygapophysis of the next vertebra, seems to be anteroposteriorly longer than transversely wide.

*Fourth dorsal vertebra* (D4; Figs. 4A, 4B and 5A–5D; Table S1): The centrum of the fourth dorsal vertebra is slightly anteroposteriorly larger than that of the D3 (Figs. 4A and 4B). Both articular surfaces are slightly concave and, despite the deformation, probably were dorsoventrally taller than transversely wide. The lateral surface of the centrum presents a wide fossa with a pleurocoel located more centrally than that of the D3, unlike

*Carnotaurus*, *Majungasaurus*, *Skorpiovenator*, *Viavenator*, and MAU-Pv-LI 665, which hold a more anterior pleurocoel, whereas *Rajasaurus* lacks pneumatic opening in the centrum of this dorsal. The parapophysis is shifted more dorsally, between the centrum and neural arch, as in *Carnotaurus*, *Eoabelisaurus*, *Rajasaurus*, *Skorpiovenator*, and MAU-Pv-LI 665, but different than in *Viavenator* that holds parapophyses entirely on the neural arch and more laterally projected. The ventral surface lacks keel or groove (Figs. 5A and 5B), as in *Carnotaurus*, *Eoabelisaurus*, but unlike *Viavenator* that has a shallow groove, and *Rajasaurus* and MAU-Pv-LI 665 that hold a longitudinal keel.

In anterior view, only the neural spine is visible, which is transversely narrower than that of the D3. The anterodorsal process of the neural spine for the interspinous ligaments is conspicuous and has a rough surface, as in *Viavenator* but unlike *Carnotaurus*, *Eoabelisaurus*, *Majungasaurus* where it is poorly developed, or even absent in *Skorpiovenator*.

In lateral view (Figs. 4A and 4B), the ventral terminus of the right acdl and pcdl are visible and diverge from each other, reaching the arch pedicels. These laminae frame a triangular centrodiapophyseal (or centroparapophyseal) fossa. The right prezygapophysis is articulated with the postzygapophysis of the D3, preventing to see its morphology. However, it seems to be anteroposteriorly longer than mediolaterally wide and tilted medially. The prezygapophysis does not have any ventral process, attributable as the lateral wall of the hypantrum, such as the one present in *Carnotaurus* and *Skorpiovenator*. This condition differs from *Eoabelisaurus*, *Majungasaurus*, and *Viavenator* that have an incipient ventral process. The lateral surface of the neural spine is slightly concave and it is the first neural spine that is longer than transversely wide, as in *Eoabelisaurus*, *Majungasaurus*, and *Skorpiovenator*. This condition differs from the wider than long neural spine of *Carnotaurus*, whereas in *Viavenator* is square in cross-section. The dorsal end of the neural spine presents a transversal thickening and a marked anterodorsal process for the interspinous ligaments. This structure is anteriorly projected, unlike the neural spine of D3 where it protrudes dorsally over the dorsal surface of the neural spine. The two grooves that separate it from the neural spine are deep, different from *Carnotaurus*, *Eoabelisaurus*, *Majungasaurus*, *Skorpiovenator*, and *Viavenator* where there are no grooves.

In posterior view, only the right postzygapophysis, articulated with the prezygapophysis of D5, was preserved. As in the preceeding vertebrae, the postzygapophysis is longer than wide and the articular facet is slightly ventrolaterally oriented, differing from the horizontal postzygapophysis of *Majungasaurus*, *Rajasaurus*, *Carnotaurus*, *Skorpiovenator*, *Viavenator*, and MAU-Pv-LI 665.

In dorsal view (Figs. 5C and 5D), the neural spine has a Y-shaped outline, due to the lateral grooves separating the anterior process and a strong concavity between two partially broken posterior processes. This morphology differs from that of other abelisaurids, since these taxa either lack or have a reduced interspinous ligamental process. Furthermore, in *Aucasaurus* the anterior process for the interspinous ligaments is anteroposteriorly longer than in other abelisaurids.

*Fifth dorsal vertebra* (D5; Figs. 4A, 4B and 5A–5D; Table S1): In the fifth dorsal vertebra the centrum is almost complete (although deformed), whereas the neural arch is incomplete. Also, this vertebra presents an anterior diagenetical displacement of the neural spine (Figs. 4A and 4B).

The anterior and posterior articular surfaces are concave and elliptical in outline with their long axis directed dorsoventrally, as in *Eoabelisaurus*, *Majungasaurus*, *Skorpiovenator*, and CPP 893, but different from *Carnotaurus* and *Viavenator* where the centrum is subcircular. The lateral surface of the centrum holds a shallower fossa than in D4, and it lack pleurocoels (Figs. 4A and 4B), as in *Eoabelisaurus* and *Majungasaurus*, but in contrast to *Carnotaurus*, *Skorpiovenator*, *Viavenator*, and CPP 893 where there are fossae with pleurocoels. The parapophysis is located on the neural arch, as in *Carnotaurus*, *Eoabelisaurus*, *Majungasaurus*, *Skorpiovenator*, *Viavenator*, and CPP 893. The ventral facet has neither a groove nor a keel (Figs. 5A and 5B), as in *Eoabelisaurus*, *Skorpiovenator*, and *Viavenator*, but different from the longitudinal crest present in *Carnotaurus*.

In anterior view, similar to the preceeding vertebrae, the articulation prevents the evaluation of various morphological characteristics of the neural arch. Ventrolateral to the right prezygapophysis there is a shallow centroprezygapophyseal fossa (cprf). This fossa is incipient in *Carnotaurus* and absent in *Eoabelisaurus*, *Majungasaurus*, and *Viavenator*. The prezygapophysis is subquadrangular and the articular facet is directed slightly dorsolaterally, as in *Carnotaurus*, *Eoabelisaurus*, *Majungasaurus*, *Skorpiovenator*, *Viavenator*, and CPP 893. The prezygapophysis of *Aucasaurus* lacks the ventral columnar process present in *Carnotaurus*, *Majungasaurus*, *Skorpiovenator*, *Viavenator*, and CPP 893. The anterior process for the interspinous ligaments of the neural spine is present, but it is less developed than that of the D4.

In lateral view (Figs. 4A and 4B), the prezygapophysis lacks a ventral process, which is present in *Carnotaurus* and *Skorpiovenator*. Despite both transverse processes are lost, the anterior centroparapophyseal lamina (acpl) is visible. This lamina is robust and ends dorsally into the parapophysis. The parapophysis is not located in its original position, due to a dorsal and posterior displacement. However, it is a pendant structure as in other abelisaurids. The parapophysis has an oval contour, as in *Carnotaurus*, *Eoabelisaurus*, *Skorpiovenator*, and *Viavenator*. The neural spine, as mentioned above, is displaced anteriorly. It is dorsoventrally taller than in the D4, and the thick distalmost portion is separated from the rest of the spine by a subhorizontal step. The presence of several anteroposteriorly directed ridges gives the surface of this area of the neural spine a rough appearance. The process for the interspinous ligaments is located at the same level of the dorsal rim of the neural spine, and the lateral grooves are shallower than in the D4, as in *Viavenator* and CPP 893. In *Carnotaurus* this process is more ventrally located, whereas it is absent in *Eoabelisaurus*, *Majungasaurus*, and *Skorpiovenator*. In posterior view, only the surface of the neural spine can be seen; this has the same transverse thickness of the anterior portion, and it becomes wider towards its distal end.

In dorsal view (Figs. 5C and 5D), the neural spine is transversely thick and anteroposteriorly longer than that of the D4. The dorsal surface of the neural spine is slightly convex transversely and rectangular in outline, with the lateral rims diverging

slightly posteriorly. The posterior rim is concave, due to the presence of the base of two posteriorly directed processes.

*Sixth dorsal vertebra* (D6; Figs. 4A, 4B, 5A–5D; Table S1): The sixth dorsal vertebra has preserved part of the centrum and the neural arch. The centrum is as high as long and is slightly larger than D2-D5 vertebrae, as seen in *Carnotaurus* and *Majungasaurus*. The concavity of the anterior and posterior articular surfaces is even greater than in the previous vertebrae, and they show an oval outline. The lateral fossa of the centrum (Figs. 4A and 4B), such as D5, is shallow and lacks pneumatic foramina, as in *Majungasaurus*, but different from *Carnotaurus* and *Skorpiovenator*, which have lateral pleurocoels. Ventrally (Figs. 5A and 5B), despite the deformation, no groove or keel are observed as in *Eoabelisaurus* and *Skorpiovenator*, but unlike the D6 of *Carnotaurus* that has a pronounced keel.

The neural arch is badly damaged and crushed. In anterior view, the neural spine is transversely wider than the D5, and the anterior process for the interspinous ligaments reaches the dorsal table of the spine. In lateral view (Figs. 4A and 4B), the surface is eroded and only the parapophysis is distinguishable. It is partially broken and displaced anterodorsally. The neural spine is fully displaced anteriorly, being positioned almost entirely dorsally to the D5 centrum. It is anteroposteriorly long, exceeding half of the length of the vertebral centrum as in *Carnotaurus* and *Skorpiovenator*, but different from *Majungasaurus* where it is much smaller. The distal portion of the neural spine is transversely expanded with faint lateral ridges directed anteroposteriorly. The anterior process for the interspinous ligaments is partially broken; however, it is separated from the spine table.

In posterior view, only the right postzygapophysis can be distinguished, which is partially articulated with the next prezygapophysis. It seems to be longer anteroposteriorly than transversely wide, and the articular facet is directed ventrally, as in *Eoabelisaurus* and *Skorpiovenator*, but unlike *Carnotaurus* that has a ventromedially oriented prezygapophysis. In dorsal view (Figs. 5C and 5D), the neural spine is transversely wider and the lateral rims diverge more posteriorly than the D5. It shows a posterior concavity that probably separated two posteriorly directed processes.

*Seventh dorsal vertebra* (D7; Figs. 4A, 4B, 5C and 5D: Table S1): Only the right prezygapophysis and neural spine are preserved of this vertebra. The prezygapophysis is partially articulated to the preceding postzygapophysis (Figs. 4A and 4B). It is longer than wide, and the articular facet is slightly directed dorsolaterally, as in *Carnotaurus* and *Viavenator*, but different than the horizontal prezygapophysis present in *Majungasaurus*, or the dorsomedially oriented condition shown in *Dahalokely*. The neural spine shows the same size as the neural spine of the D6, and the anterior process for the interspinous ligaments is conspicuous (Figs. 4A and 4B). The distalmost portion of the neural spine is thick and holds several longitudinal crests. In dorsal view (Figs. 5C and 5D), the neural spine shows a triangular outline, and the right posterior process is visible.

*Posterior dorsal vertebrae* (Figs. 6 and 7; Table S1): Only some disarticulated elements corresponding to the posterior portion of the dorsal series are preserved. Despite their taphonomic deformation, some characteristics of the preserved centra and neural spines

indicate that these elements belong to the most distal dorsal vertebrae. One isolated centrum is spool-shaped (Figs. 6A–6F), with slightly concave and subcircular articular surfaces (Figs. 6A and 6B). The lateral surface has a shallow fossa, and there is a pleurocoel on each side (Figs. 6C and 6D). Dorsally, there are no signs of the neurocentral suture (Fig. 6E), thus the centrum was separated from the neural arch after their fusion. The ventral surface lacks either a groove or keel (Fig. 6F).

Another vertebra (Figs. 6G–6K), probably more distal than the centrum described above, preserves part of the centrum and neural arch. The anterior and posterior articular surfaces are concave with a slightly oval outline (Figs. 6G and 6H). In lateral view (Figs. 6I and 6J), there is a deep fossa, just below the neurocentral suture, without a pneumatic foramen, as in the posterior dorsals of *Dahalokely*, *Eoabelisaurus*, *Huinculsaurus*, *Ilokelesia*, *Majungasaurus*, *Niebla*, and *Skorpiovenator* but different than in *Carnotaurus*, *Viavenator*, and MPCN-PV-69, in which central fossae bear pleurocoels. The ventral surface lacks either a groove or a keel (Fig. 6K). The neural arch is crushed, and only the neural spine was preserved, which is anteroposteriorly shorter than the neural arch (Figs. 6I and 6J).

Several isolated dorsal neural spines were found (Figs. 7A–7F), preserving approximately their dorsal halves. All of them have a smaller anteroposterior extension than the one observed in the seventh neural spine. Reduced neural spines in the posterior portion of the dorsal series, especially in the last three ones, are also present in *Carnotaurus* and *Majungasaurus*. All recovered neural spines have the anterior processes for the interspinous ligaments (Figs. 7A–7C), which are separated from the dorsal table of the neural spines by two shallow lateral grooves. Theses processes reach dorsally the distal rim, as in *Dahalokely*, *Majungasaurus*, *Skorpiovenator*, and *Viavenator*. However, the posterior dorsals of *Carnotaurus* have a more ventrally placed process. All neural spines have a thickened distal end, with a marked lateral step and several lateral longitudinal ridges (Figs. 7D–7F). A similar condition is also present in *Carnotaurus* and *Viavenator*, whereas in *Dahalokely*, *Majungasaurus* and *Skorpiovenator* this dorsal swallowness is less developed, and absent in *Eoabelisaurus*. The dorsal surface is transversely and anteroposteriorly convex. In dorsal view (Figs. 7D–7F), the neural spines seem to have a Y-like outline, tapering anteriorly. In the posterior end, two lateral interspinous accessory processes are present (completely preserved only in one neural spine). These processes are finger-like shaped and posteriorly directed (Figs. 7B–7F). This structure was proposed as an autapomorphic condition for *Viavenator* (*Filippi et al., 2016*) and considered as an accessory interspinous articulation. This feature differs from the dorsal expansion of the neural spines present in other abelisauroids such as *Elaphrosaurus*, *Dahalokely*, and *Huinculsaurus*.

*Sacrum* (Fig. 8; Table S1): The sacrum is partially preserved and the vertebral centra suffered some degree of deformation. The entire right side was found fused with the right ilium, while the left side is fully exposed, except for the third vertebral centrum, which is fused and covered by the pubic peduncle of the ilium and part of the iliac peduncle of the pubis (Fig. 8A). The sacrum is composed of six vertebrae, as in *Eoabelisaurus*, *Carnotaurus*

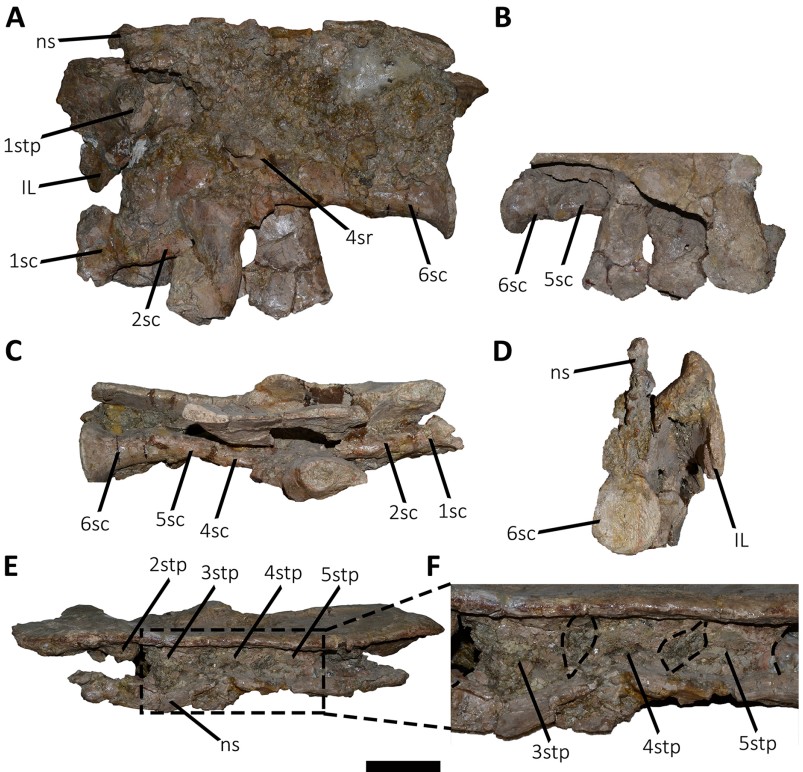

**Figure 8 Sacrum of *Aucasaurus garridoi* MCF-PVPH-236.** In lateral (A and B), ventral (C), posterior (D), and dorsal (E and F) views. Colored dashed lines marking the anterior and posterior rims of the third to fifth transverse processes. 1sc–sc, first to sixth sacral centra; 4sr, fourth sacral rib; 1stp–5stp, first to fifth sacral transverse processes; IL, ilion; ns, neural spine. Scale bar: 10 cm.

and *Masiakasaurus*, but different from the sacrum of *Majungasaurus*, and some tetanurans, which includes only five vertebrae. Although partially deformed, all six vertebral centra are fused forming an unique structure (Figs. 8A and 8B) as observed in *Ceratosaurus*, *Carnotaurus*, *Elaphrosaurus*, *Eoabelisaurus*, *Rahiolisaurus*, *Skorpiovenator*, and several Patagonian indeterminate abelisaurids (MAU-Pv-LI 547, MCF-PVPH-237, MMCh-PV 69, MPCN-PV-69), and possibly *Berberosaurus* and *Huinculsaurus*. Other abelisauroids, such as *Majungasaurus* (although adult individuals from that species are unknown), *Masiakasaurus*, *Rajasaurus*, and *Vespersaurus*, have a partially fused sacrum. Despite the deformation, the anterior surface of the first centrum is slightly concave and is dorsoventrally higher and mediolaterally wider than the remaining sacral centra. From the second to fifth sacral vertebra, the centra are transversally narrower and dorsoventrally lower than the first and sixth sacral vertebra, as observed in almost all ceratosaurs (*e.g.*, *Berberosaurus*, *Ceratosaurus*, *Elaphrosaurus*, *Carnotaurus*, *Skorpiovenator*), whereas in *Rahiolisaurus* this constriction is present from the third sacral centrum backwards; such a feature is apparently absent in *Majungasaurus*. *Aucasaurus* has apneumatic sacral centra, and the lateral walls are flat or slightly concave, as in other abelisauroids.

In lateral view (Fig. 8A), the sacrum is arched giving a concave outline to the ventral rim of the centra as in *Berberosaurus*, *Carnotaurus*, *Elaphrosaurus*, *Masiakasaurus*, *Skorpiovenator*, and MAU-Pv-LI 547, whereas in *Rahiolisaurus* this arching is less defined. Conversely, *Eoabelisaurus*, *Majungasaurus*, and *Rajasaurus* show a rather horizontal ventral margin. The lateral surfaces of the centra have shallow longitudinal fossae lacking pleurocoels, as in *Carnotaurus*, and *Majungasaurus*, and the indeterminate abelisaurids MAU-Pv-LI 547, MMCh-PV 69, and MPCN-PV-69. The neural arches are partially preserved and are fused to each other, creating a median axial wall. Unfortunately, the right side is fused to the ilium preventing us from getting additional morphological information, such as the presence or absence of fossae and laminae.

A fragment of the right rib of the first sacral vertebra was identified, and it is positioned just beneath the transverse process. This portion of the rib is dorsoventrally taller than anteroposteriorly long, different from the posterior sacral ribs, which are longer. Four left sacral ribs have be identified, being the fourth one the best preserved (the other three are poorly preserved). This rib is robust and holds a fossa on the ventral surface.

The neural spines of all sacral vertebrae are completely fused to one another forming a continuous shelf, as in *Skorpiovenator*, *Carnotaurus*, MAU-Pv-LI 547, and possibly *Majungasaurus*. *Eoabelisaurus* also possesses fused sacral neural spines, albeit it differs from more derived abelisaurids in that it lacks a dorsal shelf. Moreover, the sacral neural spines are transversely thin but with thicker distal ends. Several anteroposteriorly directed grooves and ridges stand out on the laterodorsal edge of the spines. In *Aucasaurus*, the fused neural spines are visible laterally above the dorsal edge of the ilium, as in *Eoabelisaurus*, *Majungasaurus*, *Carnotaurus*, and MAU-Pv-LI 547, but unlike *Elaphrosaurus* and *Skorpiovenator* where the sacrum is hidden by the ilia.

In ventral view (Fig. 8B), at least five of the sacral centra can be distinguished. In this view, the transverse constriction of the middle portion of the sacrum is clearly visible. The ventral surface of the vertebrae lack grooves or ridges, as seen in *Eoabelisaurus*, *Skorpiovenator*, and *Carnotaurus*.

In posterior view (Fig. 8D), the sixth sacral centrum has a posterior articular surface that is slightly concave and has an oval contour, being taller than wide. This vertebra has also the largest posterior surface when compared to the other sacral vertebrae.

In dorsal view (Figs. 8E and 8F), the transverse processes of the second through the fifth neural arches are fused to the ilium, two centimeters away from the dorsal rim, whereas the first transverse process contact the medial wall more ventrally. Moreover, the second up to the fifth sacral vertebra have transverse processes nearly horizontally directed. Conversely, the transverse process of the sixth sacral is dorsally inclined, due to the ventral position of this vertebra with respect the anterior ones. The transverse processes of the third through the fifth sacral vertebrae are anteroposteriorly longer than the other sacral transverse processes (Fig. 8F). In addition to be fused with the ilium, the transverse processes are fused each other at their distalmost ends, leaving a medial passage (Fig. 8F), as in *Masiakasaurus* and *Skorpiovenator*. The dorsal part of the neural spines form a continuous co-ossified table and among them are visible two anterior and posterior interspinous processes that contact each other, as in *Carnotaurus*, *Skorpiovenator*, and MAU-Pv-LI 547.

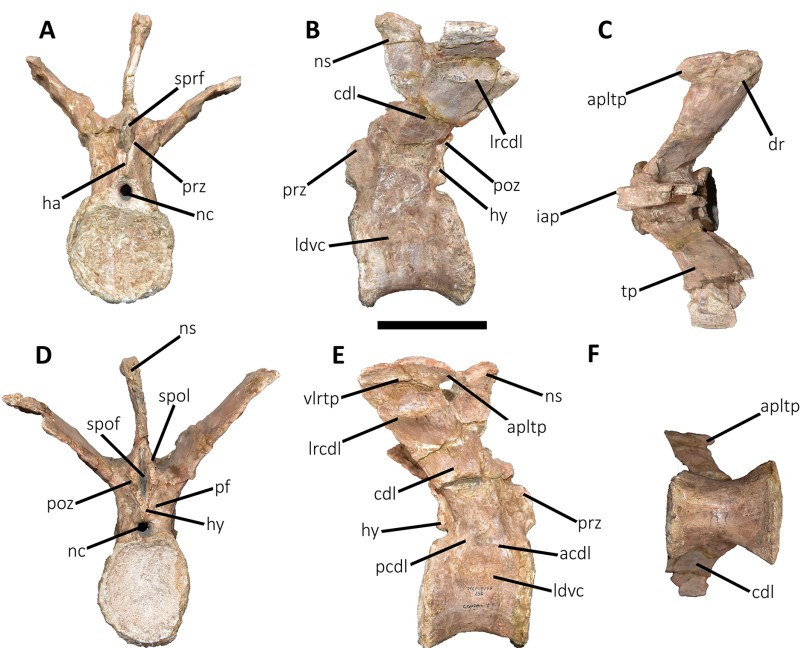

**Figure 9 First caudal vertebra of *Aucasaurus garridoi* MCF-PVPH-236.** In anterior (A), lateral (B and E), dorsal (C), posterior (D), and ventral (F) views. acdl, anterior centrodiapophyseal lamina; apltp, anterior process of lateral transverse process; cdl, centrodiapophyseal lamina; dr, dorsal roughness; ha, hypantrum; hy, hyposphene; iap, interspinous accessory process; ldvc, lateral depression of vertebral centrum; lrcdl, lateral ridge of centrodiapophyseal lamina; nc, neural canal; ns, neural spine; pcdl, posterior centrodiapophyseal lamina; pf, pneumatic foramen; poz, postzygapophysis; prz, prezygapophysis; spof, spinopostzigapophyseal fossa; spol, spinopostzigapophyseal lamina; sprf, spinoprezigapophyseal fossa; tp, transverse process; vlrtp, ventrolateral ridge of the transverse process. Scale bar: 10 cm.

*Caudal vertebrae* (Figs. 9–21; Table S1): MCF-PVPH-236 includes the articulated first to thirteenth anterior vertebrae (with their corresponding haemal arches), two posterior caudal vertebrae, and several isolated remains such as fragmentary neural spines and transverse processes. In general, there is a reduction in the general size of the centrum towards the posterior region, a transverse narrowing of the neural arch in the area of the pedicels in the distal anterior elements (between the seventh and tenth vertebrae), and a posterior displacement of the neural spine towards the rear of the tail. The transverse processes are transversely wide, with a ratio higher than 1.3 with respect to the length of the centrum. Sutures between neural arches and vertebral centra are completely obliterated in all caudal vertebrae.

*First caudal vertebra* (Fig. 9; Table S1): The first caudal vertebra is well-preserved. The centrum has a concave anterior surface and an oval outline with its major axis dorsoventrally directed (Fig. 9A), as in *Eoabelisaurus* and *Skorpiovenator*, but different from *Carnotaurus* in which the articular surface has a circular outline. In lateral view (Figs. 9B and 9E), a pleurocoel is absent and instead, there is an extensive anteroposterior depression just beneath the neurocentral suture, as in *Carnotaurus*. In *Skorpiovenator*, this depression is shallow, whereas it is absent in all caudal vertebrae in *Eoabelisaurus* and MPM 99. In this view, the centrum has a parallelogram outline, since the anterior margin is

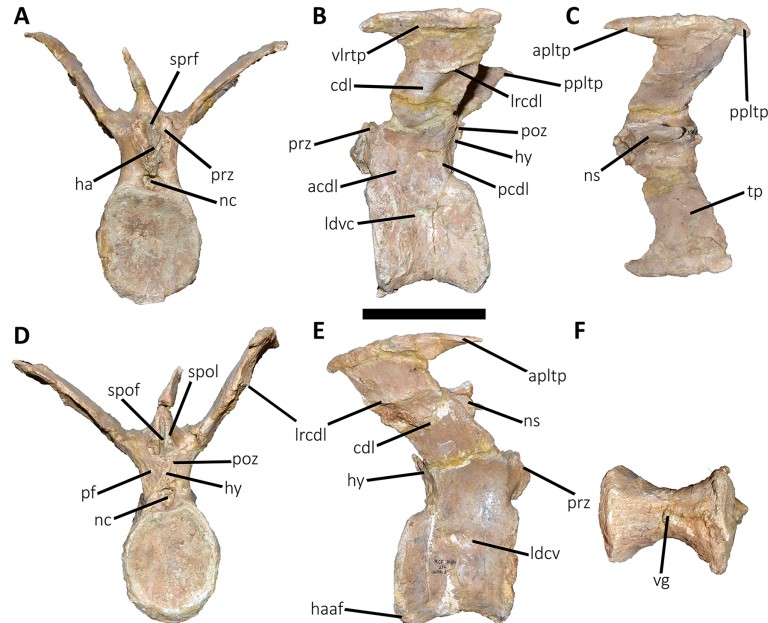

**Figure 10 Second caudal vertebra of *Aucasaurus garridoi* MCF-PVPH-236.** In anterior (A), lateral (B and E), dorsal (C), posterior (D), and ventral (F) views. acdl, anterior centrodiapophyseal lamina; apltp, anterior process of lateral transverse process; cdl, centrodiapophyseal lamina; ha, hypantrum; haaf, haemal arch articular facet; hy, hyposphene; ldvc, lateral depression of vertebral centrum; lrcdl, lateral ridge of centrodiapophyseal lamina; nc, neural canal; ns, neural spine; pcdl, posterior centrodiapophyseal lamina; pf, pneumatic foramen; poz, postzygapophysis; ppltp, posterior process of lateral transverse process; prz, prezygapophysis; spof, spinopostzigapophyseal fossa; spol, spinopostzigapophyseal lamina; sprf, spinoprezigapophyseal fossa; tp, transverse process; vg, ventral groove; vlrtp, ventrolateral ridge of the transverse process. Scale bar: 10 cm.

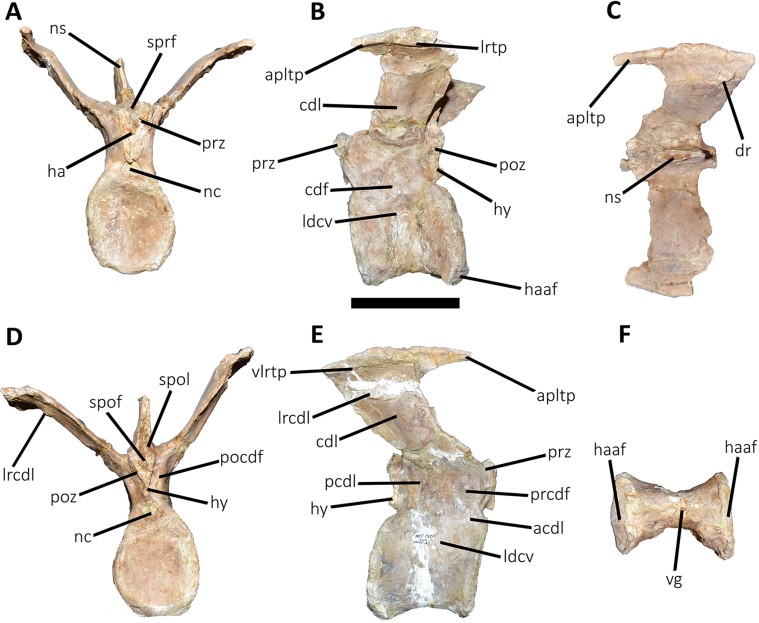

**Figure 11 Third caudal vertebra of *Aucasaurus garridoi* MCF-PVPH-236.** In anterior (A), lateral (B and E), dorsal (C), posterior (D), and ventral (F) views. acdl, anterior centrodiapophyseal lamina; apltp, anterior process of lateral transverse process; cdf, centrodiapophyseal fossa; cdl, centrodiapophyseal lamina; dr, dorsal roughness; ha, hypantrum; haaf, haemal arch articular facet; hy,

**Figure 11 (continued)**
hyposphene; ldvc, lateral depression of vertebral centrum; lrcdl, lateral ridge of centrodiapophyseal lamina; lrtp, lateral rugosity of transverse process; nc, neural canal; ns, neural spine; pcdl, posterior centrodiapophyseal lamina; pocdf, postzygapophyseal centrodiapophyseal fossa; poz, post-zygapophysis; prcdf, prezygapophyseal centrodiapophyseal fossa; prz, prezygapophysis; spof, spinopostzigapophyseal fossa; spol, spinopostzigapophyseal lamina; sprf, spinoprezigapophyseal fossa; vg, ventral groove; vlrtp, ventrolateral ridge of the transverse process. Scale bar: 10 cm.

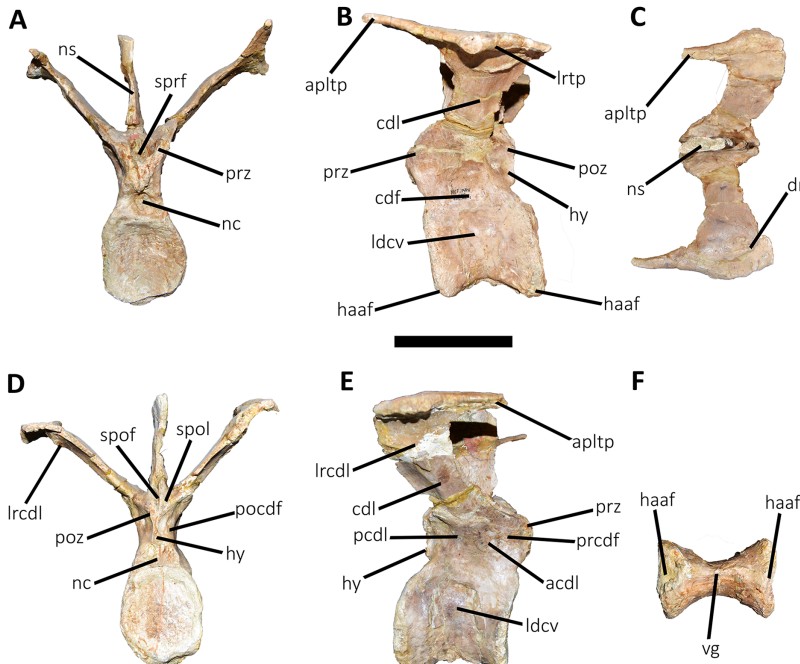

**Figure 12 Fourth caudal vertebra of *Aucasaurus garridoi* MCF-PVPH-236.** In anterior (A), lateral (B and E), dorsal (C), posterior (D), and ventral (F) views. acdl, anterior centrodiapophyseal lamina; apltp, anterior process of lateral transverse process; cdf, centrodiapophyseal fossa; cdl, centrodiapophyseal lamina; dr, dorsal roughness; haaf, haemal arch articular facet; hy, hyposphene; ldvc, lateral depression of vertebral centrum; lrcdl, lateral ridge of centrodiapophyseal lamina; lrtp, lateral rugosity of transverse process; nc, neural canal; ns, neural spine; pcdl, posterior centrodiapophyseal lamina; pocdf, postzygapophyseal centrodiapophyseal fossa; poz, postzygapophysis; prcdf, prezygapophyseal centrodiapophyseal fossa; prz, prezygapophysis; spof, spinopostzigapophyseal fossa; spol, spinopostzigapophyseal lamina; sprf, spinoprezigapophyseal fossa; vg, ventral groove. Scale bar: 10 cm.

slightly concave and the posterior margin slightly convex, as in several abelisaurids (*Méndez, 2014b*). The posterior surface is also concave and elliptical with the greater axis dorsoventrally directed (Fig. 9D), as in *Skorpiovenator*, but unlike *Kurupi* and *Carnotaurus* in which the surface is transversely wider than dorsoventrally high. The ventral end of the posterior surface bears the articular facet for the first haemal arch. In ventral view (Fig. 9F), the surface has a shallow depression, different from the flat surface observed in *Eoabelisaurus*, *Kurupi*, *Skorpiovenator*, and *Carnotaurus*, or the grooved surface present in *Dilophosaurus*, *Ceratosaurus*, and *Majungasaurus*.

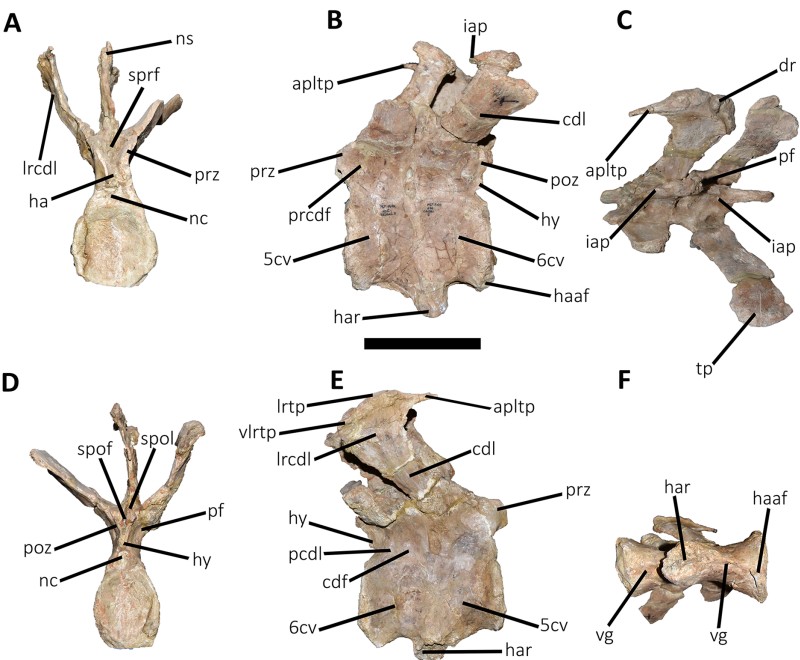

**Figure 13 Fifth and sixth caudal vertebrae of *Aucasaurus garridoi* MCF-PVPH-236.** In anterior (A), lateral (B and E), dorsal (C), posterior (D), and ventral (F) views. 5 cv, fifth caudal vertebra; 6 cv, sixth caudal vertebra; apltp, anterior process of lateral transverse process; cdl, centrodiapophyseal lamina; dr, dorsal roughness; ha, hypantrum; har, haemal arch; haaf, haemal arch articular facet; hy, hyposphene; iap; interspinous accessory process; lrcdl, lateral ridge of centrodiapophyseal lamina; lrtp, lateral rugosity of transverse process; nc, neural canal; ns, neural spine; pf, pneumatic foramen; poz, postzygapophysis; prcdf, prezygapophyseal centrodiapophyseal fossa; prz, prezygapophysis; spof, spinopostzigapophyseal fossa; spol, spinopostzigapophyseal lamina; sprf, spinoprezygapophyseal fossa; tp, transverse process; vg, ventral groove; vlrtp, ventrolateral ridge of the transverse process. Scale bar: 10 cm.

In anterior view (Fig. 9A), the neural canal shows an elliptical outline, different from the circular shape seen in *Carnotaurus*. The hypantrum is transversely reduced and the prezygapophyses are close to each other, as in *Eoabelisaurus* and *Carnotaurus*. It is likely that the articulation between the last sacral vertebra and the first caudal vertebra allowed limited lateral movements. The prezygapophysis (the right one is partially broken) has a nearly vertical orientation, as in *Eoabelisaurus* and *Carnotaurus*. The prezygodiapophyseal (prdl) and spinoprezygapophyseal laminae are lost due to weathering.

The spinoprezygapophyseal fossa (sprf) is deep but transversely narrow, different from the shallower fossa present in *Eoabelisaurus* or the wider fossa observed in *Kurupi*. A septum divides the sprf in two areas. Laterally to the prezygapophysis, the prezygapophyseal centrodiapophyseal fossa (prcdf) is a shallow depressions. This fossa is also present in *Carnotaurus* but forming a shallow concavity, whereas in *Eoabelisaurus* the surface is flat without depression. In this view, the transverse process has a strong laterodorsal inclination, at an angle of approximately 48°, as in *Eoabelisaurus* and *Carnotaurus* whereas in *Kurupi* and *Skorpiovenator* the transverse process shows an inclination less than 30°. The neural spine is transversely thin; it widens distally forming a terminal bulge, as in

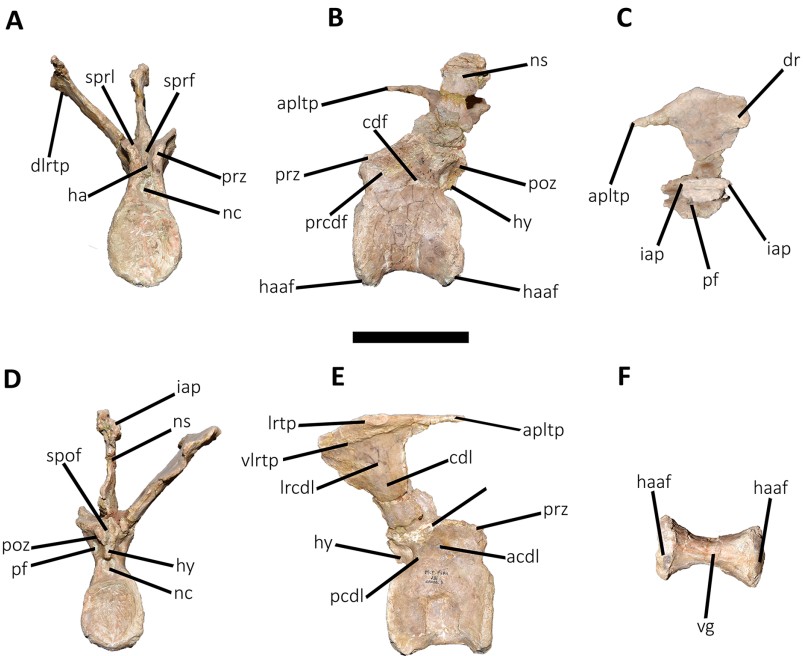

**Figure 14 Seventh caudal vertebra of *Aucasaurus garridoi* MCF-PVPH-236.** In anterior (A), lateral (B and E), dorsal (C), posterior (D), and ventral (F) views. acdl, anterior centrodiapophyseal lamina; apltp, anterior process of lateral transverse process; cdf, centrodiapophyseal fossa; cdl, centrodiapophyseal lamina; dr, dorsal roughness; ha, hypantrum; haaf, haemal arch articular facet; hy, hyposphene; iap; interspinous accessory process; lrcdl, lateral ridge of centrodiapophyseal lamina; lrtp, lateral rugosity of transverse process; nc, neural canal; ns, neural spine; pcdl, posterior centrodiapophyseal lamina; pf, pneumatic foramen; poz, postzygapophysis; prcdf, prezygapophyseal centrodiapophyseal fossa; prz, prezygapophysis; spof, spinopostzygapophyseal fossa; sprf, spinoprezygapophyseal fossa; sprl, spinopre-zigapophyseal lamina; vg, ventral groove. Scale bar: 10 cm.

*Eoabelisaurus* and *Carnotaurus*. This terminal bulge appears absent in the caudal vertebrae of *Ceratosaurus*.

In lateral view (Figs. 9B and 9E), the prezygapophysis and postzygapophysis do not exceed the anterior and posterior rims of the centrum, respectively, as in *Skorpiovenator* and *Carnotaurus* but unlike *Dilophosaurus*, *Ceratosaurus*, and *Eoabelisaurus* where they are projected beyond the rims of the centrum. Ventrally, the transverse process exhibits a centrodiapophyseal lamina (cdl) that splits ventrally in the acdl and pcdl that are poorly developed, as in *Kurupi*. In *Aucasaurus* and other abelisaurids, such as *Skorpiovenator* and *Carnotaurus*, the first and the remaining caudal vertebrae lack pneumaticity ventral to these laminae. The cdl ends laterally with a well-marked ridge, as in *Skorpiovenator* and *Carnotaurus*, which is absent in *Eoabelisaurus*. A depression separates this crest from another accessory ridge that is also directed anteroposteriorly, as in *Carnotaurus*. The neural spine, in lateral view, it is almost perpendicular to the centrum and shows a rectangular outline with the dorsal rim directed anterodorsally/posteroventrally. In contrast, in *Carnotaurus* and *Eoabelisaurus* the neural spine is inclined posteriorly, projecting beyond the posterior surface of the centrum. At the dorsalmost portion of this vertebra, the neural spine presents anteroposteriorly directed ridges and furrows for

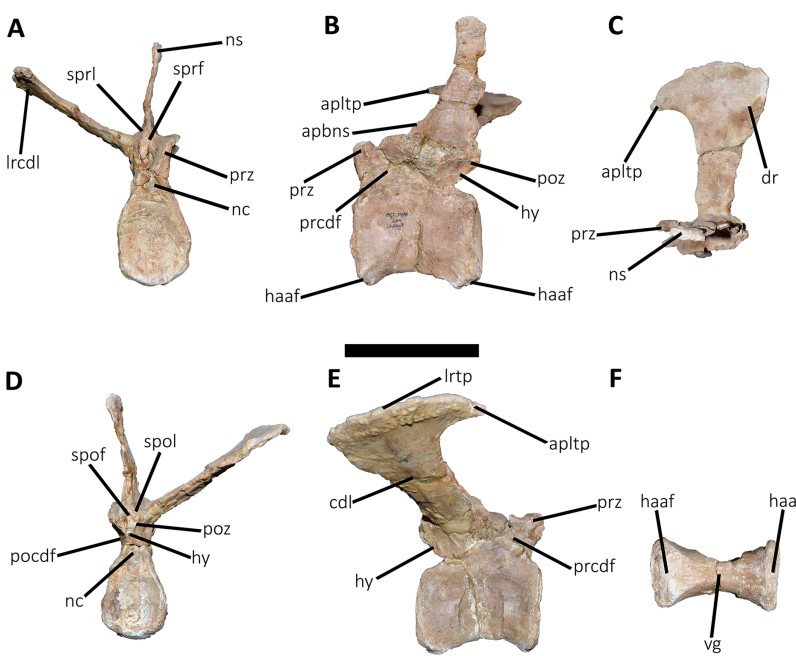

**Figure 15 Eighth caudal vertebra of *Aucasaurus garridoi* MCF-PVPH-236.** In anterior (A), lateral (B and E), dorsal (C), posterior (D), and ventral (F) views. apbns, anterior process of basal neural spine; apltp, anterior process of lateral transverse process; cdl, centrodiapophyseal lamina; dr, dorsal roughness; haaf, haemal arch articular facet; hy, hyposphene; lrcdl, lateral ridge of centrodiapophyseal lamina; lrtp, lateral rugosity of transverse process; nc, neural canal; ns, neural spine; pocdf, postzygapophyseal centrodiapophyseal fossa; poz, postzygapophysis; prcdf, prezygapophyseal centrodiapophyseal fossa; prz, prezygapophysis; spof, spinopostzigapophyseal fossa; spol, spinopostzigapophyseal lamina; sprf, spino-prezigapophyseal fossa; sprl, spinoprezigapophyseal lamina; vg, ventral groove. Scale bar: 10 cm.

ligamental anchorage. The neural spine is the half of the anteroposterior length of the neural arch at its base, different from *Ceratosaurus*, *Carnotaurus* and *Eoabelisaurus* where it is longest.

In dorsal view (Fig. 9C), the transverse process is posteriorly inclined with respect to the neural spine, surpassing the posterior surface of the centrum, as in *Eoabelisaurus*, *Kurupi*, *Skorpiovenator*, and *Carnotaurus*. Although partially broken, the transverse processes hold, at the lateral edge, the anterior awl-like processes as in *Carnotaurus*. This process is totally absent in all the caudal vertebrae of *Eoabelisaurus* and *Majungasaurus*. In the posterodorsal portion of the transverse process, there is a V-shaped rugosity, also present in *Carnotaurus* albeit much weaker. Between this scar and the lateral border of the transverse process, the dorsal surface is slightly concave. The anterior rim of the transverse process is concave, whereas the posterior one is almost straight, as in *Carnotaurus* and *Skorpiovenator* but unlike *Eoabelisaurus* where both rims are straight. In the middle of the anterodorsal surface of the transverse process, a possibly ligamentous scar is present, different from the prominent spur observed in *Kurupi*. This trait is here considered autapomorphic for *Aucasaurus garridoi* (see Discussion). There are two anteriorly directed, dorsal processes of the neural spine absent in *Eoabelisaurus* and *Carnotaurus*.

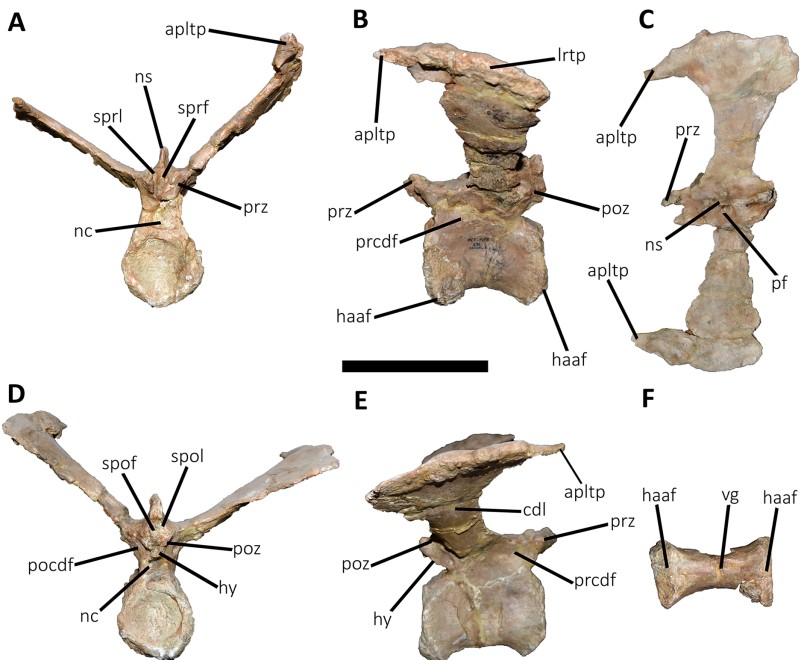

**Figure 16 Ninth caudal vertebra of *Aucasaurus garridoi* MCF-PVPH-236.** In anterior (A), lateral (B and E), dorsal (C), posterior (D), and ventral (F) views. apltp, anterior process of lateral transverse process; cdl, centrodiapophyseal lamina; haaf, haemal arch articular facet; hy, hyposphene; lrtp, lateral rugosity of transverse process; nc, neural canal; ns, neural spine; pf, pneumatic foramen; pocdf, post-zygapophyseal centrodiapophyseal fossa; poz, postzygapophysis; prcdf, prezygapophyseal centrodiapophyseal fossa; prz, prezygapophysis; spof, spinopostzigapophyseal fossa; spol, spinopostzigapophyseal lamina; sprf, spinoprezigapophyseal fossa; sprl, spinoprezigapophyseal lamina; vg, ventral groove. Scale bar: 10 cm.

In posterior view (Fig. 9D), the neural canal is wider dorsally than ventrally. There is a small depression at the entry of the neural canal. The hyposphene is prominent and formed by the union of the intrapostzygapophyseal laminae that arise ventrally to the postzygapophyses, as in several ceratosaurs (*e.g.*, *Ceratosaurus*, *Carnotaurus*, *Kurupi*). Laterally to the hyposphene, the postzygapophyseal centrodiapophyseal fossa (pocdf) is shallow and hold a pneumatic foramen (see Discussion). This fossa is also shallow in all the anterior caudal vertebrae of *Carnotaurus*, *Eoabelisaurus*, *Skorpiovenator*, and *Viavenator*, although they lack pneumatic foramina. Unlike *Carnotaurus*, *Aucasaurus* lacks centropostzygapophyseal lamina (cpol) that delimit ventrally the pocdf.
The postzygapophyses are partially preserved, and the articular surfaces are directed ventrolaterally, as in *Ceratosaurus*, *Carnotaurus*, and *Skorpiovenator*, whereas in *Dilophosaurus* they are directed ventromedially. Laterally to the postzygapophysis, the podl is low. Dorsal to the postzygapophyses, the spinopostzygapophyseal laminae (spol) are robust and join dorsally on the posterior surface of the neural spine. Between these last two laminae and the postzygapophyses the spinopostzygapophyseal fossa (spof) is transversely narrow, as in *Carnotaurus*, whereas in *Skorpiovenator* this fossa is wider.

*Second caudal vertebra* (Fig. 10; Table S1): The second vertebra is almost completely preserved, lacking only the anterior ends of the prezygapophyses and the distal half of the

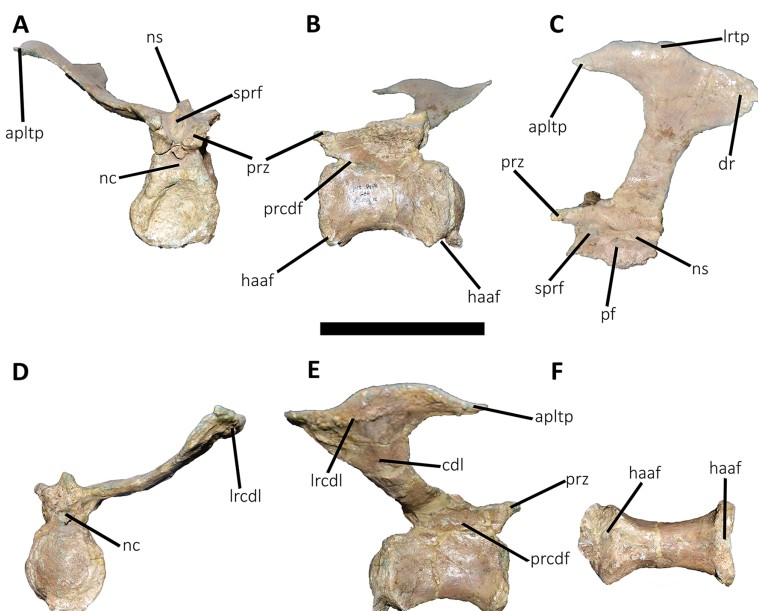

**Figure 17 Tenth caudal vertebra of *Aucasaurus garridoi* MCF-PVPH-236.** In anterior (A), lateral (B and E), dorsal (C), posterior (D), and ventral (F) views. apltp, anterior process of lateral transverse process; cdl, centrodiapophyseal lamina; dr, dorsal roughness; haaf, haemal arch articular facet; lrcdl, lateral ridge of centrodiapophyseal lamina; lrtp, lateral rugosity of transverse process; nc, neural canal; ns, neural spine; pf, pneumatic foramen; prcdf, prezygapophyseal centrodiapophyseal fossa; prz, pre-zygapophysis; sprf, spinoprezigapophyseal fossa. Scale bar: 10 cm.

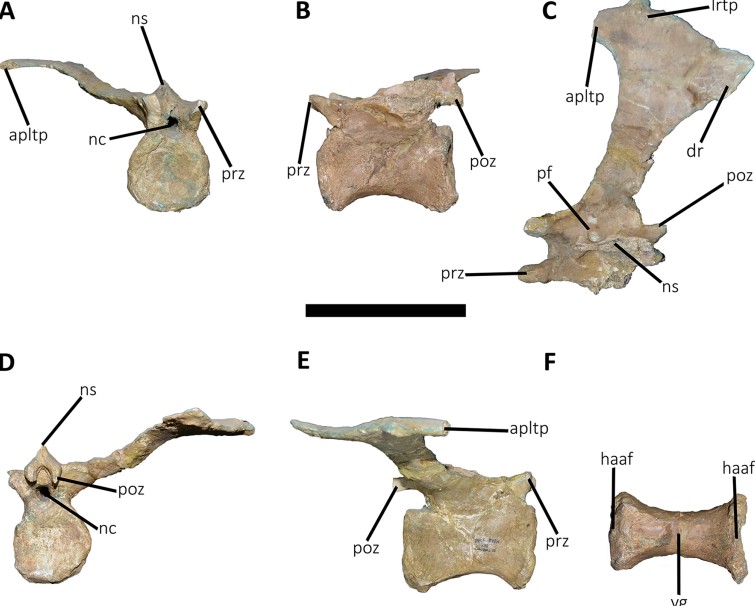

**Figure 18 Eleventh caudal vertebra of *Aucasaurus garridoi* MCF-PVPH-236.** In anterior (A), lateral (B and E), dorsal (C), posterior (D), and ventral (F) views. apltp, anterior process of lateral transverse process; dr, dorsal roughness; haaf, haemal arch articular facet; lrtp, lateral rugosity of transverse process; nc, neural canal; ns, neural spine; pf, pneumatic foramen; poz, postzygapophysis; prz, prezygapophysis; vg, ventral groove. Scale bar: 10 cm.

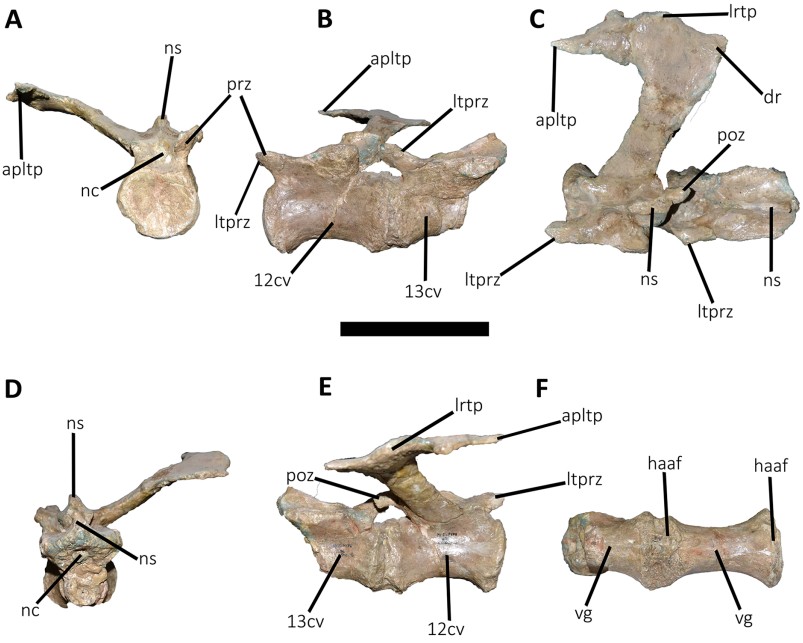

**Figure 19 Twelfth and thirteenth caudal vertebrae of *Aucasaurus garridoi* MCF-PVPH-236.** In anterior (A), lateral (B and E), dorsal (C), posterior (D), and ventral (F) views. 12 cv, twelfth posterior vertebra; 13 cv, thirteenth posterior vertebra; apltp, anterior process of lateral transverse process; dr, dorsal roughness; haaf, haemal arch articular facet; ltprz, lateral tubercle of prezygapophysis; lrtp, lateral rugosity of transverse process; nc, neural canal; ns, neural spine; poz, postzygapophysis; prz, prezygapophysis; vg, ventral groove. Scale bar: 10 cm.

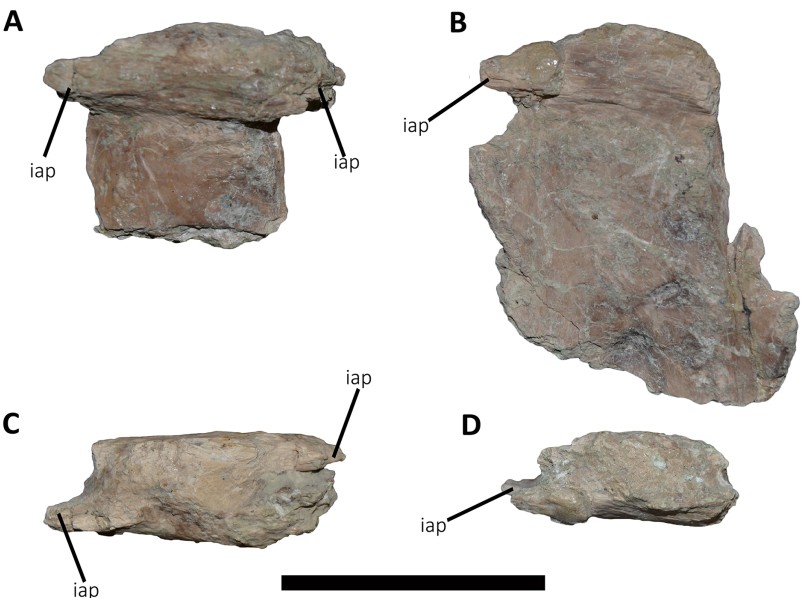

**Figure 20 Caudal neural spines of *Aucasaurus garridoi* MCF-PVPH-236.** In lateral (A and B) and dorsal (C and D) views. iap, interspinous accessory process. Scale bar: 5 cm.

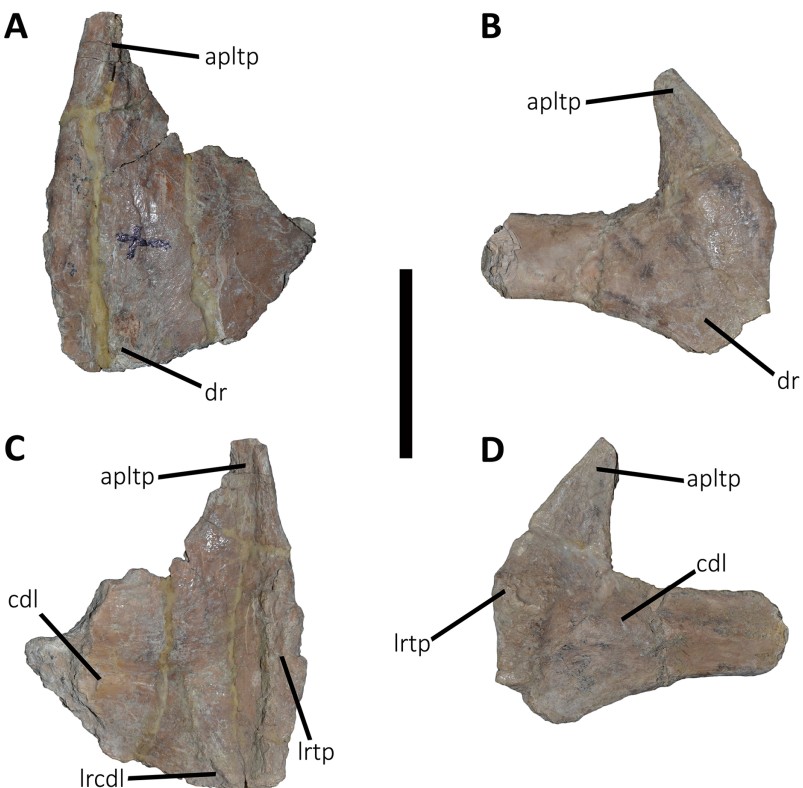

**Figure 21 Caudal transverse processes of *Aucasaurus garridoi* MCF-PVPH-236.** In dorsal (A and B) and ventral (C and D) views. Abbreviations: apltp, anterior process of lateral transverse process; cdl, centrodiapophyseal lamina; dr, dorsal roughness; lrcdl, lateral ridge of centrodiapophyseal lamina; lrtp, lateral rugosity of transverse process. Scale bar: 5 cm.   

neural spine. The centrum has an elliptical anterior articular surface being taller than wide (Fig. 10A), as in *Eoabelisaurus* and *Skorpiovenator* but different from *Carnotaurus* where it is wider than tall. Ventrally to the anterior articular surface, a low rim represents the contact area for the haemal arch. As in the first caudal vertebra, the lateral surface lacks pleurocoels (Fig. 10B), although there is a depression below the neurocentral suture. Conversely, the second caudal vertebra of *Carnotaurus* and *Skorpiovenator* lack such depression on the lateral surface of the centrum. As in the first caudal vertebra, in lateral view the centrum has a parallelogram-shaped outline. The posterior articular surface is smaller than the anterior one (Fig. 10D), although it has the same oval outline, unlike *Carnotaurus* that has an almost circular outline. The posterior contact surface for the haemal arch is more extensive with respect to the anterior facet. The ventral surface has a longitudinal groove that extends along the entire surface (Fig. 10F), and is laterally bounded by two low ridges. While, in *Carnotaurus* the ventral surface is smooth without groove or ridges.

In anterior view (Fig. 10A), the neural canal has a circular outline. The prezygapophyses are almost completely lost, thus the shape cannot be observed. Although, they possibly were oriented medially with an inclination of 60° from the horizontal plane, as *Eoabelisaurus* and *Carnotaurus*. The hypantrum is partially preserved, with an almost

complete right wall. This structure is wider than in the previous vertebra. In *Aucasaurus*, laterally to the prezygapophysis there are neither foramina nor concavities, as in *Skorpiovenator*. Despite the sprl are partially broken they seem low, delimiting a dorsoventrally deep sprf. There is a median septum in the bottom of the sprf. The transverse process continue to show a pronounced dorsal inclination (although the right one is more dorsally inclined due to the diagenetic deformation), as in *Eoabelisaurus* and *Carnotaurus*. In contrast, in *Skorpiovenator* the transverse process is approximately horizontal. In *Aucasaurus* the neural spine is partially preserved and is transversely thin.

In lateral view (Figs. 10B and 10E), the lateral rims of the transverse process has a pronounced roughness. Ventral to the transverse process there is a well-developed cdl that occupies the entire surface, as *Carnotaurus*. This condition differs from *Skorpiovenator* where the cdl is mainly developed in the anteroventral portion of the transverse process, forming a shallow depression in the posterior portion. Moreover, this lamina is laterally bounded by an anteroposteriorly directed ridge (as in the first caudal vertebra) as in *Carnotaurus*. As observed in the first caudal vertebra, there is another accessory lateral ridge located almost in the lateral edge of the transverse process. Ventral to the transverse process there are no pneumatic foramina or fossae, holding only a shallow concavity that separates the acdl from the pcdl, as in *Carnotaurus* and *Skorpiovenator*, while in *Eoabelisaurus* these two laminae are poorly developed. The transverse process present a considerable posterior inclination, since it projects beyond the centrum, as in *Skorpiovenator* and *Carnotaurus*. Only the base of the neural spine is preserved, making it impossible to observe the morphology of the dorsal region.

In dorsal view (Fig. 10C), the lateral rim of the transverse process have the typical awl-shaped anterior process. Moreover, in this view the lateral rim is slightly convex and is visible the lateral roughness. A small process is also present in the posterolateral end of the transverse process, although it does not have the same development as the same process present in some abelisaurids, such as *Ekrixinatosaurus*, *Ilokelesia*, and *Skorpiovenator*. This reduced posterior process is absent in *Carnotaurus*. On the posterolateral end the V-shaped scar is conspicuous, whereas in the second caudal vertebra of *Carnotaurus* it is less-marked. The longitudinal scar on the middle of the transverse process is less pronounced than the previous vertebra. The anterior and posterior rims of the transverse process have a slightly sigmoid outline. The preserved portion of the neural spine is transversely narrow with a leaf like contour in cross-section, being the posterior portion wider than the anterior one. In *Aucasaurus*, the transverse process is less posteriorly inclined than *Carnotaurus*.

In posterior view (Fig. 10D), the neural canal has a triangular outline and is dorsovetrally taller than the first caudal vertebra. The hyposphene is lost, but it was conspicuous. As in the first caudal vertebra, the pocdf is shallow and has a pneumatic foramen, which is absent in *Eoabelisaurus* and *Carnotaurus*. The postzygapophyses are partially broken, with the articular facets ventrolaterally oriented. The spol delimit a rectangular spof that is transversely narrower and anteroposteriorly shallower than the previous vertebra, unlike *Carnotaurus* where this fossa remains deep and wide.

*Third caudal vertebra* (Fig. 11; Table S1): The third caudal vertebra was almost completely preserved, lacking only the anterior ends of the prezygapophyses, part of the neural spine, and the anterior and posterior ends of the lateral border of the left transverse process. The anterior articular surface of the centrum is elliptical in outline with its long axis oriented dorsoventrally (Fig. 11A), as in *Eoabelisaurus* and *Skorpiovenator*. This morphology differs from *Carnotaurus* which has a circular contour. In lateral view (Fig. 11B), the neurocentral suture is obliterated. The centrum has the depression just below the neurocentral suture, which is absent in *Carnotaurus*. The anterior and posterior margins of the centrum are slightly concave and convex, respectively, giving to it a parallelogram-shaped outline, as in *Eoabelisaurus* and *Carnotaurus*. The posterior articular surface is elliptical in outline with its long axis oriented dorsoventrally (Fig. 11D), as in *Carnotaurus*. On the posteroventral end, the contact surface for the haemal arch is wide and has an inclination of 40°. In ventral view (Fig. 11F), the centrum holds a longitudinal groove, which is absent in *Carnotaurus*, *Eoabelisaurus*, and *Skorpiovenator*.

In anterior view (Fig. 11A), the neural arch is narrower transversely than the previous vertebra. The entry of the neural canal has a circular outline. Despite the hypantrum is almost completely lost, it can be inferred that it was dorsoventrally high, as in *Carnotaurus* but unlike *Eoabelisaurus* where the hypantrum is low. Only the left prezygapophysis is partially preserved, showing a dorsomedial inclination of the articular surface higher than 60°, different from *Eoabelisaurus* and *Carnotaurus* that have a lower inclination. The sprl are completely weathered, except for a portion at the base of the neural spine, thus we cannot estimated the depth and width of the sprf. However, this fossa lacks of the middle septum observed in *Carnotaurus*. The transverse process has a dorsal inclination higher than 45°, as in *Carnotaurus* but different from *Eoabelisaurus* and *Skorpiovenator* where it shows a lower inclination. The neural spine preserves only its basal third. The preserved portion of neural spine is transversely thin, as in *Eoabelisaurus*, *Skorpiovenator*, and *Carnotaurus*, and shows a leaf-shaped contour in cross-section.

In lateral view (Figs. 11B and 11E), the lateral edge of the transverse process is markedly roughened. The cdl ends laterally with an anteroposteriorly directed crest, and laterally to this crest a shallow depression is present. Ventral to the cdl, the cdf separates a well-developed acdl from the pcdl, as in *Carnotaurus*, whereas in *Eoabelisaurus* both laminae are reduced. Dorsal to the anterior pedicels, the prcdf is deep but without pneumatic foramina. In *Aucasaurus*, the transverse process has a significant posterior inclination surpassing the posterior articular surface of the centrum, as in *Skorpiovenator* and *Carnotaurus* but unlike *Eoabelisaurus* where the transverse process is laterally directed. Although incomplete, the neural spine does not exhibit the posterior orientation observed in *Carnotaurus*.

In dorsal view (Fig. 11C), the transverse processes exhibit the anteriorly directed awl-shaped processes, although the left one is almost lost. On the posterolateral corner, the transverse process lacks the posterior process present in the second caudal vertebra. The right transverse process shows a marked posterolateral rugosity, whereas the middle scar is poorly developed. The anterior and posterior rims are sinusoidal, as in

*Skorpiovenator*. In this view, the neural spine is leaf-shaped in cross-section with the widest part located anteriorly.

In posterior view (Fig. 11D), the neural canal entry is dorsoventrally higher than transversely wide. The hyposphene, although partially broken, is more conspicuous than in the previous caudal vertebrae. Lateral to the hyposphene, the pocdf is shallow and have pneumatic foramina. The postzygapophyses are partially preserved, and have a lateroventral orientation, as in *Skorpiovenator* and *Eoabelisaurus*, contrasting with the almost horizontal orientation in *Carnotaurus*. The spof is narrower than the previous vertebrae. The neural spine is wide at the base, thinning towards the distal portion.

*Fourth caudal vertebra* (Fig. 12; Table S1): The fourth caudal vertebra only lost the distal end of the neural spine. The anterior articular surface of the centrum is elliptical in outline being taller than wide (Fig. 12A), as in *Eoabelisaurus*, *Skorpiovenator*, and *Carnotaurus*. Laterally (Fig. 12B), the surface shows a deep depression below the neurocentral suture without pneumatic foramina. The anterior and posterior rims of the lateral surface remain concave/convex and slightly tilted anteriorly, as in *Eoabelisaurus* and *Carnotaurus*, while *Skorpiovenator* has a more rectangular outline. The posterior articular surface shows a less pronounced concavity with respect to the anterior one, and its contour is elliptical, being taller than wide (Fig. 12D), as in *Skorpiovenator* and *Carnotaurus*. The posteroventral surface for articulation of the haemal arch is wide. Despite the ventral surface of the centrum is partially collapsed, the longitudinal groove is present (Fig. 12F).

In anterior view (Fig. 12A), the neural canal has a dorsoventral elliptical outline, different from the circular shape seen in *Carnotaurus*. We cannot estimate the size and shape of the hypantrum, since its lateral walls were lost. The prezygapophyses are partially preserved and have a medial inclination greater than 60°, as *Skorpiovenator* but unlike *Eoabelisaurus* and *Carnotaurus* where the prezygapophysis is less inclined. The sprf has transverse narrower than the two previous vertebrae, whereas sprl are not preserved. The transverse process has a dorsal inclination greater than 45°, as in *Carnotaurus* and unlike *Eoabelisaurus* and *Skorpiovenator* that have a less inclined transverse process. The neural spine is partially preserved, probably the first two thirds, narrowing towards the distal portion.

In lateral view (Figs. 12B and 12E), the lateral rim of the transverse process is thick, showing a marked roughness with the presence of several tubercles. This rugosity and thickening of the lateral border of the transverse process is absent in *Carnotaurus* and *Skorpiovenator*, whereas it is more weakly developed in the anterior caudal vertebrae of *Viavenator*. Lateral to the cdl and the longitudinal ridge, the surface has a conspicuous accessory ridge and is strongly concave due to a ventral bowing of the lateral end. The fourth caudal vertebra of *Carnotaurus* has the accessory ridge but lacks the ventral bowing. The cdf is deep, as in *Skorpiovenator*, whereas *Eoabelisaurus* has a shallow cdf and low acdl and pcdl. The prcdf is deeper than the second and third caudal vertebrae, as in *Eoabelisaurus* and *Skorpiovenator*. In this view, the transverse process is poorly posteriorly directed, as in *Eoabelisaurus* but different from *Skorpiovenator* and *Carnotaurus* where the transverse process surpasses the caudal centrum. The neural spine is anteroposteriorly longer than the previous vertebrae, as occurs in *Eoabelisaurus* y *Skorpiovenator*. Moreover,

in *Aucasaurus* and mentioned abelisaurids the neural spine has a length of two thirds with respect the neural arch.

In dorsal view (Fig. 12C), the transverse process lacks the posterior process of the lateral margin. The awl-like anterior process is more slender than in the third vertebra, and is more anteriorly developed than *Skorpiovenator*. The anterior rim of the transverse process is sinusoidal, whereas the posterior one is slightly convex, unlike *Skorpiovenator* where both rims are straight. The lateral rim has a sinusoidal shape, being the posterior half convex and the anterior half concave, different from the straight rim observed in *Skorpiovenator*. The posterolateral rugosity is conspicuous. The scar present in the middle of the transverse process, near the anterior border, is no longer present. The neural spine is leaf-shaped in cross-section.

In posterior view (Fig. 12D), the outline of the neural canal entry is taller than wide and triangular in outline. The hyposphene is prominent and subtriangular, unlike *Eoabelisaurus* that has a reduced hyposphene. Laterally to the hyposphene, the pocdf is shallow with a pneumatic foramen, which is absent in *Eoabelisaurus* and *Skorpiovenator*. The postzygapophyses are partially broken, they are ventrolaterally oriented and anteroposteriorly short, as in *Carnotaurus* but different from *Eoabelisaurus* and *Skorpiovenator* where the postzygapophyses are longer. Despite the bad preservation of the spol laminae, they are low mounds, implying a reduced spof with respect to the previous anterior caudal vertebrae, as in *Eoabelisaurus* and *Carnotaurus*.

*Fifth caudal vertebra* (Fig. 13; Table S1): The fifth and sixth caudal vertebrae are fused together with the proximal part of the fifth haemal arch, probably due to a pathology that occurred in an early ontogenetic stage, since the sizes of both centra are smaller than the preceding and subsequent vertebrae. The anterior articular surface of the centrum is oval in outline with the long axis dorsoventrally directed (Fig. 13A), as in fifth caudal vertebra of *Eoabelisaurus* and *Skorpiovenator*. The facet for the haemal arch contact is wide. On both sides, the depression below the neurocentral suture is shallow (Fig. 13B). The anterior rim of the lateral facet is partially broken, although it appears to be concave. A vertical furrow marks the posterior rim, which divide the fifth caudal centrum for the sixth one. The posterior articular surface is not visible, although it appears to have an oval outline, as in *Eoabelisaurus* and *Skorpiovenator* but unlike the circular outline observed in *Kurupi*. The posteroventral end is not visible, due to the pathological fusion with the haemal arch. Ventrally (Fig. 13F), a longitudinal groove is present, as in *Kurupi*.

In anterior view (Fig. 13A), the hypantrum is wide and high, whereas in *Eoabelisaurus* is low. The prezygapophysis is nearly vertically positioned, thus its articular facet is oriented almost completely medially, as in *Skorpiovenator* but different from *Eoabelisaurus* and *Carnotaurus* in which the prezygapophysis is dorsomedially oriented. The sprf is transversely and anteroposteriorly reduced with respect to the previous vertebrae. *Aucasaurus* lacks the septum that divide the sprf in two subfossae observed in *Carnotaurus*. The transverse process is dorsally directed with an inclination of 60°, as in *Carnotaurus* and different from *Eoabelisaurus*, *Kurupi*, and *Skorpiovenator* that show a lesser inclination. The neural spine is transversely thin and presents a distal swelling, as in *Skorpiovenator* and *Carnotaurus*, whereas it is absent in *Eoabelisaurus*.

In lateral view (Figs. 13B and 13E), the lateral rim of the right transverse process (the left one is broken) shows a pronounced roughness, which is absent in the fifth caudal vertebra of *Skorpiovenator* and *Carnotaurus*. However, it does not show the ventral torsion of the lateral rim of the fourth caudal vertebra. Moreover, the depression between the lateral rim of the transverse process and the lateral crest of the cdl is shallower than the fourth caudal vertebra. The cdl is prominent and ends laterally with an oblique ridge, which is longitudinal directed in *Carnotaurus* and absent in *Skorpiovenator*. The prcdf is deep but without pneumatic foramina. The transverse process is significantly posterior directed extending beyond the posterior articular surface, as in *Skorpiovenator* and *Carnotaurus* but different from *Eoabelisaurus* where the transverse process is directed laterally. In lateral view, the neural spine is almost complete, being anteroposteriorly shorter and dorsoventrally lower than the previous vertebrae. A similar condition is observed in *Eoabelisaurus*, whereas in *Skorpiovenator* the neural spine is anteroposteriorly longer. In *Aucasaurus*, there is a low process in the ventral portion of the anterior and posterior rims of the neural spine, as in *Carnotaurus*. The dorsal swelling of the neural spine shows lateral striae, probably designed for ligament attachment.

In dorsal view (Fig. 13C), the transverse process has a sinusoidal lateral rim, as in *Carnotaurus* and different from a straight lateral rim observed in *Majungasaurus* and *Skorpiovenator*. In *Aucasaurus*, the awl-like process of the lateral rim of the transverse process is anteroposteriorly reduced compared to the previous vertebrae. Conversely, in *Skorpiovenator* this structure increases slightly in size. *Aucasaurus* shows a concave anterior rim and sinusoidal posterior rim of the transverse process. The scar at the posterolateral corner is more marked than in *Carnotaurus*. The transverse process is anteroposteriorly reduced compared to the previous caudal vertebrae. At the base of the neural spine, especially on the right side, there is a small pneumatic foramen. The dorsal swelling of the neural spine has a rectangular outline. The neural spine preserves only one of the anteriorly directed processes, and the posterior ones are missing. These processes possibly are present in *Carnotaurus* but absent in *Eoabelisaurus* and *Skorpiovenator*.

*Sixth caudal vertebra* (Fig. 13; Table S1): As previously mentioned, the sixth caudal vertebra is fused to the fifth one. Consequently, the morphology of the anterior surface of the sixth caudal is not discernible. However, it seems to have an oval outline being taller than wide, as in *Eoabelisaurus* and *Skorpiovenator*. In lateral view (Fig. 13B), despite the collapsed right side, the centrum lacks depression below the neurocentral suture. The posterior rim of the centrum remains convex. The posterior surface presents a concavity more pronounced than in all previous vertebrae and is elliptical in outline with its major axis directed dorsoventrally (Fig. 13D), as in *Eoabelisaurus*. In ventral view (Fig. 13F), a low keel runs across the surface anteroposteriorly, bounding, on the left, a longitudinal groove.

Due to the fusion with its preceding vertebra, it is not possible to observe the morphology of the anterior portion of the neural arch. In lateral view (Figs. 13B and 13E), the prezygapophysis shows a strong medial inclination being greater than 60°, as in *Skorpiovenator* but unlike *Eoabelisaurus* and *Carnotaurus* that shows a lower inclination. The left transverse process is partially preserved lacking the distal end, whereas the right

one is broken at the base, therefore it is not possible to appreciate the morphology of the lateral end. In the ventral part of the transverse process, a conspicuous cdl is visible giving to the transverse process a triangular cross-section, as in *Skorpiovenator* and *Carnotaurus*. The cdf is deep, bounded anterior and posteriorly by prominent acdl and pcdl. The neural spine is almost complete, and it is anteroposteriorly slender than the fifth caudal vertebra. The dorsal part of the spine is laterally thickened, with longitudinal scars for ligament attachment.

In dorsal view (Fig. 13C), the neural spine holds the anterior processes, whereas lost the posterior ones. These processes and the lateral swelling of the distal part of the neural spine are absent in *Eoabelisaurus* and *Skorpiovenator*. The preserved portion of both transverse processes has a slightly concave anterior rim and a sigmoid posterior one. Moreover, the transverse processes are projected beyond the centrum. The pneumatic foramina present at the base of the neural spine are anteriorly placed with respect to the previous vertebra.

In posterior view (Fig. 13D), the neural canal has an elliptical outline. Dorsally to the canal, the hyposphene is dorsoventrally reduced but transversely wider than the previous vertebrae, as in *Eoabelisaurus* and *Carnotaurus*. *Aucasaurus*, unlike *Eoabelisaurus*, has straight lateral surfaces of the hyposphene, whereas they are concave in the Jurassic taxon. The pocdf include a pneumatic foramen, absent in *Eoabelisaurus* and *Skorpiovenator*. The spof is transversely narrow; this condition differs from a wider fossa in *Eoabelisaurus*, whereas in *Skorpiovenator* disappears. The postzygapophysis has a lateroventral orientation and it does not surpass posteriorly the centrum. The transverse process shows a strong dorsal inclination, as in *Carnotaurus*, but it differ from *Eoabelisaurus* and *Skorpiovenator* in that the latter has a lesser dorsal inclination.

*Seventh caudal vertebra* (Fig. 14; Table S1): The seventh caudal vertebra lacks only the left transverse process. The centrum is dorsoventrally lower than the previous vertebrae. The anterior surface has an oval outline and is almost flat (Fig. 14A). The anterior articular facet for the haemal arch of this vertebra is transversely and dorsoventrally wider than the anterior vertebrae. In lateral view (Fig. 14B), the surface lacks of the depression below the neurocentral suture, as in *Skorpiovenator*. The anterior and posterior rims are straight and parallel to each other, giving a subrectangular contour. In posterior view (Fig. 14D), the surface is oval with the articulation facet for the haemal arch anteroposteriorly wide, as in *Kurupi*. In ventral view (Fig. 14F), the groove runs anteroposteriorly along the entire surface, unlike *Kurupi* where is appreciable only near the contact surfaces for the haemal arches. Laterally and posterior to the groove, there are nutrient foramina.

In anterior view (Fig. 14A), the neural arch is transversely narrower than the anterior vertebrae. The neural canal is tall with an oval outline. Like the previous vertebrae, the preserved portion of the prezygapophyses show a strong medial orientation, as observed in *Skorpiovenator* but different from *Eoabelisaurus* and *Carnotaurus* where they show a lesser medial inclination. The hypantrum is not preserved but we consider that it was reduced, based on the reduction of the hyposphene of the sixth vertebra. The sprf is transversely narrow and the sprl, although partially preserved, is reduced compared to the most anterior vertebrae. In *Aucasaurus*, the transverse process has the same dorsal inclination of the fifth and sixth vertebrae as in *Carnotaurus*, whereas *Eoabelisaurus*, *Kurupi*, and

*Skorpiovenator* have a less inclined transverse process. The neural spine shows lateral expansion in its most dorsal portion, which is absent in *Eoabelisaurus*, *Skorpiovenator*, and *Carnotaurus*.

In lateral view (Figs. 14B and 14E), the prdl is prominent with a posterior displacement of the transverse process, as in *Skorpiovenator* but unlike *Eoabelisaurus* in which the transverse process occupies a central position with respect to the neural arch. Ventrally to the transverse process, the lateral rim of the process has a rough texture.
The anteroposterior ridge that marks where the cdl ends is less marked than in the previous vertebrae, whereas the accessory ridge (vlrtp) is prominent as in *Kurupi*. The acdl and pcdl are well-developed, bounding a deep cdf, as in *Kurupi*. Anterior to the acdl, the prcdf occupies almost half of the anteroposterior length of the neural arch, unlike *Eoabelisaurus* where it is less developed. The anterior process of the base of the neural spine is more conspicuous than the previous vertebrae, while the posterior one is only partially preserved. These processes are absent in the same vertebra of *Eoabelisaurus* and *Skorpiovenator*, while in *Carnotaurus* only the posterior one is observed. In the distalmost portion of the neural spine, the surface has lateral roughness, as in *Skorpiovenator*.

In dorsal view (Fig. 14C), the posterolateral scar is well-developed turning a posterior directed process. The awl-shaped anterior process is slender and anteroposteriorly long and its lateral rim is strongly sinusoidal, as in *Kurupi* and *Carnotaurus*. The anterior rim of the transverse process is concave, while the posterior one is sinusoidal. At the base of the neural spine, the pneumatic foramina have an oval contour. The neural spine is located in the posterior half of the neural arch. The anterior and posterior processes of the neural spine are present but incomplete.

In posterior view (Fig. 14D), the neural canal shows a heart-like outline.
The hyposphene is reduced with respect to the sixth vertebra but still prominent, as in *Kurupi*. Laterally to the hyposphene, the pocdf has a reduced pneumatic foramen, which is absent in *Eoabelisaurus*, *Skorpiovenator*, and *Carnotaurus*. The postzygapophyses are poorly preserved therefore it is impossible to deduce size and shape. The spof, as in the sixth caudal vertebrae, is a fissure, whereas in *Eoabelisaurus* it is transversely wider.

*Eighth caudal vertebra* (Fig. 15; Table S1): The eighth caudal vertebra is almost completely preserved, lacking only the left transverse process. In anterior view (Fig. 15A), the centrum shows a similar morphology of the seventh caudal vertebra, except for a more pronounced concavity of the articular surface. In lateral view (Fig. 15B), as in the previous vertebra the centrum has a subrectangular outline. Despite the collapsing of the lateral surfaces, they lack the depression below the neurocentral suture. In posterior view (Fig. 15D), the articular surface is broken on the left side, although it shows a drop-like outline due to narrowing of the dorsal portion, unlike *Eoabelisaurus* and *Skorpiovenator* that have an oval contour. The articulation surface with the haemal arch is wide. In ventral view (Fig. 15F), the longitudinal groove is deeper towards the posterior end of the surface, forming two low tubercles in correspondence of the articular facet for the haemal arch. These tubercles are observed in all following vertebrae.

In anterior view (Fig. 15A), the neural arch is transversely narrow. The prezygapophysis has an almost vertical orientation, and the hypantrum is dorsoventrally deep although it is
transversely narrower than the seventh vertebra. A similar condition is observed in *Skorpiovenator*, whereas *Eoabelisaurus* has a less inclined prezygapophysis and a reduced hypantrum. The sprf is shallower and laterally reduced than the previous vertebra. The right transverse process is less dorsally inclined than the seventh caudal vertebrae, whereas in *Eoabelisaurus* it is horizontal. The neural spine shows a transverse reduction of the dorsal swelling.

In lateral view (Figs. 15B and 15E), the transverse process is positioned on the posterior portion of the neural arch, as in *Skorpiovenator* but different from *Eoabelisaurus* that has a centrally positioned transverse process. The awl-like process is partially preserved on the right side. The lateral rim of the transverse process is ornamented by roughness. On the ventral surface of the transverse process, the accessory ridge is rugose. The cdl is less prominent than the previous vertebrae, and the acdl and pcdl are low, as in *Skorpiovenator*. The prcdf is shallow but anteroposteriorly long. The neural spine is anteroposteriorly reduced than the seventh caudal vertebra, and positioned on the posterior half of the neural arch. The anterior process of the basal neural spine was partially preserved, giving to the latter an L-like shape. The dorsal end of the neural spine has several longitudinal ridges.

In dorsal view (Fig. 15C), the transverse process is mediolaterally larger than the previous vertebra. The posterolateral process is reduced to a scar. The prezygapophysis slightly surpasses the centrum. The pneumatic foramina present at the base of the neural spine are conspicuous. The dorsal swelling of the neural spine is transversely reduced when compared with the seventh caudal vertebra.

In posterior view (Fig. 15D), the hyposphene is dorsoventrally low, and the postzygapophyses are partially broken. The foramen inside the pocdf is reduced with respect to previous vertebrae. The spof is transversely narrow, unlike in *Eoabelisaurus* where this fossa is subcircular.

*Ninth caudal vertebra* (Fig. 16; Table S1): The ninth caudal vertebra is complete excepting the neural spine. The centrum shows a circular outline of the anterior surface and a strong concavity due to a marked rim (Fig. 16A), unlike *Eoabelisaurus* and *Skorpiovenator* that have an oval anterior contour. In lateral view (Fig. 16B), the anterior and posterior rims of the centrum are slightly convex. In posterior view (Fig. 16D), the surface, like the anterior one, has a circular outline and is strongly concave due to a prominent rim, different from the oval outline present in *Eoabelisaurus* and *Skorpiovenator*. The posterior facet for the haemal arch is wide. The ventral groove is deep and slightly wider than the previous vertebrae (Fig. 16F).

In anterior view (Fig. 16A), the hypantrum is lacking. The prezygapophysis has a medial inclination greater than 60°, as *Skorpiovenator*. The sprf is transversely narrow, anteroposteriorly long, and has a septum on the bottom, unlike *Eoabelisaurus* that has a reduced and circular fossa. In *Aucasaurus* the sprl is reduced to low mound. The transverse process shows the same dorsal inclination of the eighth caudal vertebra. The neural spine is preserved only at the base.

In lateral view (Figs. 16B and 16E), the prezygapophysis is projected dorsally and anteriorly, surpassing the anterior rim of the centrum. This dorsal inclination increases in posterior caudal vertebrae. The transverse process slightly exceeds the posterior rim of the

 

centrum. The lateral border of the transverse process has an irregular surface due to the presence of a marked roughness, especially for the presence of a conspicuous tubercle. That tubercle is present up to the twelfth caudal vertebra. The cdl ends laterally at the lateral rim of the transverse process, and is no longer separated in the acdl and pcdl. For this reason, the cdf disappears, different from *Eoabelisaurus* where the acdl, pcdl, and cdf are still present. The prcdf extends far the half of the neural arch, whereas in *Eoabelisaurus* is anteroposteriorly reduced. The postzygapophysis does not exceed the centrum posteriorly.

In dorsal view (Fig. 16C), the pneumatic foramina at the base of the neural spine are wider than the previous vertebrae. The posterolateral scar of the transverse process is present but incipient. The awl-like process is still well-developed. The lateral rim of the transverse process are sinusoidal.

In posterior view (Fig. 16D), the hyposphene is poorly developed, and the postzygapophysis is ventrolaterally oriented. The spof has a fissure-like morphology and is dorsoventrally reduced with respect to eighth caudal vertebra, unlike the shallow depression observed in *Eoabelisaurus*. The pocdf is shallow and holds a small pneumatic foramen.

*Tenth caudal vertebra* (Fig. 17; Table S1): The tenth caudal vertebra lacks the neural spine and the left transverse process. In anterior view, the centrum shows a circular outline and, as in the ninth caudal vertebra, has a marked rim giving the surface an accentuated concavity (Fig. 17A), unlike an oval surface present in *Eoabelisaurus*. The lateral surface has a subrectangular outline with straight anterior and posterior rims (Fig. 17B).
In posterior view (Fig. 17D), the presence of a fragment of the following vertebra prevents the observation of the articular surface, although the contour seems to be circular, different from the oval shape shown by *Eoabelisaurus*. In ventral view (Fig. 17F), the facet for the haemal arch articulation is reduced and the two low ridges bound the groove.

In anterior view (Fig. 17A), the neural canal is reduced and shows a circular outline. The prezygapophyses are partially broken, although they were reduced in size and strongly medially oriented. The sprf is anteroposteriorly reduced with respect to the ninth caudal vertebra and presents the vestige of a septum in its posteriormost portion, whereas in *Eoabelisaurus* this fossa is a shallow depression. The transverse process has a dorsal inclination of 30°.

In lateral view (Figs. 17B and 17E), the prezygapophysis is slightly dorsally directed and surpasses anteriorly the centrum. The right transverse process still presents a rugose accessory ridge on the ventral surface. The awl-like anterior process is conspicuous. Moreover, the posterior end of the transverse process has a reduced posteriorly projected process. The cdl is low, and the prcdf is reduced to an anteroposteriorly extended depression, different from *Eoabelisaurus* where the cdl is well-developed and the prcdf is deeper. In *Aucasaurus*, the pocdf is shallow, although a pneumatic foramen is present.

In dorsal view (Fig. 17C), the lateral rim of the transverse process has a pronounced tubercle on its middle portion. The posterolateral scar is barely developed. The transverse process is reduced anteroposteriorly with respect the previous vertebra, and the anterior and posterior rims are slightly concave, unlike *Eoabelisaurus* where the anterior rim is convex and the posterior one is sinusoidal. The foramina at the base of the neural spine are

deep and wide, being the right one slightly wider. In posterior view (Fig. 17D), the neural canal has a circular outline. It is not possible to observe the morphology of the neural spine, spol, spof, and postzygapophysis because they are poorly preserved.

*Eleventh caudal vertebra* (Fig. 18; Table S1): As in the preceding vertebra, the eleventh caudal vertebra lacks the neural spine and left transverse process. In anterior view (Fig. 18A), the surface is circular in outline, and ventrally the facet for the haemal arch articulation is greatly reduced, whereas in *Eoabelisaurus* the anterior contour is slightly oval. In lateral view (Fig. 18B), the anterior and posterior rims of the centrum are slightly convex. In posterior view (Fig. 18D), the articular surface is strongly concave and in its ventral end the surface for contact with the haemal arch is wider than the anterior one. In ventral view (Fig. 18F), the groove is anteroposteriorly reduced than the tenth vertebra, running for three quarter of the whole surface.

In anterior view (Fig. 18A), the neural canal is circular. The prezygapophyses, even though incomplete, are further away from each other than in the preceding vertebrae. However, the articular facet of the prezygapophysis is medially directed. The sprf disappears from this vertebra, as in *Eoabelisaurus*. The right transverse process is almost horizontally directed.

In lateral view (Figs. 18B and 18E), the prezygapophysis exceeds anteriorly the centrum, as in *Eoabelisaurus*. The transverse process has the same morphology and orientation of the tenth caudal vertebra. The prcdf is shallow and anteroposteriorly reduced, as in *Eoabelisaurus*. The cdl is poorly developed and the accessory ridge of the transverse process is still present.

In dorsal view (Fig. 18C), the shaft of the transverse process is shorter than the previous vertebra. The anterior and posterior rims of the transverse process are concave, but lack a posterior process. The posterolateral scar is barely developed. The lateral border of the transverse process is anteroposteriorly longer than the neural arch. The left pneumatic foramen at the base of the neural spine is wider than the right one. The postzygapophyses are partially preserved, surpassing the posterior rim of the centrum such as in *Eoabelisaurus*.

In posterior view (Fig. 18D), the pneumatic foramen of the pocdf disappears. A deep fossa stands out between the postzygapophyses, forming a shelf dorsally to the neural canal. This fossa is absent in all the middle caudal vertebrae of *Eoabelisaurus*.

*Twelfth and Thirteenth caudal vertebrae* (Fig. 19; Table S1): The twelfth and thirteenth caudal vertebrae remain articulated. The right prezygapophysis, most of the neural spine, and the left transverse process are missing in the twelfth vertebra. The thirteenth caudal vertebra has lost most of the neural spine, the two transverse processes, the postzygapophyses and the posterior half of the centrum. The anterior articular surface of the centrum of both vertebra is circular in outline (Fig. 19A), although it appears slightly wider than tall with respect to the eleventh caudal vertebra. Conversely, *Eoabelisaurus* shows an oval outline. In lateral view (Fig. 19B), both vertebrae have a flat surface without pleurocoels or depressions. The posterior articular surface of the twelfth caudal vertebra seems to have a circular outline. In ventral view (Fig. 19F), both vertebrae have the groove that runs anteroposteriorly for three quarter of the surface.

In anterior view (Fig. 19A), the articular surfaces of the prezygapophyses are widely spaced and are strongly medially inclined (being almost vertical in the thirteenth caudal vertebra), unlike *Eoabelisaurus* where they have a lesser medial inclination. The neural canal opens 2 cm away from the dorsal rim of the anterior articular surface. The transverse process has a dorsal inclination of 10° to 15°.

In lateral view (Figs. 19B and 19E), the prezygapophysis is anterodorsally projected, surpassing the centrum anteriorly. Moreover, it has a rugose protuberance directed dorsolaterally. A similar structure is also present in the megaraptoran *Aoniraptor* (*Motta et al., 2016*). The transverse process of the twelfth caudal vertebra is almost identical, in shape and morphology, to the previous vertebra. The neural spine is positioned in the posterior half of the neural arch and is "L"-shaped, since there is a low ridge that runs anteriorly from the neural spine to a small process. The right postzygapophysis of the twelfth caudal vertebra arises posterodorsally, ending with the posteriormost portion almost horizontally. Moreover, it exceeds the centrum posteriorly. Conversely, *Eoabelisaurus* has the postzygapophysis that does not exceed the centrum.

In dorsal view (Fig. 19C), the pneumatic foramina at the neural spine base disappear in both vertebrae, replace by shallow depressions. The right transverse process of twelfth caudal vertebra has anterior and posterior borders straight and parallel to each other. The awl-like process is conspicuous, surpassing the anterior surface of the centrum. In this view, the lateral rim of the transverse is markedly sinusoidal with the presence of a prominent tubercle. The posterolateral scar is reduced to a low prominence. In posterior view, the twelfth caudal vertebra has a fossa between the two postzygapophyses, as in the previous one; this region is not preserved in thirteenth caudal vertebra (Fig. 19D).

*Posterior caudal vertebrae*: The holotype of *Aucasaurus garridoi* MCF-PVPH-236 includes two incomplete posterior centra. Both elements were partially separated from the neural arch and preserve only a portion of a concave and circular outlined anterior articular surface, different from *Elemgasem* that shows oval outlines. The anterodorsal surfaces of the centra preserved the base of the prezygapophyses. Laterally, the centra have a low anteroposteriorly directed ridge with no pits or depressions. The ventral surface shows a faint anteroposteriorly directed ridge bounded laterally by two grooves, in proximity to the articular facet for the haemal arch.

*Other caudal vertebrae remains* (Figs. 20 and 21): Two isolated neural spines (Figs. 20A–20D), are interpreted as belonging to some of the anterior caudal vertebrae due to their anteroposterior length, reduced transverse width, and morphology of their distal end. In anterior view, both spines are transversely narrow with an expanded distal end.

In lateral view (Figs. 20A and 20B), the distalmost portion of both neural spines is dorsally convex. In addition, they presents several longitudinal grooves and ridges on the lateral surface of the expanded portion. This distal swelling is separated from the ventral part of the neural spine by a marked step. The anterior and posterior rims are rugose due to the attachment of interspinous ligaments. In dorsal view (Figs. 20C and 20D), both anterior and posterior interspinous processes are visible. The anterior processes are separated by a concavity deeper than the posterior ones.

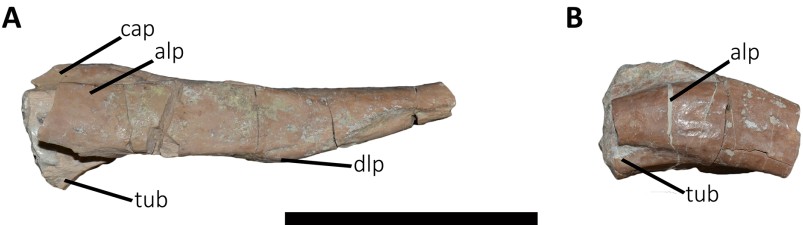

**Figure 22 Proximal fragments of two cervical ribs of *Aucasaurus garridoi* MCF-PVPH-23.** In lateral (A and B) views. alp, anterolateral process; cap, capitulum; dlp, dorsolateral process; tub, tuberculum. Scale bar: 5 cm.

Two differently-sized isolated transverse processes (Figs. 21A–21D) are interpreted as belonging to anterior caudal vertebrae. The anterior awl-like processes are well-developed (Figs. 21A and 21B). The lateral rims are convex, rugose and turn somewhat ventrally. In the posterolateral corner, the scar is conspicuous. In ventral view (Figs. 21C and 21D), the cdl ends laterally in the anteroposteriorly directed ridge.

*Cervical ribs* (Fig. 22): The cervical ribs are fragmentary, since preserved only two proximal ends. These two elements are similar in morphology, differing slightly in size (Figs. 22A and 22B). Both fragments preserved up to where the tuberculum and capitulum split, although lacking the articular portions, and the base of the anterolateral process. Thus, the proximal end of the cervical ribs shows a triradiate morphology. Based on their morphology, we considered that these ribs belong to the posterior portion of the neck, between the seventh and the ninth element. In fact, the preserved ribs of *Aucasaurus* are similar to the seventh to ninth cervical ribs of *Carnotaurus* and *Majungasaurus*, since the dorsolateral processes of these elements is reduced to a low mound in all specimens. The dorsal rim of the cervical fragments is sinusoidal due to the presence of the dorsolateral process, while the ventral one is concave. Moreover, *Aucasaurus*, like *Carnotaurus* and *Majungasaurus*, has a subrectangular-shaped proximal end of the posterior ribs in lateral view, whereas other large theropods have a subtriangular proximal end (*e.g.*, *Allosaurus*, *Tyrannosaurus*).

*Dorsal ribs* (Fig. 23): Several dorsal rib fragments are preserved (Figs. 23A–23G), some corresponding to the anterior region of the trunk and others to the abdominal region (Figs. 23A–23C and 23E–23G). Additionally, several tubercula are preserved separate from the rib shafts (Fig. 23D). The dorsal ribs of *Aucasaurus* present well-defined tuberculum and capitulum, and the tuberculum separated from the capitotubercular lamina as in *Majungasaurus*, but unlike *Carnotaurus* and MAU-Pv-LI 665 where the tubercula are in line with the lamina or slightly offset. The articular surfaces of the tubercula and capitula are oval in outline, although the former is broader. The capitotubercular lamina is thin and has a more pronounced concavity than in *Carnotaurus*. The capitula are triangular in lateral view, widening towards the rib shaft (Fig. 23A). Pneumatic foramina are not observed, as in *Majungasaurus* but unlike *Carnotaurus*, *Ceratosaurus*, *Masiakasaurus*, and MAU-Pv-LI 665 that have pneumatic dorsal ribs. Anteriorly and posteriorly, intercostal ridges runs from the tuberculum towards the shaft (Figs. 23A–23C), as in *Niebla*. Noteworthy, it is the presence of a roughness in the proximal part of the anterior

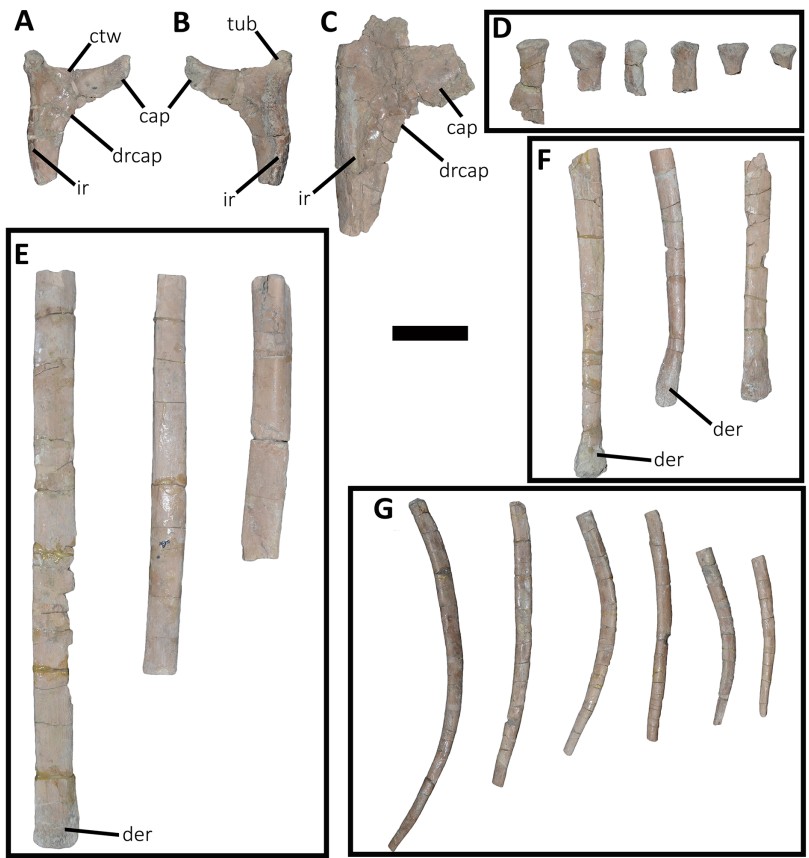

**Figure 23 Fragments of dorsal ribs of *Aucasaurus garridoi* MCF-PVPH-236.** In lateral (A and C–G) and medial (B) views. cap, capitulum; ctw, capitotuberculum web; der, distal expansion of rib; drcap, distal ridge of capitulum; ir, intercostal ridge; tub, tuberculum. Scale bar: 5 cm.

intercostal ridge that would be the area of insertion of some soft tissue. From the capitulum, a ridge runs distally on the medial portion of the shaft, giving to the proximal end a T-shaped cross-section, as in other abelisaurids (MAU-Pv-LI 665 and MMCh-PV 48), whereas the middle portion of the shafts have a triangular cross-section, as in *Niebla*. Distal fragments of proximal dorsal ribs show an oval cross-section, ending distally with a rectangular shape (Fig. 23E), as in *Majungasaurus*. Distal fragments of posterior ribs taper distally and someone ends with a pronounced swelling (Fig. 23F).

*Gastralia* (Fig. 24): Multiple fragments of gastralia are preserved (Figs. 24A–24D); some of them show the median suture between middle elements (Figs. 24A and 24B), others represent portions of the diaphysis of middle or lateral elements (Figs. 24C and 24D). Among them, two middle elements are almost completely preserved (Figs. 24A and 24B), lacking only the proximal end of the shafts.

The middle gastralium elements are completely fused (Figs. 24A and 24B), creating an angle of approximately 80°. In ventral or dorsal view, the shafts of the middle elements have a sinusoidal morphology, being laterally convex in their proximal half and laterally concave in their distal half. In the left middle element, a ventrolateral groove is the site

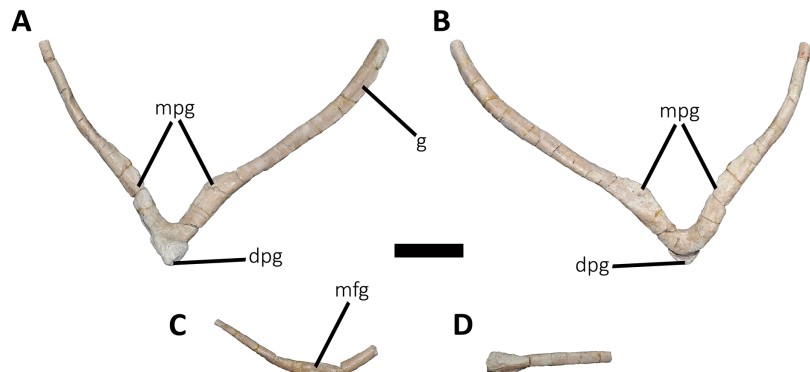

**Figure 24 Gastralia of *Aucasaurus garridoi* MCF-PVPH-236.** In ventral (A, C, and D) and dorsal (B) views. dpg, distal process of gastralia; g, groove; mfg, medial fusion of gastralia; mpg, medial process of the gastralium. Scale bar: 5 cm.

where articulated the lateral element (Fig. 24A), such as observed in several theropods (*e.g.*, *Norell & Makovicky, 1997*; *Chure, 2000*; *Allain & Chure, 2002*; *Claessens, 2004*). The shaft of these gastralia has a cylindrical shape for almost its whole length; however, there is a wing-like process with a rugose surface in the distal portion of the shaft (Figs. 24A and 24B). This medially directed process has a cross-section with teardrop-shaped outline.

The two middle elements form a distal process with a triangular outline, with the apex directed ventrodistally (Figs. 24A and 24B). This process forms a dorsal platform, possibly contacting the following middle gastralia, unlike the imbricate-type system observed in tyrannosaurids and allosaurids (*Claessens, 2004*). *Poekilopleuron* and possibly *Juravenator* have a distal process (*Allain & Chure, 2002*; *Chiappe & Göhlich, 2010*), which is less developed than *Aucasaurus*. A chevron-shaped morphology with an acute angle is typical of the posteriormost gastralia in several theropods, such as in *Acrocanthosaurus*, *Poekilopleuron*, *Tyrannosaurus*, or *Troodon* (*Harris, 1998*; *Chure, 2000*; *Allain & Chure, 2002*; *Claessens, 2004*). However, these taxa lack the triangular distal process observed in *Aucasaurus*.

*Haemal arches* (Fig. 25 and 26; Table S2): Twelve haemal arches are preserved in articulation with their corresponding vertebrae (Figs. 25A–25H and 26A–26E). A proximal fragment of a more distal haemal arch and three fragments from the middle portion of the shaft of two distal haemal arches are also preserved. The first three haemal arches show the articular surface open proximally, with a "V"-shaped haemal canal (Figs. 25A–25C, in anterior and posterior views). This morphology differs from that in *Camarillasaurus*, *Majungasaurus*, *Ilokelesia*, and *Carnotaurus* where canal is dorsally closed. This trait was originally considered an autapomorphyic condition of *Aucasaurus* (*Coria, Chiappe & Dingus, 2002*). In the fourth haemal arch, the proximal end is partially fused anteriorly (Fig. 25D). From the fifth haemal arch until the last one preserved, the proximal end of the haemal canal is fully closed (Figs. 25E–25H and 26A–26E). In the first to four haemal arch, the articulation surfaces for the centra are divided in four facets, two of them directed anteroproximally and two posteroproximally (Figs. 25A–25D). Since the fifth and the following haemal arches have a completely closed canal, the articular surfaces

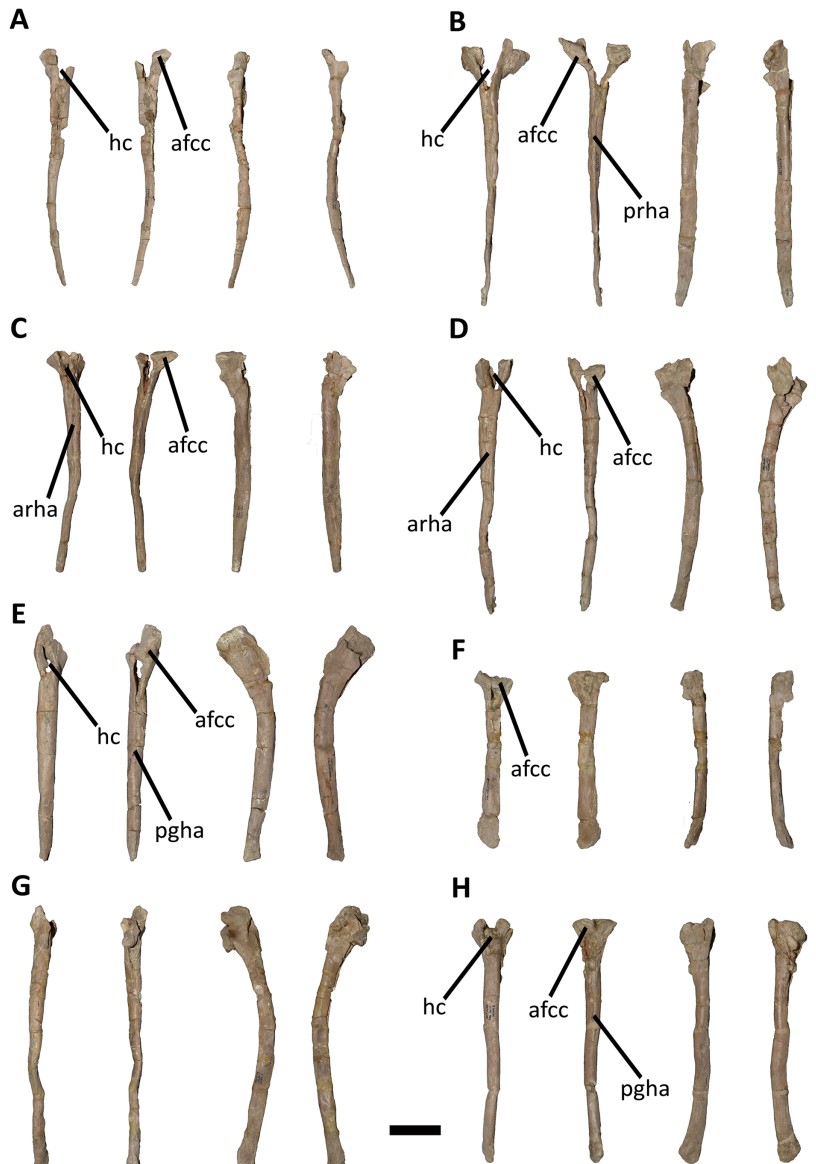

**Figure 25 First to eighth haemal arches of *Aucasaurus garridoi* MCF-PVPH-236.** In anterior, posterior, and lateral (A–H; from left to right) views. afcc, articular facet for the caudal centrum; arha, anterior ridge of haemal arch; hc, haemal canal; pgha, posterior groove of the haemal arch; prha, posterior ridge of the haemal arch. Scale bar: 5 cm.

for the centra are reduced to two facet, the first one inclined anteroproximally and the second one posteroproximally (Figs. 25E–25H and 26A–26E).

The anteroproximal articular surface, which articulates with the posteroventral end of the previous centrum, is generally wider than the posteroproximal surface along the entire series of haemal arches. This morphology is also reflected in the size of the articular surface for the haemal arches of the centra, where the posteroventral facet is wider than the anteroventral one. Anteriorly to the anteroproximal surface and separate from it, there are

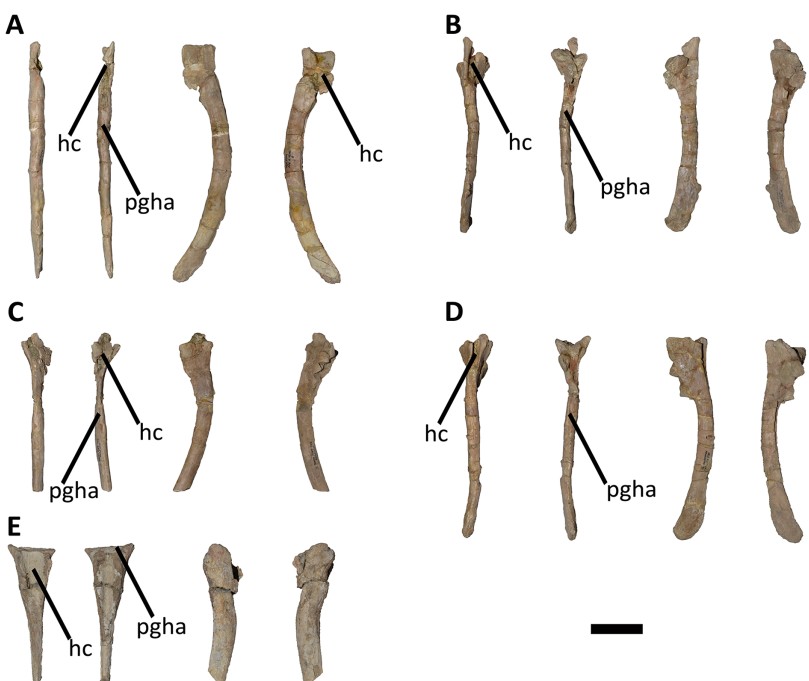

**Figure 26 Ninth to thirteenth haemal arches of *Aucasaurus garridoi* MCF-PVPH-236.** In anterior, posterior, and lateral (A–E; from left to right) views. hc, haemal canal; pgha, posterior groove of the haemal arc. Scale bar: 5 cm.

two proximally directed processes. However, the separation among them is shallower posteriorly. Moreover, these two processes are connected with the haemal shaft by ridges.

In anterior view, the haemal canal of the anterior haemal arches has a triangular outline (Figs. 25A–25D), whereas it shows a drop-shaped outline from the fifth to the last element (Figs. 25E–25H and 26A–26E). In this view, the shaft distal to the haemal canal is transversely flat or slightly concave with the presence of a rough ridge in the middle of the surface and directed distally (Figs. 25C and 25D). This morphology is also observed in *Majungasaurus*, *Ilokelesia*, and *Carnotaurus*, but unlike *Camarillasaurus* where there is a groove that crosses the entire anterior surface of the shaft.

In lateral view, the proximal end of all haemal arches have a triangular outline (Figs. 25A–25H and 26A–26E), due to anterior and posterior projections of the articular surface. Distally to the proximal rim, the surface is proximodistally concave, due to the lateral bowing of the proximal articular surface. The shaft is straight in the anteriormost haemal arches, whereas it curve backwards in the remaining haemal arches. The lateral surfaces of the shaft show an anteroposterior convexity throughout the series.

In posterior view, there are two processes (visible at least in the first to four haemal arches; Figs. 25A–25D) positioned distally to the posteroproximal articular surface and connected distally to the shaft by a ridge. The shaft of the first to third elements holds a rough ridge that runs the whole length (Fig. 25B). From the fourth haemal arch onwards, a groove replaces the aforementioned ridge, reaching up to the mid-shaft (Figs. 25E and 25H), as in *Ilokelesia*, *Carnotaurus*, and *Camarillasaurus*. The shaft of the first to four

haemal arches shows a triangular cross-section proximally, while distally it develops an oval cross-section. The remaining haemal arches show a heart-shaped cross-section of the proximal portion of the shaft, whereas they have a lenticular cross-section distally.

The morphology of the fifth and sixth haemal arches stand out among the entire series in that their size does not follow the normal posterior size reduction (Figs. 25E–25F). In fact, the fifth haemal arch is more robust than the other ones, whereas the sixth haemal arch is reduced in size when compared to other haemal arches. Therefore, the morphology of these two haemal arches is likely due to the pathology observed in the fifth and sixth caudal vertebrae. The three distal haemal arch fragments correspond to a distal portion of a haemal canal with the proximal portion of the shaft, and two fragments of shafts that exhibit the proximodistal groove on the posterior surface.

## Further comparisons

We compare the caudal series of *Aucasaurus* with other taxa in which the precise position of the vertebrae is uncertain; comparisons exclude the autapomorphic traits of *Aucasaurus garridoi*, which are unique to this taxon.

Several named and unnamed abelisaurids preserved caudal elements, allowing a direct comparison with *Aucasaurus*. The indeterminate abelisaurid MPM 99 preserves three anterior caudal vertebrae, one of the proximal portion of the tail and the other two vertebrae from the mid-posterior portion of the anterior region of the tail. *Aucasaurus* differs from MPM 99 in having the transverse processes strongly dorsally inclined; in the latter specimen these processes are slightly dorsally inclined or horizontally directed. Conversely to *Aucasaurus*, MPM 99 has straight and smooth lateral rims of the transverse processes. However, the caudal neural spine in MPM 99 presents a widening of the dorsal end with two reduced dorsal processes directed anteriorly and posteriorly, as in *Aucasaurus*. In addition, both specimens share the presence of the awl-like projection of the transverse processes, a marked posterior scar on the dorsal surface of the processes, prominent cdl, acdl, and pcdl, and the presence of a groove on the ventral surface of the centrum.

The holotype of *Ekrixinatosaurus* (MUCPv 294) includes several anterior and middle caudal vertebrae. *Aucasaurus* and *Ekrixinatosaurus* share a well-developed hyposphene in the anterior caudal vertebrae, a prominent cdl that divides ventrally in the acdl and pcdl, and a dorsal swelling of the neural spine. However, *Ekrixinatosaurus* shows a lesser dorsal inclination of the transverse processes of the anterior vertebrae, and it lacks the dorsal processes of neural spines and the groove on the ventral surface of caudal centra.

*Tralkasaurus* is a brachyrostran abelisaurid from the same litostratigraphic unit of *Huinculsaurus*, *Ilokelesia*, and *Skorpiovenator*. The holotype of *Tralkasaurus* comprises anterior caudal vertebrae that differs from *Aucasaurus* in having transverse processes less inclined with prominent posterior awl-like projections and straight lateral rims.

*Aucasaurus* and *Viavenator* share several morphological features observable in the anterior caudal vertebrae. Both taxa have anterior caudal vertebrae with articular surfaces taller than wide, lateral surfaces of the centra with a parallelogram-shaped outline without pleurocoels. With respect to neural arches, both abelisaurids share the presence of

dorsoventrally-developed and a strongly medially inclined prezygapophysis, a wide hypantrum, a sinusoidal lateral rim of the transverse process with an awl-like anterior process, a ridge at the lateral end of the cdl, and the presence of a septum at the bottom of the sprf. Moreover, they have transverse processes longer than the anteroposterior length of the centra, prominent acdl, pcd, and cdl (the latter ending laterally with a ridge), the presence of a posterodorsal scar, strongly sinusoidal lateral rim, and reduced or absent posterior process (unlike basal forms such as *Ekrixinatosaurus*, *Ilokelesia*, and *Skorpiovenator*). However, *Aucasaurus* presents a deeper ventral groove on the centra and slightly more inclined transverse processes. It is noteworthy the presence of two isolated transverse processes of a indeterminate abelisaurid (MAU-Pv-LI 547) from the same geological levels of *Viavenator*, which shows a convex or sinusoidal lateral rim and a ventral longitudinal ridge similar to those in *Aucasaurus*.

*Aucasaurus* also shows similarities and differences with the anterior caudal vertebra of MACN-PV-RN 1012. In fact, these abelisaurids have centra with a longitudinal groove on the ventral surface and lack pleurocoels on the lateral surface. The sprf in MACN-PV-RN 1012 has a septum that divided it in two areas, as observed in some vertebrae of *Aucasaurus*. With respect to neural arch, MACN-PV-RN 1012 has a conspicuous anterior awl-like projection and a longitudinal ventrolateral ridge in the transverse process, like in *Aucasaurus*. However, *Aucasaurus* differs from MACN-PV-RN 1012 in having more inclined transverse processes with straight or slightly concave posterior rims. Moreover, the latter abelisaurid has a ventral surface of the transverse process with a cdl positioned more anteriorly when compared with *Aucasaurus*.

The anterior caudal vertebra of MPCN PV 69 has an overall similar morphology to the anteriormost caudal vertebrae of *Aucasaurus*. However, all the anterior caudal vertebrae of the latter (except the first) present a groove on the ventral surface of the centrum, which is absent in MPCN PV 69.

The indeterminate abelisauroid MPEF PV 1699/1-2 constitutes of two anterior caudal vertebrae from the La Paloma Formation (Hauterivian-Barremian, Lower Cretaceous) of Chubut Province (Argentina). *Aucasaurus* and MPEF PV 1699/1-2 share the presence of a groove on the ventral surface of the centra, transversely long transverse processes, a well-developed hyposphene-hypantrum articulation, and prominent cdl, acdl, and pcdl. However, *Aucasaurus* has a more medially inclined prezygapophysis and a dorsal inclination of the transverse process greater than MPEF PV 1699/1-2. Although both vertebrae of this Early Cretaceous specimen show somewhat lateral expansion of the transverse processes, their fragmentary preservation prevents determining the presence of anterior awl-like projections.

The anterior caudal vertebrae of the Brazilian *Pycnonemosaurus* and *Aucasaurus* share a ventral groove on the centra, transverse processes with an anterior awl-like projection, and prominent hyposphene. However, the former abelisaurid shows less inclined transverse processes and prezygapophyses. *Spectrovenator*, another Brazilian abelisaurid, has transverse processes with evident anteroposterior awl-like processes and straight lateral rims, unlike *Aucasasurus* that has only anterior prominent awl-like projections and sinusoidal lateral rims.

With respect to *Majungasaurus*, *Aucasaurus* shares with the Malagasy abelisaurid the presence of a ventral groove on the anterior centrum, transversely long transverse processes, and a dorsal expansion of the neural spines. However, *Majungasaurus* differs from *Aucasaurus* in having a less medially inclined prezygapophysis, a transverse process that is less dorsally inclined and lacks an awl-like projection, absence of accessory processes on the dorsal neural spines, and absence of a distinct hyposphene-hypantrum articulation.

The anterior caudal centra of *Aucasaurus* differ from the anterior caudal vertebrae of *Rajasaurus* in the absence of an anteroposteriorly directed keel on the ventral surface of the latter. A second Indian taxon, *Rahiolisaurus*, has well-developed cdl, acdl, and pcdl, as in *Aucasaurus*, but the transverse process is less inclined.

The French abelisaurid *Arcovenator* is the most complete laurasian abelisaurid to include anterior caudal vertebrae. This taxon shares with *Aucasaurus* the presence of a strongly medially tilted prezygapophysis, but unlike the latter, the transverse process is nearly horizontal and the hyposphene is reduced.

## DISCUSSION

### Phylogenetic analysis

The first round of our cladistics analysis recovered most parsimonious trees (MPTs) on 161 replicates of a total 1,000 replicates, resulting in 1610 MPTs (10 MPTs per each replicate) with a length of 556 steps, a consistent index of 0.493, and a retention index of 0.725. However, the second round of TBR found more than 50000 MPTs, due to an overflow of trees in the memory space. The strict consensus shows a large polytomy among all ceratosaurs (Fig. 27A), and the IterPCR procedure detected 11 unstable taxa: *Afromimus*, *Berberosaurus*, *Dahalokely*, *Huinculsaurus*, *Kryptops*, *Kurupi*, *Quilmesaurus*, *Rahiolisaurus*, *Thanos*, MNN-Tig6, and MPCN-PV-69. When these "wildcards" were *a posteriori* pruned, the internal relationships among Ceratosauria were better solved. Major internal clades were recovered, such as Majungasaurinae, Brachyrostra, and Furileusauria; although, some polytomies are observed among more inclusive majungasaurines and among furileusaurians (Fig. 27B). The 100 replicates of Jackknife found 22 unstable taxa, 20 final nodes, and a nodal support average of 72.2 (Data S3). The unique node with a value of 100% is Neotheropoda. Regarding Abelisauridae, this clade is recovered with a value of 73%, whereas all internal nodes show values lower than 85% except for the node *Spectrovenator* plus more derived abelisaurids (97%) (Data S3).

Previously, *Aucasaurus* has been recovered as a derived abelisaurid by several phylogenetic studies, which disagree from each other in the proposed sibling relatioships of this taxon. Most of the phylogenetic analyses regarded *Carnotaurus* as sister taxon of *Aucasaurus* (*Coria, Chiappe & Dingus, 2002*; *Calvo, Rubilar-Rogers & Moreno, 2004*; *Canale et al., 2009*, *2016*; *Pol & Rauhut, 2012*; *Farke & Sertich, 2013*; *Gianechini et al., 2015*; *Rauhut & Carrano, 2016*; *Longrich et al., 2017*; *Baiano, Coria & Cau, 2020*). However, other analyses have recovered either *Abelisaurus* (*Filippi et al., 2016*; *Delcourt, 2018*; *Cerroni et al., 2020*; *Gianechini et al., 2021*; *Agnolín et al., 2022*) or *Niebla* (*Baiano et al., 2022*) as the closest taxon to *Aucasaurus*. Our analysis nests *Aucasaurus* in an unresolved

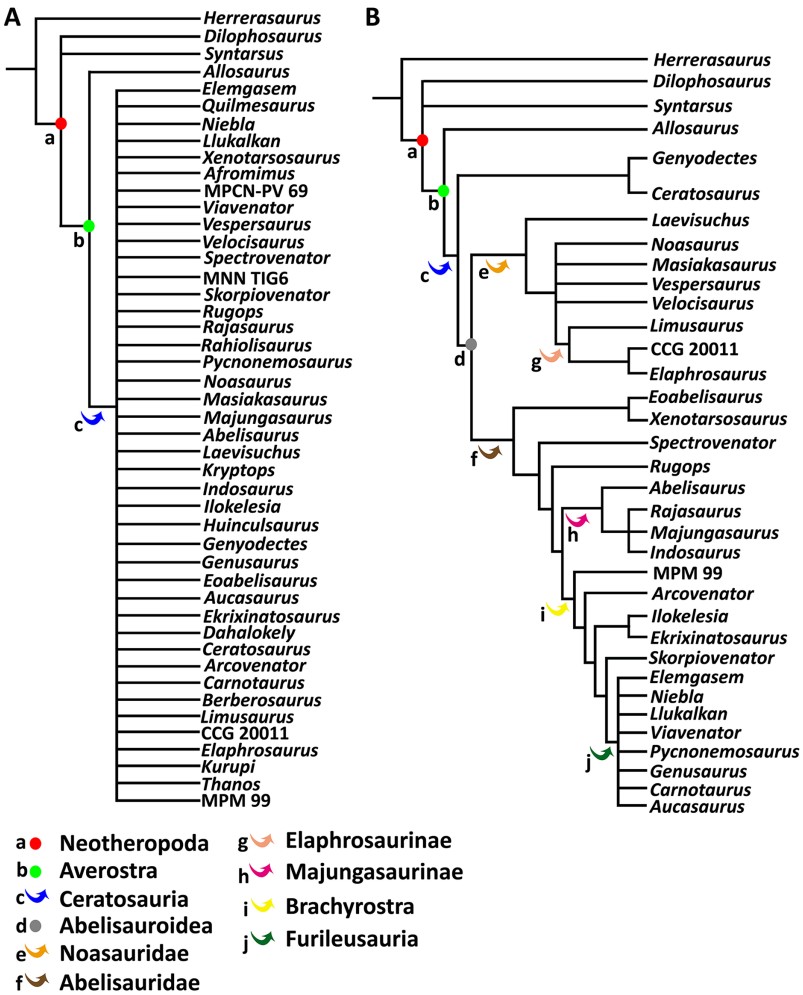

**Figure 27** Phylogenetic relationships of *Aucasaurus garridoi* MCF-PVPH-236. The results show a quite unresolved strisct consensus (A), and a more resolved topology of the reduced consensus. (B) Colored dots were used for node-based taxa, colored arrows for stem-based taxa.

brachyrostran furileusaurian clade, and confirms several phylogenies (*e.g.*, *Filippi et al., 2016*; *Gianechini et al., 2021*; *Baiano et al., 2022*) recovering *Carnotaurus*, *Elemgasem*, *Genusaurus*, *Llukalkan*, *Niebla*, *Pycnonemosaurus*, and *Viavenator* (Fig. 27B) within the same clade.

Irrespective of which taxon is most closely related to *Aucasaurus*, the latter shares axial apomorphies with other abelisaurids that should be considered in future phylogenetic analyses of abelisaurids. Based on these, Abelisauridae (including *Aucasaurus*) is diagnosed by having caudal vertebrae with reduced neural spines when compared to posterior dorsal vertebrae (ch. 139:1) and caudal vertebrae with well-defined anterior and posterior centrodiapophyseal laminae (ch. 141:2). Furthermore, in *Aucasaurus* the bases of the neural arch of the anterior caudals are wider than the mid-centrum (ch. 142:1), a condition shared by several abelisaurids (plus *Kurupi* and MPCN-PV-69) and *Masiakasaurus*. *Aucasaurus*, *Spectrovenator*, and more nested abelisaurids (plus *Kurupi*)

have anterior and middle caudal vertebra expanded posteriorly (ch. 144:1), a condition reverted in *Majungasaurus* where they are not expanded. *Aucasaurus* shares with Majungasaurinae and Brachyrostra (plus *Kurupi*) the presence of caudal vertebrae with transverse processes that are more than 1.4 times the length of caudal centra (ch. 147:1). *Aucasaurus*, *Majungasaurus*, and Brachyrostra (plus *Dahalokely*) have cervical vertebrae with postaxial tear-shaped zygapophyses (ch. 107:1). *Aucasaurus* has tall prezygapophyses-hypantrum complex (ch. 240:1), a condition shared with MPM 99, *Arcovenator*, and several brachyrostrans. Moreover, *Aucasaurus*, MPM 99, and brachyrostrans present transverse processes directed dorsolaterally (ch. 244:2) (although in *Aucasaurus* and *Carnotaurus* this condition is exacerbated). Addtionally, the inclusion of *Aucasaurus* within Brachyrostra is supported by the presence of the following synapomorphies: anterior caudal vertebrae with an inclination of the prezygapophyses greater than 50° (ch. 242:1), and anterior caudal vertebrae with a ventrolateral ridge at the lateral end of the transverse processes (ch. 245:1). Finally, the inclusion of *Aucasaurus* within furileusaurians is supported by the presence of cervical epipophyses with an anterior prong (ch. 112:1; condition shared also with *Noasaurus* and *Rahiolisaurus*) and a sinusoidal lateral rim of the anterior and middle caudal vertebrae (ch. 246:2).

## Autapomorphic axial traits in *Aucasaurus*

Several traits in the cervical, dorsal, and caudal vertebrae of abelisaurids distinguish this group from any other theropod clade, however, detailed description of the axial skeleton have been produced for only a few taxa: *Carnotaurus* (*Bonaparte, Novas & Coria, 1990*; *Méndez, 2014a*), *Majungasaurus* (*O'Connor, 2007*), and *Viavenator* (*Filippi et al., 2018*). These studies have allowed us to identify new autapomorphic traits for *Aucasaurus*, which are discussed below.

*Odontoid with a triangular outline in dorsal view* (Fig. 28): The odontoid of *Aucasaurus* preserved in anatomical articulation with the dorsal surface of the atlas intercentrum. When compared to other ceratosaurian theropods (*e.g.*, *Masiakasaurus*, *Thanos*, *Majungasaurus*, *Carnotaurus*), and even some tetanuran theropods (*e.g.*, *Allosaurus*, *Tyrannosaurus*), the odontoid of *Aucasaurus* is more triangular in dorsal view (Figs. 28A and 28D).

*Atlas with a subcircular articular surface* (Fig. 29): This morphology of the atlas is absent in several medium and large theropods, including those abelisaurids in which the atlas is known. A transversely oval atlas is seen in *Herrerasaurs*, *Dilophosaurus*, *Ceratosaurus*, *Allosaurus*, *Sinraptor*, *Aerosteon*, and *Tyrannosaurus*, where the occipital condyle is also wider than tall. Among abelisaurids, only *Viavenator* shows a similar condition as *Aucasaurus*, but in the former it is slightly wider than tall producing an oval contour; in *Carnotaurus* the articular surface of the atlas is strongly transversely oval (Figs. 29B and 29C).

*Interspinous accessory processes extended to sacral and caudal neural spines* (Fig. 29): The interspinous ligament scar on the neural spines of cervical and dorsal vertebrae is a feature present in several theropods (*Foth et al., 2015*; *Wilson et al., 2016*; see also the chapter Discussion). However, some ceratosaur theropods show anteriorly and/or

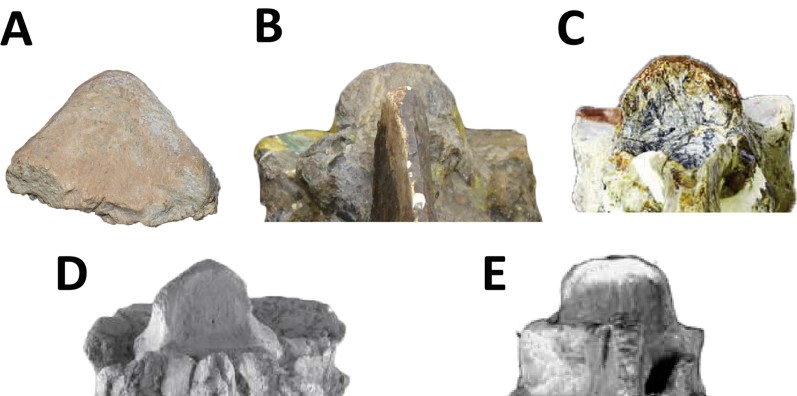

**Figure 28 Photographs of autapomorphies of *Aucasaurus garridoi*.** Dorsal view of the odontoids of *Aucasaurus* (A), *Carnotaurus* (B), *Thanos* (C), *Majungasaurus* (modified from *O'Connor, 2007*) (E), and *Masiakasaurus* (modified from *Carrano, Loewen & Sertich, 2011*) (D). Image not to scale.

posteriorly expanded distal end of the neural spine, giving to this spine a fan-shaped outline. Moreover, some theropods have the distal portion of the dorsal neural spines with well-developed processes. These morphologies imply some accessory interspinous ligamental insertion among consecutive vertebrae. A fan-shaped neural spine is present in the noasaurids *Elaphrosaurus* (*Rauhut & Carrano, 2016*) and *Huinculsaurus* (*Baiano, Coria & Cau, 2020*). Furthermore, fan-shaped neural spines of the dorsal vertebrae are present in several coelurosaurs, such as the compsognathids *Compsognathus*, *Sinocalliopteryx*, and *Sinosauropteryx* (*Currie & Chen, 2001*; *Peyer, 2006*; *Ji et al., 2007*). Distal accessory interspinous process can be observed in *Dilophosaurus* (*Welles, 1984*; *Marsh & Rowe, 2020*), *Dahalokely* (*Farke & Sertich, 2013*), and *Siats* (*Zanno & Makovicky, 2013*). However, a contact among consecutive accessory interspinous processes was first reported in the dorsal vertebrae of the abelisaurid *Viavenator* (*Filippi et al., 2016*; Fig. 6). In fact, Filippi and colleagues proposed this condition as an autapomorphic trait for *Viavenator*. Here we show that this condition is also present in *Aucasaurus*, although in this taxon it is present in the dorsal, sacral, and caudal vertebrae (Figs. 29D–29F).

*A tubercle lateral to the prezygapophysis of middle and posterior caudal vertebrae* (Fig. 29): The presence of a rough tubercle on the lateral surface of the prezygapophysis of the middle and posterior caudal vertebrae is absent in other abelisaurids that preserved elements of this section of the tail (Fig. 29G). *Motta et al. (2016)* mentioned the presence of a low swelling on the lateral prezygapophysis for the megaraptoran *Aoniraptor*. Some tyrannosaurids, such as *Alioramus*, *Tarbosaurus*, and *Tyrannosaurus*, have a bulge on the ventral side of the prezygapophysis (Fig. 29H) of the posterior caudal vertebrae (*Brusatte, Carr & Norell, 2012*), which is different from *Aucasaurus*.

*Presence of pneumatic foramina laterally to the base of the neural spine in the anterior caudal vertebrae* (Fig. 29): Pneumaticity (fossae or foramina) on the dorsal surface of the neural arch is a condition present in several theropods. For instance, the noasaurid *Elaphrosaurus* and the theropod *Spinostropheus* have shallow fossae on the dorsal surface of the cervical transverse processes (*Carrano & Sampson, 2008*; *Rauhut & Carrano, 2016*).

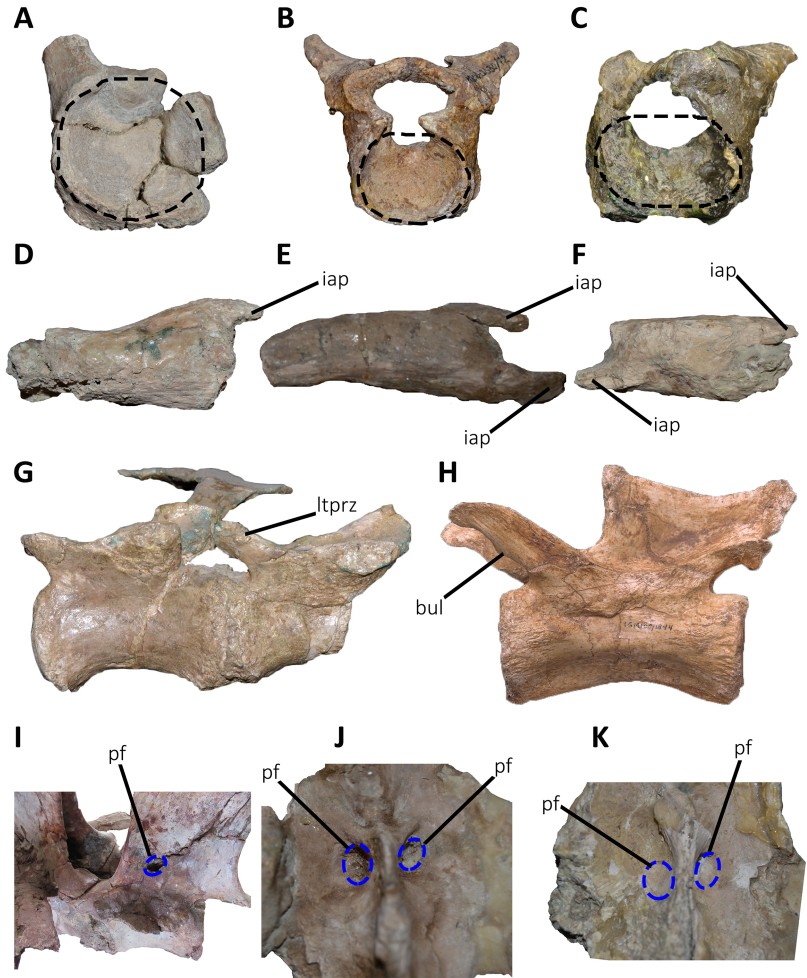

**Figure 29 Photographs of autapomorphies of *Aucasaurus garridoi*.** Outline (in red dashed line) of the anterior articular surface of the atlas of *Aucasaurus* (A), *Viavenator* (B), and *Carnotaurus* (C). Interspinous accessory processes on the dorsal (D), sacral (E), and caudal (F) neural spines of *Aucasaurus*. Lateral tubercle of prezygapophysis in the middle caudal vertebrae of *Aucasaurus* (G), and ventral bulge on prezygapophysis of the posterior caudal vertebrae of *Alioramus* (modified from *Brusatte, Carr & Norell, 2012*) (H). Foramina on the dorsal surface of the caudal neural arch in *Meraxes* (I), whereas *Aucasaurus* holds pneumatic foramina on the dorsal surface of the neural arches (framed by blue dashed lines) of the ninth (J) and eleventh (K) caudal vertebrae. Image not to scale.

The paravian *Unenlagia* present deep fossae with internal foramina laterally to the base of the neural spine of the thirteenth dorsal vertebrae. The foramina possibly communicate with the internal neural arch. This trait is regarded as a peculiar condition of *Unenlagia*, due to its absence in other non-avian theropods (*Novas et al., 2021*; *Gianechini & Zurriaguz, 2021*). Few groups of theropods show pneumatic traits with external manifestation in their caudal vertebrae. A pleurocoel is present on the lateral surface of the centra of Megaraptora, Oviraptorosauria, Therizinosauria, and possibly *Torvosaurus* (*e.g.*, *Britt, 1991*, *1993*; *Zhang et al., 2001*; *Xu et al., 2007*; *Zanno et al., 2009*; *Benson, Carrano & Brusatte, 2010*; *Balanoff & Norell, 2012*). However, Megaraptora is the only clade with highly pneumatized caudal vertebrae, extending to the centra and the neural arches (*Coria*

*& Currie, 2016*; *Motta et al., 2016*; *Aranciaga Rolando, Garcia Marsá & Novas, 2020*). To date, the only theropods to exhibit foramina on the dorsal surface of the caudal neural arches are *Acrocanthosaurus* and *Meraxes* (Fig. 29I), while *Giganotosaurus* has only shallow depressions (*Britt, 1993*; *Aranciaga Rolando, Garcia Marsá & Novas, 2020*; *Canale et al., 2022*). Thus, the presence of foramina laterally to the neural spine of the anterior to middle caudal vertebrae of *Aucasaurus* (Figs. 29J and 29K) is considered an autapomorphic condition for this abelisaurid (see Discussion).

*A marked rugosity with a prominent tubercle on the lateral rim of the transverse processes of caudal vertebrae fourth to twelfth* (Fig. 30): Among abelisaurids the transverse processes of the anterior and middle caudal vertebrae exhibits a unique morphology, being extremely specialized in the Brachyrostra clade. The latter group includes abelisaurids with anteroposteriorly expanded lateral end of the transverse processes and a straight or concave lateral rim (Coria & Salgado, 2000; *Calvo, Rubilar-Rogers & Moreno, 2004*; *Canale et al., 2009*). More derived brachyrostran, such as the furileusaurians *Aucasaurus*, *Carnotaurus*, and *Viavenator*, have extremely developed an anterior awl-like projection on the lateral end of the transverse processes. Furthermore, the lateral rim of the caudal transverse processes in these abelisaurids is extremely convex, becoming concave laterally to the awl-like processes. However, *Aucasaurus* exhibits ornamentation on the lateral rim, with the presence of a prominent tubercle and rugosity (Figs. 30A–30C), whereas in *Carnotaurus* and *Viavenator* this trait is far less developed.

*Presence of a small ligamentous scar near the anterior edge of the dorsal surface in anteriormost caudal transverse processes* (Fig. 30): *Aucasaurus* also differs from other abelisaurids in having an anterodorsal scar on the middle portion of the transverse processes (Figs. 30D and 30E). Such scar is particularly visible in the first to sixth caudal vertebrae, disappearing further distally. Despite the fact that this morphology seems unique among abelisaurids, the recently described *Kurupi* (*Iori et al., 2021*) is diagnosed by strikingly conspicuous, cuneiform processes located in the same area of *Aucasaurus*'s scar (Fig. 30F; see also Discussion).

*Distinct triangular process located at the fusion point of posterior gastralia* (Fig. 30): Among ceratosaurs, *Masiakasaurus*, *Aucasaurus*, and *Majungasaurus* are the only taxa preserving gastral elements, although described as pathological in the latter (*Gutherz et al., 2020*). The middle gastralia of *Aucasaurus* are fused to each other medially, forming a conspicuous triangular, ventral process (Figs. 30G and 30H) that could have either articulated with the subsequent gastral element and/or receive the ligamental insertion of *m. rectus abdominis*.

*Anterior haemal arches with the neural canal proximally open* (Fig. 30): *Coria, Chiappe & Dingus (2002)* mentioned the presence of proximal haemal arches with a proximal open canal (Figs. 30I and 30J) as an autapomorphic trait of *Aucasaurus*. This statement is based on the absence of this condition in other abelisaurids. While taphonomic or ontogenetic factors raise a note of caution regarding this interpretation, a taphonomic bias is unlikely due to: (1) the haemal arches were found perfectly articulated with the corresponding caudal vertebrae (*Coria, Chiappe & Dingus, 2002*; Fig. 2); (2) there is a gradually closure of the haemal canal from the first to four haemal arches. Ontogenetic causes can also be ruled

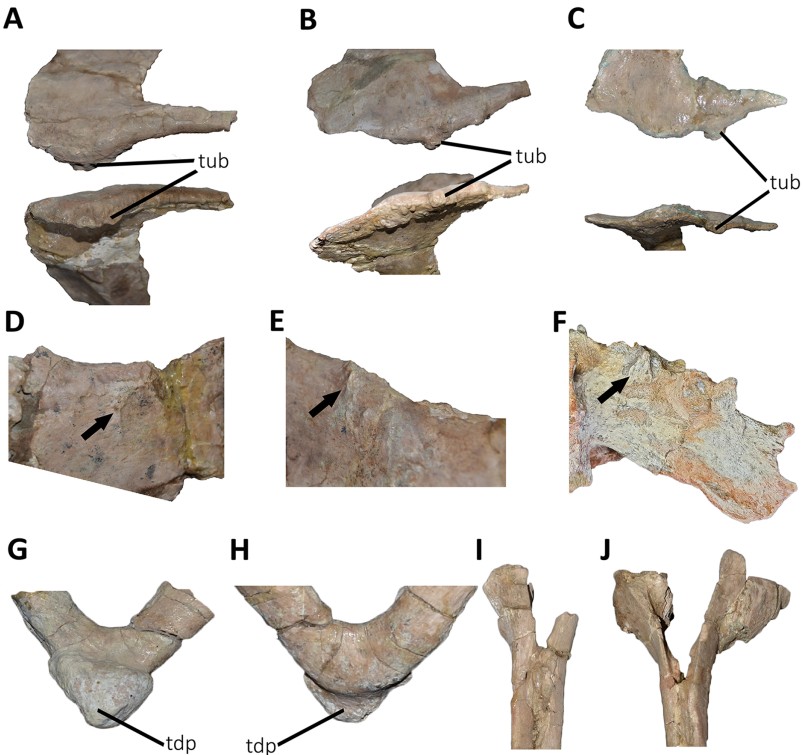

**Figure 30** **Photographs of autapomorphies of *Aucasaurus garridoi*.** Lateral rugosity and tubercle of the transverse processes of the fourth (A), ninth (B), and eleventh (C) caudal vertebrae of *Aucasaurus* in dorsal (upper) and lateral (lower) views. Anterodorsal scar (black arrows) of the transverse processes of the first (D), and second (E) caudal vertebrae of *Aucasaurus*, and cuneiform process (black arrow) on the anterodorsal surface of the anterior caudal vertebra of *Kurupi* (F). Triangular distal process (red lines) of posterior gastralia in ventral (G), and dorsal (H) views. Proximal portion of the first (I), and second (J) haemal arches showing a dorsally open haemal canal. Image not to scale.

out given that morphological traits (*e.g.*, obliterated vertebral neurocentral fusion, fused pelvic elements, fused distal ends of tibia and fibula with astragalocalcaneum; *Baiano, 2021*) and a recently histological study (*Baiano & Cerda, 2022*) confirm the somatic and sexual maturity of the holotype of *Aucasaurus*. Thus, for these reasons we consider this condition a valid autapomorphy for *Aucasasurus garridoi*.

## Inferences about Abelisauridae axial pneumaticity

CT scans show camellated tissue in the neural arches and centra (Figs. 31A–31Q). The camellated tissue present in the neural arches can be also seen around the foramina at the base of the neural spine of the first, fifth, sixth, ninth, twelfth and thirteenth caudal vertebrae.

Among living tetrapods, only birds are characterized by having extensive postcranial pneumaticity, but such pneumaticity was characteristic of several groups of extinct ornithodires, including pterosaurs and non-avian saurischian dinosaurs (*Owen, 1857*; *Seeley, 1870*; *Britt, 1993*; *Britt et al., 1998*; *O'Connor & Claessens, 2005*; *O'Connor, 2006*; *Sereno et al., 2008*; *Wedel, 2009*). Within non-avian saurischians, pneumaticity has been

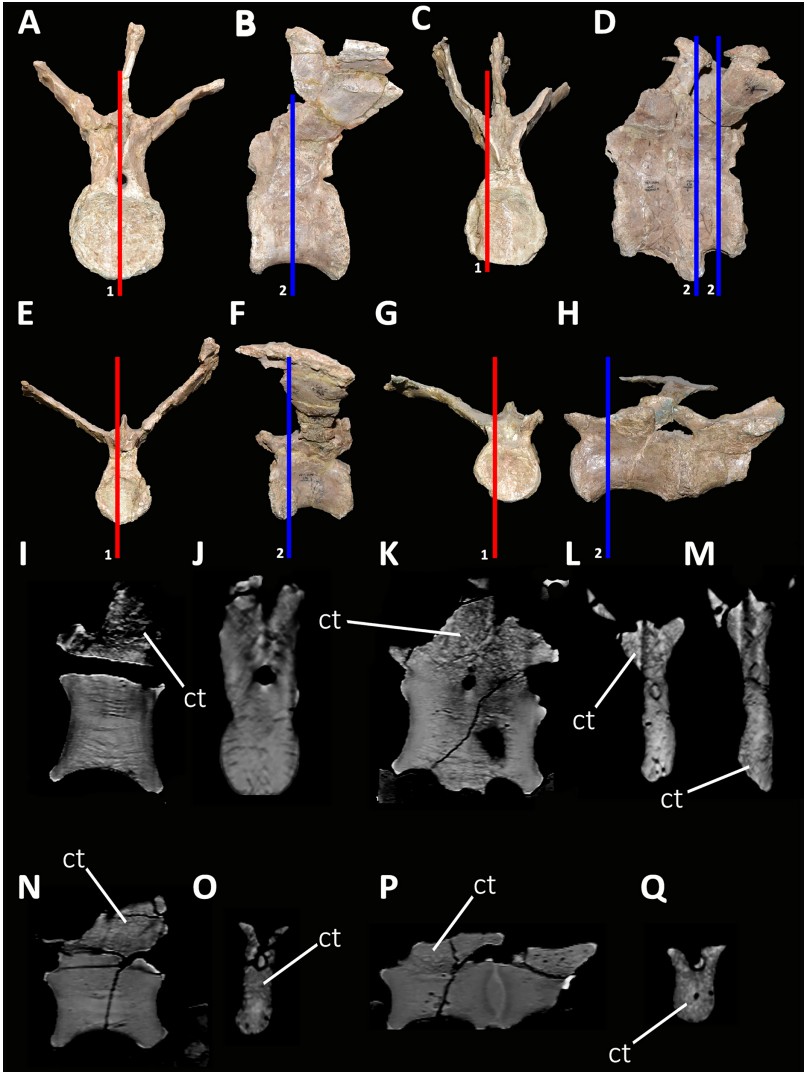

**Figure 31 Select computed tomography sections of selected caudal vertebrae of *Aucasaurus garridoi*.**
First (A, B, I and J), fifth and sixth (C, D, K and L), ninth (E, F, N and O), and twelfth and thirteenth (G, H, P and Q) caudal vertebrae in anterior (A, C, E and G), and posterior (B, D, F and H) views. Red lines indicate sagittal sections, while blue lines indicate transverse sections. ct, camellate tissue.

best-studied and documented in sauropods, much less so among non-avian theropods (*e.g.*, *O'Connor, 2007*; *Aranciaga Rolando, Garcia Marsá & Novas, 2020*; *Gianechini & Zurriaguz, 2021*). Postcranial skeletal pneumaticity (PSP) is often manifested by the presence of foramina piercing cortical bone, especially of vertebrae, and connecting with chambers inside these elements (*O'Connor, 2006*). *Aucasaurus garridoi* presents two sets of foramina in the caudal vertebrae: at the basis of the spine (Figs. 29J and 29K) and inside the pocdf (Figs. 32A–32C). The first set of foramina, visible from the fifth to eleventh caudal vertebrae, is here considered an autapomorphy of this taxon. These foramina also show homogeneity in size among the right and left side (Table S3). The foramina located inside the pocdf also show homogeneity among the right and the left side, at least until the ninth

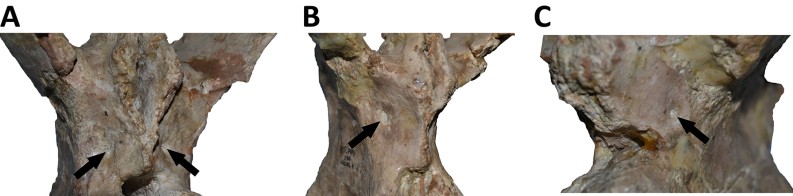

**Figure 32 Photographs of possible external correlates of pneumaticity in *Aucasaurus garridoi*.** Foramina (black arrows) whitin the pocdf of the first (A), fourth (B), and ninth (C) caudal vertebrae of *Aucasaurus*.

vertebra (Table S3). These external correlates of pneumatization are also present in the caudal vertebrae of *Carnotaurus* and in an unpublished caudal series of an abelisaurid (MPEF 10826; *Gasparini et al., 2015*) from Patagonia, Argentina.

Although the structures described above have characteristics of pneumatic foramina (*Britt, 1993*), the resolution of the CT scans makes it difficult to discern a connection between these foramina and the internal chambers or camellated tissue; however, an incipient camellated tissue at the basis of the spines is visible, thus supporting the assumed connection between air cells and the external pneumatopore. Unfortunately, the resolution of the CT scan also precludes to determinate the presence of internal connections between the foramina located in the postzygapophyseal centrodiapophyseal fossa (pocdf) and the internal airspaces of the vertebral centra. Once again, the CT scans does show what appears to correspond to camellated tissue inside all scanned vertebrae (Figs. 31I–31Q).

Regarding the postsacral portion of the axial skeleton, PSP is present at least in three brachyrostran abelisaurids (*Aucasaurus*, *Kurupi*, and MPM 99) with camellated tissue in the centra and the neural arches of the anterior caudal vertebrae (Figs. 33A and 33B). Pneumatic caudal vertebrae are so far unknown in Majungasaurinae, although only *Majungasaurus* was subject to such type of study (*O'Connor, 2007*). Moreover, noasaurids such as *Masiakasaurus* or *Vesperasaurus* also have apneumatic caudal vertebrae (*Carrano, Sampson & Forster, 2002*; *Carrano, Loewen & Sertich, 2011*; *Langer et al., 2019*). Therefore, the presence of the pneumatic traits in the caudal series, at least in the anterior portion, could be a unique condition of brachyrostran abelisaurids within the clade Ceratosauria, although more studies using CT imaging are needed, especially among basal ceratosaurs, nosasaurids, majungasaurines, and more derived brachyrostrans. Other clades that have signs of pneumaticity along the tail include Carcharodontosauridae, Megaraptora, Ornithomimosauria, Therizinosauroidea, Oviraptorosauria, and possibly *Torvosaurus* (*Britt, 1991*, *1993*; *Benson et al., 2012*; *Novas et al., 2013*; *Watanabe et al., 2015*; *Aranciaga Rolando, Garcia Marsá & Novas, 2020*). There is a different degree of pneumaticity among these taxa, being highest in Megaraptora and lowest in Carcharodontosauria (*Aranciaga Rolando, Garcia Marsá & Novas, 2020*; Fig. 10). While megaraptorans have extensively pneumatized anterior and middle caudal neural arches and centra (*e.g.*, *Aranciaga Rolando, Garcia Marsá & Novas, 2020*), carcharodontosaurids show moderate pneumatization only in the arches of the anterior vertebrae (*Britt, 1993*). Among other theropod groups, Ornithomimosauria shows evidence of pneumatization in only the

neural arches of the anterior and middle caudal vertebrae (*Watanabe et al., 2015*), while in Therizinosauroidea, penumaticity is observed mainly in the anterior vertebrae (neural arch and centrum; *e.g.*, *Zanno et al., 2009*; *Zanno, 2010*). Finally, oviraptorosaurs hold pneumatic foramina in anterior, middle, and posterior caudal centra (*e.g.*, *Xu et al., 2007*; *Balanoff & Norell, 2012*). Among non-tetanuran theropods (and possibly among non-avetheropodan theropods), Brachyrostra appears to be the only clade characterized by having pneumatic caudal vertebrae, as shown in the present study. Such diversified pattern of the pneumaticity among the caudal series of different theropod groups supports hypotheses of independent evolution among these lineages (*Benson et al., 2012*).

Finally, the pattern of pneumaticity identified in *Aucasaurus* and other abelisaurid taxa appears to support the "neural arch first" hypothesis of *Benson et al. (2012)*, in which the pneumaticity of the posterior axial skeleton develops from the neural arches into the centra. This assumption is due to the location of foramina and associated camellated tissue in the caudal vertebrae of *Aucasaurus garridoi*, a condition corresponding to a highly conserved pattern of pneumatization in theropods (*Benson et al., 2012*).

## Implications for reduction of movements in the axial skeleton of abelisauridae

Skeletal stiffness and robustness in abelisaurids, especially among derived forms, was suggested by several authors and based primarily on craniocervical modifications of this group (*e.g.*, *O'Connor, 2007*; *Sampson & Witmer, 2007*; *Méndez, 2014a*, *2014b*; *Filippi et al., 2016*; *Delcourt, 2018*; *Gianechini et al., 2022*). Some studies have proposed specific behaviors for abelisaurids based on the peculiar features of the caudal portion of their skull, cervical vertebrae, and ribs (*e.g.*, hypertrophied epipophyses, low neural spines, ribs with aliform processes; *O'Connor, 2007*; *Sampson & Witmer, 2007*; *Delcourt, 2018*; *González, Baiano & Vidal, 2021*). Hence, behavioral inferences, especially as related to feeding habits and intraspecific behaviors, were tested by biomechanical analyses of the skull and/or the cervical portion of the axial skeleton (*Mazzetta, Fariña & Vizcaíno, 1998*; *Mazzetta et al., 2009*; *Therrien, Henderson & Ruff, 2005*; *Snively et al., 2011*).

The postcervical portion of the axial skeleton of abelisaurids—particularly Brachyrostra (*e.g.*, *Méndez, 2014b*)—also has features that are related to increased axial rigidity. For instance, abelisaurids (*e.g.*, *Majungasaurus*, *Aucasaurus*, *Carnotaurus*) have D-shaped transverse processes, which may have increased the surface for the attachment of robust epaxial musculature. Additionally, *Viavenator* holds conspicuous longitudinal ridges on the dorsal surface of the transverse processes, from the second to the ninth dorsal (Figs. 34A and 34B). The indeterminate abelisaurid MAU-Pv-LI 665 also has a similar ridge in the transverse processes of the dorsals (Figs. 34C and 34D). These structures are likely the insertion sites of ligaments of strong epaxial muscles, such as *m. longissimus dorsi* and/or *m. iliocostalis*. Furthermore, *Aucasaurus* and *Viavenator* have interspinous accessory articulations on the dorsal end of the neural spines that may correspond to ossified supraspinous ligaments. Despite *Filippi et al. (2016)* stated that these accessory processes are present on the posterior portion of the dorsal series, they appear to have been also present in the anterior and middle dorsal vertebrae (Figs. 34E and 34F). These processes

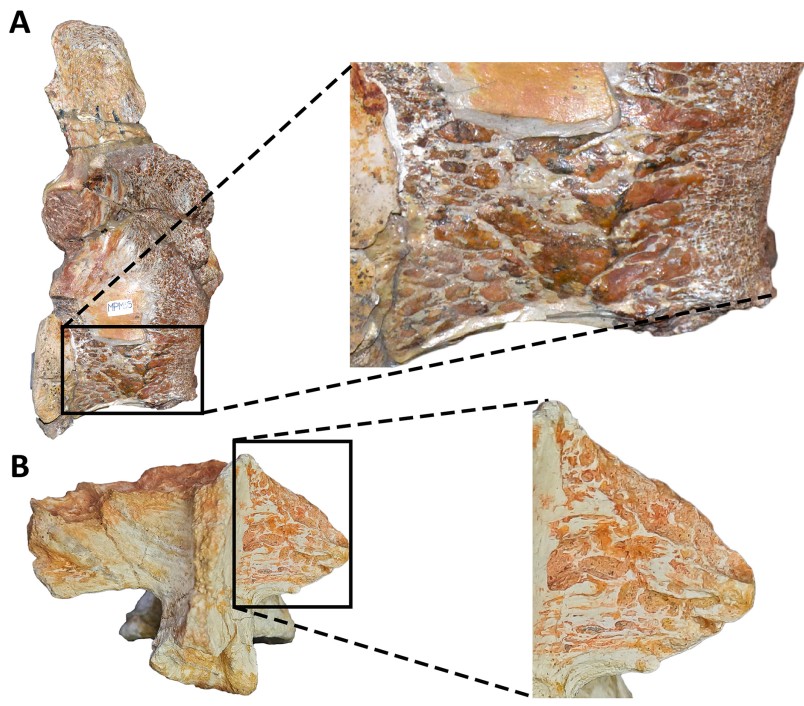

**Figure 33 Internal tissue in caudal vertebrae of two brachyrostran abelisaurids.** The camellate tissues is visible in the centrum of Abelisauridae indet. MPM 99 (A), and the transverse process of *Kurupi* (B). On the right, details of the camellate tissues in both specimens. Image not to scale.

articulated with one another, thus reducing further the mobility of the trunk by turning the backbone into a single rigid structure (*Filippi et al., 2016*). Surprisingly, *Aucasaurus* holds these processes on the sacral and caudal neural spines as well.

The sacrum is generally a rigid portion of the axial skeleton, due to several anatomical aspects such as its inclusion between the ilia and the partial or complete fusion of vertebrae. Abelisaurids—as in *Coelophysis*, *Syntarsus*, and *Masiakasaurus*—have sacral neural spines tightly fused to one another forming an anteroposterior wall (*Carrano & Sampson, 2008*). Moreover, some abelisaurids such as *Aucasasurus*, *Carnotaurus*, and MAU-Pv-LI 547 are characterized by having a transversely expanded dorsal end of the sacral neural spines with longitudinal lateral ridges, forming a T-like structure more conspicuous than that of other ceratosaurs (*e.g.*, *Masiakasaurus*, *Elaphrosaurus*, *Majungasaurus*; *Carrano & Sampson, 2008*). A similar T-like structure is recorded in the neural spines of some sauropods (*Cerda et al., 2015*; and references therein), but the origin and function of this condition is still debated. *Cerda et al. (2015)* proposed a ligamentous origin for this structure, based on histological observations of sauropod specimens. However, cartilaginous (*Bonaparte, 1996*) or tendinous (*Giménez, Salgado & Cerda, 2008*) origins were also suggested for the supraspinous rod that connects the dorsal portion of the neural spines of the sacral vertebrae. The function (and homology) of the supraspinous ligamental ossification is so far unknown, but it may be related to stressing forces acting on this region of the skeleton (*e.g.*, tensile forces; *Cerda et al., 2015*); however, it is not clear to

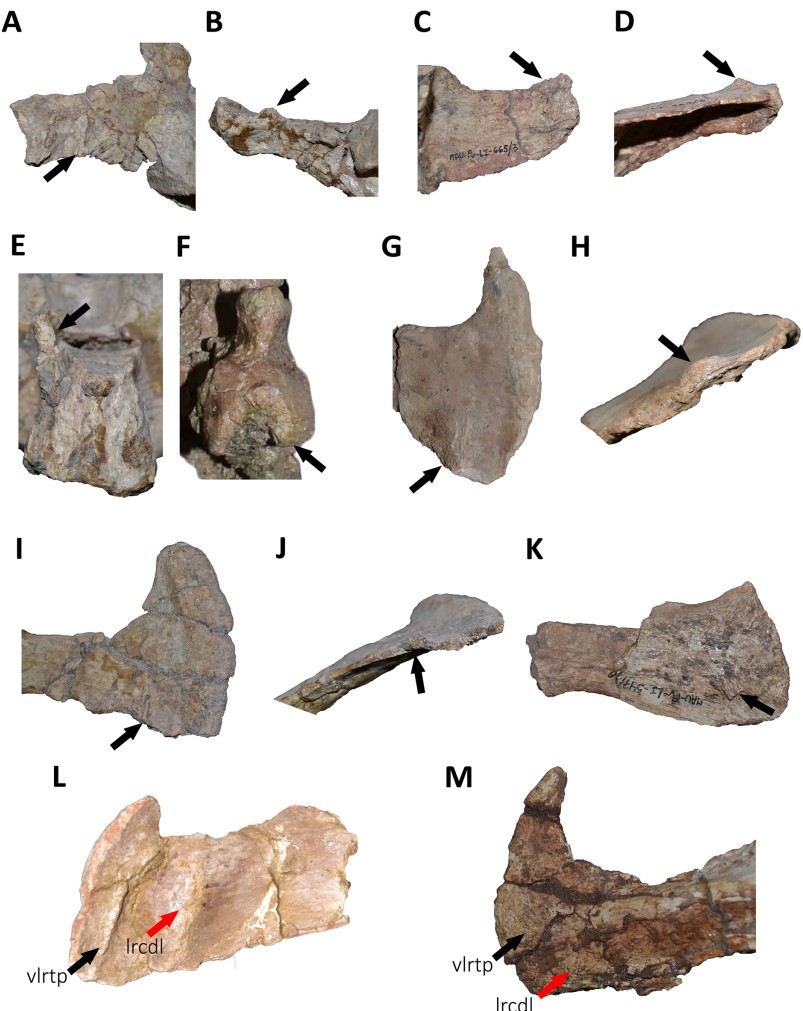

**Figure 34 Details of the dorsal and caudal vertebrae of several abelisaurids.** Structures on the dorsal surface of the transverse process in the second dorsal vertebra of *Viavenator* (A and B), and anterior dorsal vertebra of MAU-Pv-LI 665 (C and D). Interspinous accessory process on the dorsal neural spine of the fourth dorsal vertebra of *Viavenator* (E), and *Aucasaurus* (F). Scar (black arrows) on the dorsal surface of the mid caudal transverse processes of *Aucasaurus* (G and H), *Viavenator* (I and J), and MAU-Pv-LI 547 (K). Ventrolateral ridge (black arrows) of the transverse process in *Aucasaurus* (L), and *Viavenator* (M) (lateral ridges of centrodiapophyseal lamina indicate by red arrows). Image not to scale.

what extent this condition has an ontogenetic component (*Cerda et al., 2015*). Up to now, this portion of the sacral neural spines are unexplored histologically, but the morphological similarity among sauropods and some abelisaurids (*e.g., Aucasaurus, Carnotaurus*) suggest a similar developmental origin. Thereby, the T-like structure plus accessory interspinous processes of the dorsal vertebra of *Viavenator*, and the dorsal, sacral, and caudal vertebrae of *Aucasaurus*, could well be related to the ossification of supraspinous ligaments, as it was also proposed for some sauropod dinosaurs (*Cerda et al., 2015*). Further histological and biomechanical studies of these structures are likely to shed additional light on the stiffening of the axial skeleton of abelisaurids, in turn contributing to a better understanding of their locomotory and postural role in these theropods.

The morphology of the caudal vertebrae of brachyrostran abelisaurids has been underscored by several studies due to the specialized morphology of their transverse processes (*e.g.*, *Persons & Currie, 2011*). Whitin Brachyrostra, the caudal transverse processeses can adopt two morphologies: (1) the presence of an anteroposteriorly developed lateral end (Coria & Salgado, 2000; *Calvo, Rubilar-Rogers & Moreno, 2004*; *Canale et al., 2009*; *Cerroni et al., 2020*; *Gianechini et al., 2022*) or (2) the presence of a lateral end with an anterior awl-like process (*e.g.*, *Bonaparte, Novas & Coria, 1990*; *Coria, Chiappe & Dingus, 2002*; *Ezcurra & Méndez, 2009*; *Méndez, 2014b*; *Filippi et al., 2016*; *Delcourt, 2017*). These distinct morphologies along with other previously highlighted traits (*e.g.*, hyposphene-hypantrum articulation, large and dorsally inclined transverse processes, robust cdl, acdl, and pcdl) suggest that the tail of these dinosaurs was somewhat rigid, at least along its proximal and middle regions (*Persons & Currie, 2011*; *Méndez, 2014b*). We propose new traits of the caudal vertebrae that support a significant stiffening of the tail. The caudal vertebrae of the abelisaurids *Aucasaurus*, *Carnotaurus*, *Viavenator*, and MAU-Pv-LI 547 have a rough scar near the posterolateral rim of the dorsal surface of the transverse process (Figs. 34G–34K). This scar is visible up to the twelfth caudal in *Aucasaurus* (posterior to this it is unknown due to preservation). Another scar is located more medially in *Aucasaurus*, and this structure is extremely developed in *Kurupi* (cuneiform process of transverse process in *Iori et al., 2021*) (Fig. 30F). Derived brachyrostrans (*e.g.*, *Aucasaurus*, *Carnotaurus*, and *Viavenator*) also show a marked boundary between *m. ilio-ischiocaudalis* and *m. longissimus*, due to the presence of ornamentation on the lateral rim of the transverse processes (more evident in *Aucasaurus*). Finally, *Aucasaurus*, *Carnotaurus*, *Ekrixinatosaurus*, *Viavenator*, and other abelisaurids (*e.g.*, MAU-Pv-LI 547, MACN-PV-RN 1012) have an accessory longitudinal ridge (vlrtp) on the lateroventral end of the transverse processes (Figs. 34L and 34M). These dorsal and ventral ridges, and scars, suggest strong ligamental attachments and insertion points for the epaxial and hypaxial musculature of the caudal vertebrae (*e.g.*, *m. transversospinalis*, *m. longissimus*, *m. ilio-ischiocaudalis*, and particularly *m. caudofemoralis*) (*Persons & Currie, 2011*). We believe that such degree of caudal musculature (*Persons & Currie, 2011*), in addition to the overlapped lateral transverse processes (*e.g.*, *Persons & Currie, 2011*; *Cerroni et al., 2020*), would have rendered an extremely rigid tail in some brachyrostran abelisaurids, an interpretation that is congruent with previously proposed paleobiological interpretations of some abelisaurids as fast-runners/powerful sprinters (*Bonaparte, Novas & Coria, 1990*; *Mazzetta, Fariña & Vizcaíno, 1998*; *Persons & Currie, 2011*). Interestingly, several authors (*Dollo, 1886*; *Organ, 2006a*) have considered the stiffness of the tail ornithopod dinosaurs, *via* ossified tendons, as a response to the forces generated by retractor muscles of the femur (*e.g.*, *m. caudofemoralis*), which pulls back this bone (*Organ, 2006a*) and gives stability to the tail (*Siviero et al., 2020*). Despite the fact that ossified tendons are so far unknown in non-avian theropods, these mineralized structures are common among birds (*e.g.*, *Organ, 2006b*). These structures stiff the axial skeleton, while storing elastic energy and redistributing internal forces (*Organ, 2006a*, *2006b*). *Wilson et al. (2016)* claim similar functions for supraspinous anterior and posterior bone outgrowths (mineralized supraspinous ligament *via* metaplasia) of dorsal neural spines in some

non-avian theropods. However, *Wilson et al. (2016)* stated that the presence of the mineralized supraspinous ligament is a body-size and ontogenetic-dependent factor, since these structures are present in large non-avian theropods (*Foth et al., 2015*) and increase throughout ontogeny. The axial skeleton of abelisaurids shows traits that appear analogous to ossified tendons (as well as to the notarium and expanded synsacrum of living birds). Namely, these traits include accessory interspinous processes, procumbent osteological correlates of the epaxial musculature (*e.g.*, longitudinal ridge on the dorsolateral surface of dorsal transverse processes), and completely fused sacral vertebrae with a dorsal swelling of the neural spines. Further studies of the myological correlates of the vertebral column of these theropods may confirm or rebut previously proposed paleobiological inferences.

## CONCLUSIONS

Our detailed study of the axial skeleton of the abelisaurid *Aucasaurus garridoi* allowed us to expand the original diagnosis of this species. On the basis of the information gathered from the axial skeleton, *Aucasaurus garridoi* is distinguished by a unique combination of characters (plus the autapomorphy proposed by *Coria, Chiappe & Dingus, 2002*) including (1) atlas with a subcircular articular surface; (2) interspinous accessory processes extended to sacral and caudal neural spine; (3) a tubercle lateral to the prezygapophysis of middle caudal vertebrae (a similar structure is mentioned in *Aoniraptor*, *Motta et al., 2016*); (4) presence of pneumatic foramina laterally to the base of the neural spine in the anterior caudal vertebrae; (5) a prominent tubercle and extensive rugosity on the lateral rim of the transverse processes of caudal vertebrae fourth to twelfth; (6) presence of a small ligamentous scar near the anterior edge of the dorsal surface in the anteriormost caudal transverse processes; and (7) distinct triangular process located at the fusion point of posterior gastralia.

Our phylogenetic analysis allowed us to recognize several new axial characters, and to detect apomorphic conditions shared by *Aucasaurus* and other abelisaurid taxa. The phylogeny presented here confirms the position of *Aucasaurus* among derived abelisaurids; our results recover *Aucasaurus* as a brachyrostran furileusaurian, although in a polytomy with other abelisaurids.

The presence of a pair of foramina laterally to the neural spines, a foramen inside the pocdf (the latter trait is shared with other abelisaurids, such as *Carnotaurus*), and camellated tissue at the base of the neural spines and internally to the caudal vertebrae, are among key features of the axial skeleton of *Aucasaurus garridoi* suggesting the extension of pneumaticity into the caudal series. We hypothesize that the pneumaticity of the caudal portion of the axial skeleton of several brachyrostran abelisaurids (*e.g.*, *Aucasaurus*, *Kurupi*, and MPM 99) was independently acquired along the Brachyrostra lineage. Likewise, the phylogenetic distribution of pneumaticity in caudal vertebrae suggest that the varying degrees of pneumatization in Megaraptora, Carcharodontosauria, Ornithomimosauria, and Therizinosauroidea evolved independently in each of these clades.

We also analyse some traits that possibly increased the stiffness of the axial skeleton of abelisaurids. We recognize several traits (in some cases known only for a singular taxon) as

related to attachment points for ligaments, which in turn would have increased the rigidity of the axial skeleton. These traits include the presence of a ridge on the dorsal surface of the transverse processes of dorsal vertebrae (*e.g.*, *Viavenator*) and the presence of a scar on the posterolateral portion of the dorsal surface of the transverse processes of caudal vertebrae (*e.g.*, *Aucasaurus*, *Carnotaurus*, *Viavenator*).

This study is the second detailed description of the axial skeleton of an abelisaurid theropod, after *O'Connor's (2007)* description of *Majungasaurus*, which delves into the pneumaticity and stiffness of the vertebral column. The detailed information provided here is expected to contribute to our understanding of the paleobiology and paleoecology of abelisaurid theropods.

## INSTITUTIONAL ABBREVIATIONS

**MACN**  Museo Argentino de Ciencias Naturales "Bernardino Rivadavia", Buenos Aires, Argentina.

**MAU**  Museo Municipal Argentino Urquiza, Rincón de Los Sauces, Argentina.

**MCF**  Museo Carmen Funes, Plaza Huincul, Argentina.

**MHNA**  Muséum d'Histoire Naturelle d'Aix-en-Provence, Aix-en-Provence, France.

**MMCh**  Museo Municipal "Ernesto Bachmann", Villa El Chocón, Argentina.

**MPCA**  Museo Provincial Carlos Ameghino, Cipolletti, Argentina.

**MPCN**  Museo Paleontológico de Ciencias Naturales, General Roca, Argentina.

**MPEF**  Museo Paleontologico Egidio Feruglio, Trelew, Argentina.

**MPM**  Museo Regional Provincial "Padre Manuel Jesús Molina", Río Gallegos, Argentina

**MUC**  Museo Universidad Nacional del Comahue, Neuquén, Argentina

**UNPSJB**  Universidad Nacional de la Patagonia San Juan Bosco, Comodoro Rivadavia, Argentina.

## ACKNOWLEDGEMENTS

We want to thank D. Pol (MPEF-CONICET), J. I. Canale (MMCh-CONICET), M. Ezcurra (MACN-CONICET), J. L. Carballido (MPEF-CONICET), J. Calvo (UNCo), G. Casal (UNPSJB), L. Ibiricu (CENPAT-CONICET), M. Luna (UNPSJB), P. Chafrat (MPCN), I. A. Cerda (IIPG-CONICET), C. Aguilar (MPM), M. Gutiérrez (MCF), L. Filippi (MAU), C. Fuentes (MAU), A. Garrido (MOZ), Y. Dutour (MHNA), T. Tortosa (MHNA), M. A. Rolando (MACN-CONICET), F. V. Iori (MPMA), T. S. Marinho (UFTM), and N. E. Jalil (MNHN), for providing access to specimens under their care. D. Pol, J. I. Canale., M. Ezcurra, S. Apesteguía (Fundación Azara-CONICET), F. Gianechini (UNSL-CONICET), M. Cerroni (MACN-CONICET), and A. Cau provided comments to an earlier version of the manuscript. We are grateful to the Sanatorio de Plaza Huincul and technician G. Iril for providing access to a CT scanner and for assisting us during the scanning process. We are grateful to S. Brusatte (UE) and M. Ellison for providing photos of the holotype specimen of *Alioramus altai* (IGM 100/1844). Thanks to S. Hartman (UW) for permission to use the *Aucasaurus* silhouette (https://www.skeletaldrawing.com/). We also

acknowledge CONICET, Museo Municipal Carmen Funes, and Universidad Nacional de Río Negro for logistical support, and the contributions of Academic Editor Mathew Wedel and reviewers Ariel Méndez and Juan I. Canale—their insightful comments greatly improved the quality of the manuscript.

### Funding

This work was supported by the International Research Program-Sepkoski. Grants 2021 (Paleontological Society) to Mattia Antonio Baiano. There was no additional external funding received for this study The funders had no role in study design, data collection and analysis, decision to publish, or preparation of the manuscript.

### Grant Disclosures

The following grant information was disclosed by the authors:
International Research Program-Sepkoski.
Grants 2021 (Paleontological Society) to Mattia Antonio Baiano.

### Competing Interests

Luis Chiappe is an Academic Editor for PeerJ.

### Author Contributions

- Mattia Antonio Baiano conceived and designed the experiments, performed the experiments, analyzed the data, prepared figures and/or tables, authored or reviewed drafts of the article, and approved the final draft.
- Rodolfo Coria conceived and designed the experiments, analyzed the data, authored or reviewed drafts of the article, and approved the final draft.
- Luis M. Chiappe conceived and designed the experiments, analyzed the data, authored or reviewed drafts of the article, and approved the final draft.
- Virginia Zurriaguz conceived and designed the experiments, performed the experiments, analyzed the data, prepared figures and/or tables, authored or reviewed drafts of the article, and approved the final draft.
- Ludmila Coria conceived and designed the experiments, performed the experiments, analyzed the data, prepared figures and/or tables, and approved the final draft.

### Data Availability

   The raw measurements are available in the Supplemental File.

### Supplemental Information

Supplemental information for this article can be found online at http://dx.doi.org/10.7717/peerj.16236#supplemental-information.

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
