# Peer review of "Osteology of the axial skeleton of Aucasaurus garridoi: phylogenetic and paleobiological inferences"

_PeerJ, doi:10.7717/peerj.16236_

## Round 0.1 · original submission · Minor Revisions

The comments by the reviewers seem both reasonable and straightforward. I look forward to seeing an even better version of this manuscript soon!

·

Basic reporting

The English of the manuscript sounds clear and unambiguous to me, but given I am not a native speaker, I suppose my opinion is not very relevant at this point.
Also I provide here a list of errors I found on the text, which are listed following the lines of the .pdf archive:

Line 69: Striking instead of Stricking

Line 70: In particular or Particularly, not “in particularly”

Line 72: have preserved instead of preserved

Line 103: Río Gallegos

Line 105: Universidad Nacional de la Patagonia

Line 118: were taken, instead was taken

Line 142: Jackknife

Line 233: proatlas, instead protoatlas

Line 286: I think it should be “teardrop-like”

Line 415: anteroposteriorly instead anteroposterior

Line 614: I suggest to use “largest posterior surface”

Line 641: vertebrae instead of vertebra

Line 710: “…small depression posterior at the entry…” this sounds odd to me, please check the phrase

Line 779: I suggest to rewrite this sentence.

Line 836: spine instead of spines

Line 879: concave/convex

Line 1540: hyposphene-hypantrum

Line 1555: lesser medially inclined

Line 1580: Jackknife

Line 1750: taphonomic bias

Line 1771: Please capitalize Abelisauridae

Line 1839: blue

Line 1908: morphologies

Experimental design

No comment

Validity of the findings

No comment.

Additional comments

Dear Dr. Baiano and colleagues:
I had the pleasure of act as a revisor of your interesting manuscript entitled "Osteology of the axial skeleton of Aucasaurus garridoi (Coria, Chiappe and Dingus 2002): phylogenetic and paleobiological inferences", which was submitted to PeerJ. I really enjoyed reading it, I think it is a very welcome detailed description of the axial anatomy of such an important representative of the clade Abelisauridae, as it is Aucasaurus. It will represent a useful tool for future comparisons in theropod (and dinosaur) analysis. Also the phylogenetic analysis and the paleobiological inferences realized makes the paper even stronger. I will support its publication, after some minor changes, which are detailed in the "Basic reporting" section.
Anything else I can do for help, please let me know.
Sincerely yours.
Dr. Juan Ignacio Canale

·

Basic reporting

The work presented here is clearly written. The authors have extensive expertise in the subject of the article, which ensures a correct management of prior knowledge. The figures and tables are appropriate for detailed work such as the one presented here.

Experimental design

The research is clear as to its purpose. The data provided here are relevant for a better understanding of phylogenetic, morphological, evolutionary and functional issues of the family Abelisauridae.

Validity of the findings

Research conclusions are well supported

Additional comments

I consider that the publication of this work is of great importance, since it deals with a material briefly described more than 20 years ago and provides new and relevant information about the members of the Abelisauridae clade.

---

## Round 0.2 · accepted · Accept

Thank you for your diligence in responding to the reviewers' suggestions. My only advice is to double-check that all of your reasoning is as clear in the manuscript as it is in the rebuttal letter. You'll have an opportunity to make any last changes as you handle the production tasks and at the proof stage. But in my view, the manuscript is ready for publication. Thanks again, and congratulations.